# scFFPE-ATAC enables high-throughput single cell chromatin accessibility profiling in formalin-fixed paraffin-embedded samples

Ram Prakash Yadav[1,3], Pengwei Xing[1,3], Miao Zhao [1], Peter Hollander[1], Carina Strell [1,2], Minglu Xie[1], Maede Salehi[1], Emma Torell[1], Tobias Sjöblom [1], Gunilla Enblad [1], Rose-Marie Amini [1], Fredrik Johansson Swartling [1], Ingrid Glimelius[1], Patrick Micke [1], Mats Hellström[1] & Xingqi Chen [1] ✉

Formalin-fixed paraffin-embedded (FFPE) samples are the gold standard for tissue preservation in clinical and research settings. Current single-cell chromatin accessibility technologies cannot resolve cell-type-specific epigenetic profiles in FFPE tissues due to extensive DNA damage. We present scFFPE-ATAC, a high-throughput single-cell chromatin accessibility assay for FFPE samples that integrates an FFPE-adapted Tn5 transposase, ultra-high-throughput DNA barcoding (>56 million barcodes per run), T7 promoter-mediated DNA damage repair, and in vitro transcription. We benchmark scFFPE-ATAC on FFPE mouse spleen and validate its performance against fresh tissue. We apply it to human lymph node samples archived for 8–12 years and to lung cancer FFPE tissues, revealing distinct regulatory trajectories between tumor center and invasive edge. Analysis of archived follicular lymphoma and transformed diffuse large B-cell lymphoma samples identifies relapse- and transformation-associated epigenetic dynamics. scFFPE-ATAC enables retrospective, spatial, and mechanistic epigenetic studies in long-term archived specimens.

Formalin-fixed paraffin-embedded (FFPE) preservation has been the gold standard for archiving clinical and biomedical samples for over 130 years[1]. In clinical practice, more than 99% of patient-derived samples are stored in FFPE format[2]. Over 400 million to 1 billion FFPE tissue samples are archived in hospital pathology departments and hospitals all over the world[3,4], making it an invaluable resource for both basic research and retrospective studies. Epigenetics, which investigates stable phenotypic changes without alterations in DNA sequence, plays a crucial role in understanding gene regulation[5]. At the core of epigenetic regulation is chromatin accessibility, which governs gene expression by modulating the interaction between transcription factors and DNA[6–10]. Large-scale retrospective studies of epigenetic regulation and genomic in FFPE samples, particularly when combined with clinical and pathological records, have the potential to provide

critical insights into human diseases, including cancer[11–14]. Tumor relapse and metastasis remain major challenges in cancer treatment[15–18]. However, metastasis-specific mutations have not been consistently identified across all tumor types[19,20]. Instead, copy number variations have been observed[14,19]. The reversible nature of epigenetic modifications likely plays a key role in metastasis. Therefore, profiling epigenetic landscapes in paired primary and relapse tumor samples is essential for deciphering the underlying mechanisms of tumor progression and treatment resistance[21,22]. Since it is hard to predict when and whether relapse or metastasis will occur in patients, these paired samples are typically preserved in FFPE format. It is of paramount importance to establish single-cell epigenetic profiling technologies that can work with FFPE samples. Many studies[13,23–27], including our own, have demonstrated that chromatin structure remains intact in

[1]Department of Immunology, Genetics and Pathology, Uppsala University, Uppsala, Sweden. [2]Centre of Cancer Biomarkers (CCBIO), Department of Clinical Medicine, University of Bergen, Bergen, Norway. [3]These authors contributed equally: Ram Prakash Yadav, Pengwei Xing. ✉e-mail: xingqi.chen@igp.uu.se

FFPE samples despite formalin fixation. Various highly sensitive bulk epigenetic profiling technologies have been developed for FFPE samples[23,25–29], such as our recently developed FFPE-ATAC and FACT-Seq, by using the strategy of transposase-mediated accessible chromatin profiling. However, these methods provide only bulk population-level insights and lack single-cell resolution. Given the well-established heterogeneity of tumor cells[30,31], single-cell resolution is crucial for accurately capturing the complexity of the tumor micro-environment. Unlike single-cell transcriptomic analysis, which has already been successfully implemented in clinical FFPE samples[32–34], single-cell chromatin accessibility profiling has remained a significant challenge due to extensive DNA damage caused by formalin fixation and parafin embedding.

To overcome this technical barrier, we introduce scFFPE-ATAC, a high-throughput single-cell chromatin accessibility assay for FFPE samples. scFFPE-ATAC features a newly designed FFPE-Tn5 transposase, high-throughput DNA barcoding with over 56 million cell barcodes per run, T7 promoter-mediated DNA damage rescue, and in vitro transcription. This resulted in the establishment of scFFPE-ATAC, the single-cell chromatin accessibility profiling technology for FFPE archived biomedical and clinical samples from epigenetic regulation at single-cell resolution. Our scFFPE-ATAC method operates robustly across a wide range of FFPE sample formats—including FFPE punch cores and FFPE tissue sections—providing an unprecedented opportunity to decode tumor epigenetic heterogeneity at the single-cell level. We benchmarked scFFPE-ATAC using mouse FFPE spleen samples, comparing them to fresh mouse tissue. We also successfully applied scFFPE-ATAC to clinically archived FFPE human lymph node samples stored for 8–12 years, demonstrating its ability to resolve single-cell chromatin landscapes from archived tissues. As part of our validation, we applied this technology to FFPE human lung cancer samples, comparing the chromatin accessibility profiles of epithelial cells from the tumor center and invasive edge. This analysis uncovered spatially distinct epigenetic regulators and revealed two distinct developmental paths from the tumor center to the invasive edge, each enriched for unique gene regulatory programs and epigenetic mechanisms. Additionally, as a case study, we used scFFPE-ATAC to investigate tumor relapse by analyzing FFPE clinical tumor samples from one patient with paired primary follicular lymphoma (FL) and relapsed FL with a 2-year interval, as well as from another patient with FL that had transformed into diffuse large B-cell lymphoma (DLBCL) over a 7-year interval. This enabled us to identify patient-specific epigenetic regulators driving tumor relapse and transformation in a real clinical setting using long-term archived FFPE samples.

Overall, our scFFPE-ATAC enables high-throughput, high-sensitivity chromatin accessibility analysis in long-term archived biomedical and clinical FFPE specimens. It paves the way for both basic research and retrospective epigenetic studies, providing deeper insights into tumor progression, relapse, and metastasis. Furthermore, scFFPE-ATAC lays the foundation for spatial epigenetic profiling and multi-omics integration in FFPE samples, ultimately advancing the field of basic research and personalized medicine.

## Results

### Conventional scATAC-Seq fails to resolve cell-type-specific epigenetic profiles in FFPE samples

To profile single-cell chromatin accessibility in FFPE samples, obtaining high-quality nuclei is critical. Unlike fresh/frozen samples, the harsh treatments involved in FFPE sample preparation, including formalin fixation and paraffin embedding, present significant challenges. We followed published protocols[25,26,35,36] to isolate nuclei from FFPE samples and observed the presence of cellular debris in the isolated nuclei. The presence of debris affects nuclei counting for single-cell assays as well as downstream chemical reactions. Several approaches were used to enrich nuclei and reduce debris, including Fluorescence-

Activated Cell Sorting (FACS)[14,37,38] and density gradient centrifugation[39,40]. While FACS can achieve high-purity nuclei by gating out debris and aggregates, this approach requires specialized instrumentation and expertise that were not available to our group when this study was initiated. We therefore focused on optimizing density gradient centrifugation, which provides a robust and broadly accessible strategy. To remove debris, we applied density gradient centrifugation using the suggested density gradient layers (25%-30%-40%) for fresh/frozen nuclei[39], optimizing the procedure with purified mouse FFPE spleen nuclei. Although we successfully obtained the nuclei layer (between the 30% and 40% interface) after density gradient centrifugation (Supplementary Fig. 1a), unlike with fresh/frozen nuclei, extracellular and cellular debris could not be removed from the nuclei layers in FFPE samples. We reasoned that the density of nuclei and extracellular and cellular debris might change following formalin fixation and paraffin embedding, making it difficult to separate them. Thus, we created a finer density gradient layer between 30% and 40% to separate nuclei from extracellular and cellular debris in FFPE samples (Methods). With further optimization with FFPE nuclei, we observed the formation of two distinct layers in the gradient solution containing 25%, 36%, and 48% density gradients after density gradient centrifugation (Supplementary Fig. 1a). FFPE samples exhibited a bottom layer (between the 36% and 48% interface) consisting of a large amount of cellular debris and few extracellular matrix, while the top layer (between the 25% and 36% interface) contained pure nuclei (Supplementary Fig. 1b). However, only a single top layer containing nuclei was observed in the fresh sample (Supplementary Fig. 1a). This observation indicates that purified nuclei from FFPE samples are lighter than the cellular debris and extracellular matrix and remain in the upper layer after density gradient centrifugation—a distribution that differs from nuclei purified from fresh samples. The high quality and purity of single FFPE nuclei obtained from the top layer after density gradient centrifugation enable us to perform single-cell experiments on FFPE samples.

Next, we used purified mouse FFPE spleen nuclei to perform standard ATAC-Seq[40,41] and single-cell ATAC-Seq (scATAC-Seq) with split-and-pool barcoding[9,42,43] (Fig. 1, and Supplementary Fig. 2). Previous studies[44], including our own, have shown that a reverse cross-linking step is necessary to remove formaldehyde fixation from fixed samples. The DNA length distribution from purified mouse FFPE spleen nuclei with and without reverse crosslinking (+/−RV) differs from that of DNA obtained from fresh samples (Supplementary Fig. 2a). In the −RV condition, only short DNA fragments (ranging from 50 to 300 bp) are enriched. Both long and short DNA fragments are recovered in the +RV condition (Supplementary Fig. 2a), this suggests substantial DNA damage and fragmentation occur during reverse crosslinking and purification in FFPE samples. To obtain a comprehensive view, our standard bulk ATAC-Seq and scATAC-Seq analyses included both conditions: with reverse crosslinking (+RV) and without reverse crosslinking (−RV). We compared both bulk ATAC-Seq and scATAC-Seq data from FFPE nuclei to data from fresh samples (Fig. 1a, and Supplementary Fig. 2b, c). After PCR amplification, only short DNA fragments were clearly observed in FFPE samples under both +RV and −RV conditions (Supplementary Fig. 2b). This suggests that reverse crosslinking may help increase DNA yield in FFPE samples. However, DNA damage and fragmentation in these purified samples likely prevent the longer fragments from being effectively amplified during PCR.

Genome-wide correlation analysis of accessible chromatin peaks demonstrated high reproducibility in bulk ATAC-Seq datasets (Supplementary Fig. 2c). We also observed a strong correlation between +RV and −RV bulk ATAC-Seq samples (Pearson correlation = 0.94), as well as between merged single-cell +RV and −RV datasets (Pearson correlation = 0.94). These results suggest that the reverse crosslinking condition in FFPE samples does not improve the recovery of accessible chromatin regions. This is in contrast to mild fixation conditions—such

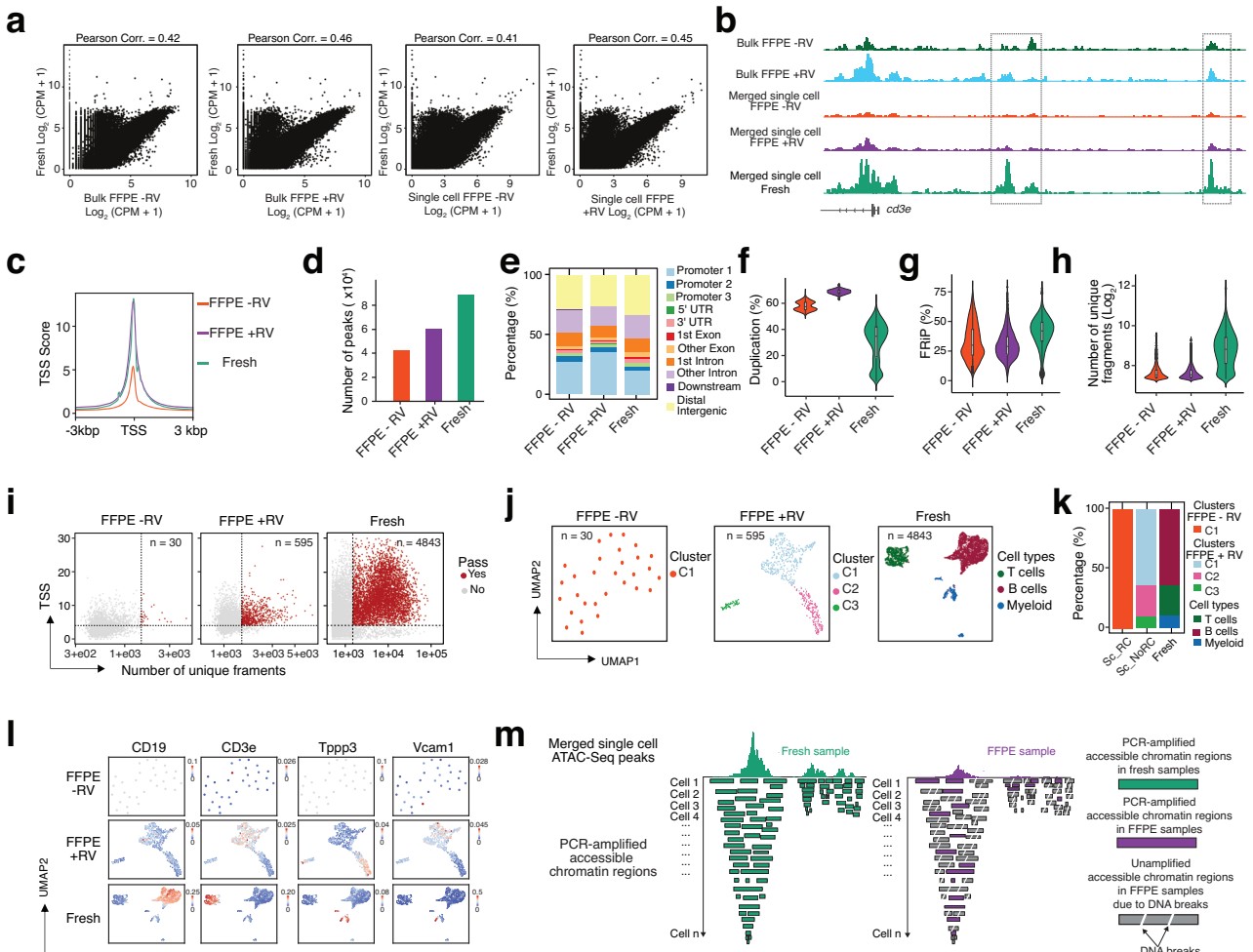

**Fig. 1 | Profiling single-cell chromatin accessibility in FFPE samples with conventional scATAC-Seq. a** Genome-wide comparison of chromatin accessibility reads from mouse spleen FFPE samples with reverse croslinking (FFPE + RV) or without reverse crosslinking (FFPE -RV), and fresh spleen samples, prepared using conventional bulk and single-cell ATAC-Seq. **b** Representative genome browser tracks at the T-cell marker gene *Cd3e* in mouse FFPE and fresh spleen samples, prepared using conventional bulk and single-cell ATAC-Seq; dotted squares highlight peaks present in fresh samples but absent in FFPE samples. **c** TSS enrichment in mouse FFPE samples with reverse crosslinking (FFPE + RV) or without reverse crosslinking (FFPE -RV) and fresh spleen samples, assayed by conventional single-cell ATAC-Seq. **d–h** Number of ATAC-Seq peaks (**d**) genomic annotation of ATAC-Seq peaks (**e**) duplication rate (**f**) fraction of reads in peaks (FRiP) (**g**) and number of unique fragments per cell **h** in mouse FFPE samples with reverse crosslinking (FFPE + RV) or without reverse crosslinking (FFPE -RV) and fresh spleen samples, assayed by conventional single-cell ATAC-Seq. Box boundaries in panels (**f**–**h**) represent the interquartile range (IQR), spanning from the first quartile (Q1) to the third quartile (Q3). Duplication rate (**f**)—Fresh (n = 5,258, max = 59.59, min = 0.19, Q1 = 18.77, median = 34.73, Q3 = 42.02); FFPE + RV (n = 2,687, max = 74.35, min =

63.00, Q1 = 67.62, median = 68.85, Q3 = 70.10); FFPE − RV (n = 1980, max = 68.96, min = 50.34, Q1 = 55.11, median = 57.41, Q3 = 60.83). FRiP per cell (**g**)—Fresh (n = 5258, max = 78.11, min = 2.72, Q1 = 33.38, median = 42.01, Q3 = 49.29); FFPE + RV (n = 2687, max = 79.05, min = 4.59, Q1 = 22.24, median = 28.93, Q3 = 37.15); FFPE − RV (n = 1980, max = 75.71, min = 3.07, Q1 = 21.47, median = 29.94, Q3 = 43.30). Fragments per cell (**h**)—Fresh (n = 5258, max = 145,645, min = 1502, Q1 = 3303, median = 6722.5, Q3 = 12,302); FFPE + RV (n = 2687, max = 11,046, min = 1501, Q1 = 1646, median = 1873, Q3 = 2354.5); FFPE − RV (n = 1980, max = 13,354, min = 1501, Q1 = 1632.5, median = 1865, Q3 = 2366). **i** Scatter plots show number of cells identified with same parameter cutoff (TSS enrichment score and number of unique fragments). **j, k** Number of clustered cells **j** and their proportional distribution **k** identified from conventional single-cell ATAC-seq in fresh and FFPE samples. **l** Projection of gene activity across different cell clusters or cell types under varying conditions. **m** Schematic illustrating that DNA breaks in FFPE samples contribute to low library complexity at the single-cell level (Right panel) compared with fresh samples (Left panel), as detected by conventional scATAC-Seq. Source data are provided as a Source Data file for Fig. 1a, c–l.

as 10-minute fixation in cultured cells or fresh tissues—where chromatin accessibility is better preserved[44]. As expected, the merged scATAC-Seq data from FFPE samples showed good correlation with bulk data under both +RV and −RV conditions (Pearson correlation = 0.96 and 0.93, respectively), supporting the effectiveness of our split-and-pool single-cell strategy for FFPE-isolated nuclei. However, the genome-wide correlation of accessible peak regions between FFPE and fresh samples was much lower under both +RV and −RV conditions, at both bulk and merged single-cell levels (Fig. 1a; Pearson correlation ranging from 0.41 to 0.46). A large number of accessible chromatin

peaks were detected only in fresh samples but not in FFPE samples (Figs. 1a, b; and Supplementary Fig. 2d, e), even though the captured peaks in FFPE samples were enriched at transcription start sites (TSS) (Fig. 1c; and Supplementary Fig. 2f). Notably, the TSS enrichment score was higher under the +RV condition compared to the −RV condition. Additionally, insert size distribution analysis confirmed that only short DNA fragments were enriched in FFPE samples under both +RV and −RV conditions (Supplementary Fig. 2g). We did not observe a significant enrichment of longer DNA fragments in the +RV condition compared to the −RV condition. This further supports the conclusion

that while reverse crosslinking may help increase DNA yield in FFPE samples, it does not enhance the enrichment of accessible chromatin regions due to extensive DNA fragmentation and damage. This observation is also reflected in the number of peaks identified in FFPE samples: 32,512 peaks in the −RV condition and 47,809 peaks in the +RV condition, compared to 89,328 accessible chromatin peaks in fresh samples (Supplementary Fig. 2h). Importantly, the distribution of ATAC-Seq peaks in the genome from FFPE samples differs from that in fresh samples (Supplementary Fig. 2i). A similar pattern was observed in the merged scATAC-Seq data from FFPE samples (Figs. 1d, e). Furthermore, the complexity of the sequencing library was comparable between the +RV and −RV conditions in FFPE samples, but much lower than that of fresh samples (Supplementary Fig. 2j). Taken together, our data strongly suggest that the reverse crosslinking (RV) condition does not fully restore the profile of accessible chromatin regions when using conventional bulk or single-cell ATAC-seq. This limitation is likely due to DNA damage and fragmentation inherent to FFPE samples.

Next, we focused on the analysis of conventional scATAC-Seq in FFPE mouse spleen, comparing it in parallel with scATAC-Seq in fresh mouse spleen (Fig. 1f–l; and Supplementary Figs. 3a). To maximize library complexity, read-length parameters for mapping FFPE samples were optimized (see Methods). The number of decoded reads, unique fragments, and mapping rate (from 50 million reads) were calculated for each minimum length tested. Reducing the minimum fragment length from 50 bp to 14 bp increased the decoding rate from 48% (23.77 million) to 70% (35.10 million) and increased the number of unique fragments by 35.55% (from 18.18 million to 24.64 million), with only a minor decrease in the mapping rate (from 97.85% to 92.68%). To preserve uniquely mapped features and avoid potential multi-mapping fragments, a minimum read length of 17 bp was selected for our study (Supplementary Fig. 3b). This strategy maximized usable information from heavily fragmented FFPE DNA while maintaining high mapping quality. Considering the lower library complexity in FFPE samples, we sequenced them at two to three times greater depth compared to fresh samples (Supplementary Table 1). Using the same cutoff for the number of fragments per cell ($n \geq 1500$ per barcode), we identified only 1980 cell barcodes under the −RV condition and 2687 cell barcodes under the +RV condition in FFPE samples. In contrast, 5258 cell barcodes were recovered from fresh samples, despite with two to three times less sequencing depth (Supplementary Table 1). Furthermore, the median duplication rate reached 57.41% in the −RV condition and 68.85% in the +RV condition, compared to 34.73% in fresh samples (Fig. 1f). The median fraction of reads in peaks (FRiP) was 42.01% in fresh samples, which is higher than that observed in FFPE samples− 29.95% for −RV and 28.93% for +RV (Fig. 1g). The median number of final fragments per cell was 6722 in fresh samples, compared to only 1865 (−RV) and 1863 (+RV) in FFPE samples (Fig. 1h)−approximately 3.5 times higher than in FFPE samples. These results further highlight the reduced library complexity in FFPE samples when using conventional ATAC-Seq, both at the bulk and single-cell levels, compared to fresh tissue.

Next, we identified cells in FFPE samples (both −RV and +RV) that met the criteria of TSS enrichment score $\geq 4$ and unique fragments $\geq 1500$. This resulted in only 30 cells passing the cutoff in the −RV FFPE sample, 595 cells in the +RV FFPE sample, and 4843 cells in the fresh sample (Fig. 1i). We then identified cell types by combining high-dimensional clustering methods with marker gene activity analysis (Fig. 1j–l; Supplementary Fig. 4). In fresh samples, scATAC-Seq clearly resolved cell-type-specific epigenetic profiles by identifying T cells, B cells, and myeloid cells, as demonstrated by distinct clusters and predicted gene activity patterns. However, in the −RV FFPE samples, we were unable to identify specific clusters or cell types due to the limited number of cells passing quality filters−only 30 cells (Fig. 1j–l; and Supplementary Fig. 4). In the +RV FFPE sample, three subclusters were identified from 595 cells (Figs. 1j, k), and these clusters occupied

proportions similar to those observed in fresh spleen samples. However, gene activity analysis did not confirm that these clusters correspond to distinct spleen cell-type epigenetic features (Fig. 1j–l; and Supplementary Fig. 4).

This conclusion remained unchanged even when we lowered the filtering thresholds. Using TSS $\geq 4$ and unique fragments $\geq 500$ (Supplementary Fig. 5a–d), we extracted 876 cells and identified four clusters for the −RV condition, and 8777 cells with seven clusters for the +RV condition. With a stricter cutoff of TSS $\geq 4$ and unique fragments $\geq 1000$ (Supplementary Fig. 5e–h), 67 cells were extracted with no clusters identified for the −RV condition, while 1957 cells and seven clusters were identified for the +RV condition. However, in both parameter settings, no cell type–specific genes characteristic of mouse spleen were detected.

Taken together, we hypothesize that reverse crosslinking can increase DNA yield in FFPE samples; however, it does not improve ATAC-Seq library complexity in either bulk or single-cell experiments. While bulk ATAC-Seq can capture some accessible chromatin peaks despite the lower library complexity and DNA damage in FFPE samples, scATAC-Seq suffers from random DNA breaks in individual cells, resulting in reduced library complexity per cell (Fig. 1m). Consequently, conventional scATAC-Seq fails to resolve cell-type-specific epigenetic profiles in FFPE samples.

## Development of scFFPE-ATAC: A single-cell chromatin accessibility profiling method for FFPE samples

To bridge the technological gap, we developed a high-throughput single-cell chromatin accessibility profiling method for FFPE samples, termed scFFPE-ATAC (Fig. 2a). This method works robustly across various sample formats, including punch cores and tissue sections, enabling the decoding of epigenetic heterogeneity at the single-cell level.

We originally introduced FFPE-ATAC and FACT-Seq to profile the epigenetic code in FFPE samples, utilizing a T7 promoter-mediated Tn5 transposase and in vitro transcription (IVT) to rescue DNA damage[25,26]. Notably, the complexity of the sequencing libraries generated by FFPE-ATAC and FACT-Seq is higher than that of conventional ATAC-Seq and CUT&Tag. However, the sensitivity of these technologies does not achieve single-cell resolution. We hypothesized that multiple indexing of single cells with DNA barcodes, combined with T7 promoter-mediated DNA damage rescue, would enable the deciphering chromatin accessibility at single-cell resolution (Fig. 2a). To this end, we introduced a newly designed FFPE-Tn5 system, which combines a standard Tn5 transposase with a custom adaptor carrying 64 DNA barcodes for indexing different samples or spatial locations within FFPE samples (Supplementary Table 2). This is followed by three subsequent split-and-pool ligation steps using unique combinations of DNA barcode sequences (a combination of 96 × 96 × 96 DNA barcodes) to index individual cells. Together with FFPE-Tn5 indexes, this results in 56,623,104 cell barcodes in a single run. The T7 promoter sequence is positioned at the end of the third ligation barcode (Fig. 2a, and Supplementary Fig. 6). After the third ligation, each nucleus acquires a unique combination of barcodes and undergoes reverse cross-linking, which introduces DNA breakage. However, the T7 IVT system enables the generation of RNA molecules from all broken accessible chromatin sites where the three ligations occur, thereby rescuing the DNA breakage effect in FFPE nuclei. Since IVT transfers the DNA template randomly into RNA molecules of different lengths[25,26], some IVT molecules contain only the DNA barcode without the genome sequence of accessible chromatin. To minimize such cases, the length of our uniquely designed ligated oligos for ligation 1 and ligation 2 is limited to only 22 nt (Supplementary Table 2). Following IVT, we prepared sequencing libraries from IVT-RNA to profile both the DNA barcodes for each cell and its accessible chromatin sites. This streamlined design enables the indexing of 64 samples or

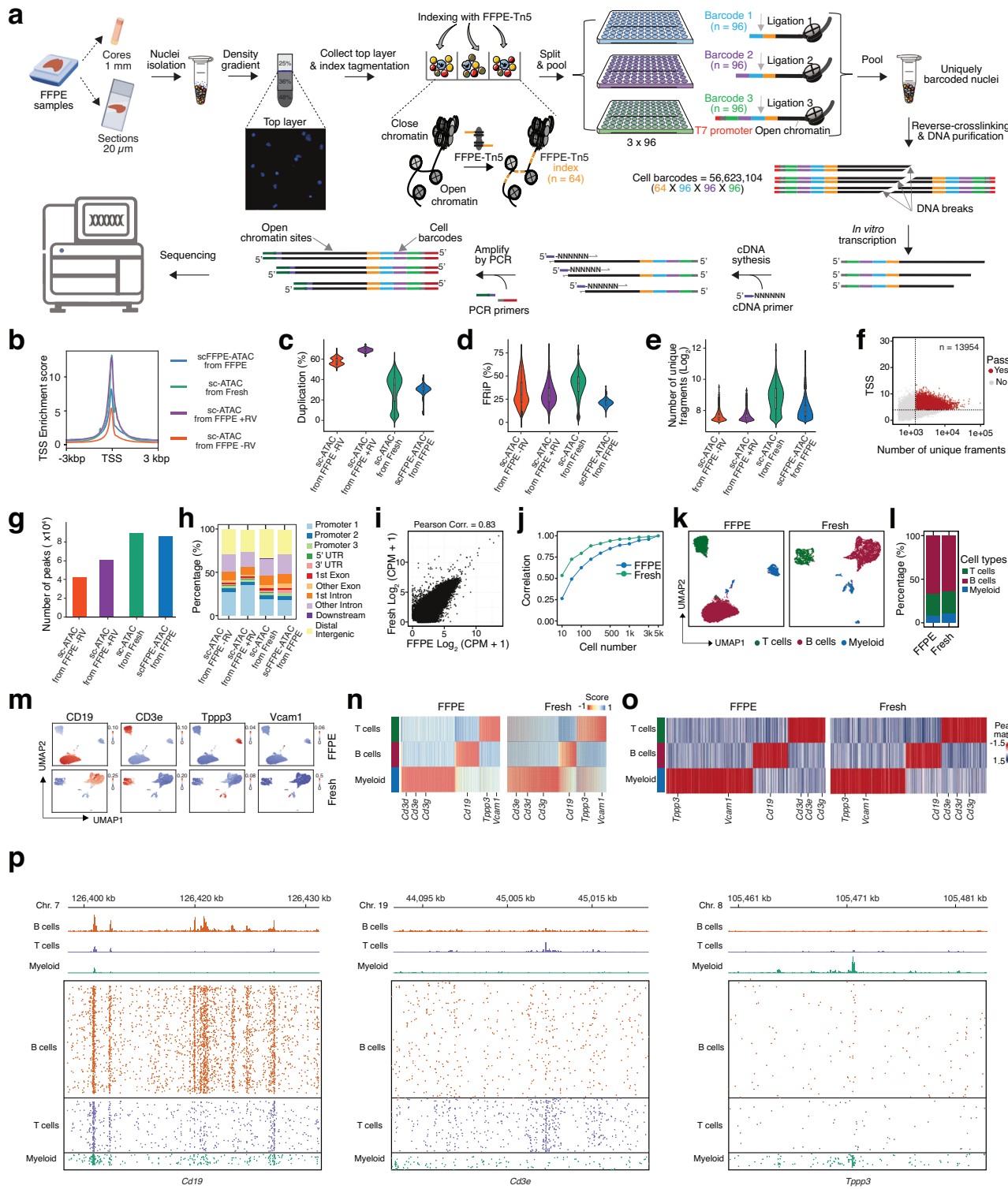

different sample locations, with a potential of 56,623,104 unique cell barcodes for FFPE samples.

After performing scFFPE-ATAC and decoding, we identified 18,200 cell-associated DNA barcodes (Supplementary Fig. 7a). We compared the single-cell chromatin accessibility data obtained from FFPE mouse spleen and fresh mouse spleen from multiple perspectives (Fig. 2b-p, Supplementary Fig. 7-10). The TSS enrichment score for the merged scFFPE-ATAC data from FFPE samples was 7.5, which was lower than that of fresh mouse spleen but maintained a similar TSS enrichment pattern (Fig. 2b). One potential explanation is that standard

scATAC-Seq relies on PCR amplification using two different Tn5 adaptors without additional DNA breaks or damage. In contrast, the T7 IVT-mediated method requires only a single T7 promoter adaptor insertion (Even there is DNA break/damage in between of two insertion) to amplify the insertion site. While this approach can rescue those DNA damage sites, it may also introduce noise into the sequencing library. Another possibility is that T7 IVT introduces sequence preference bias, which could interfere with library uniformity. The median duplication rates were comparable—30.81% for FFPE samples and 34.73% for fresh samples—under similar sequencing depth (Fig. 2c).

**Fig. 2 | scFFPE-ATAC unlocks single cell chromatin accessibility in FFPE samples. a** Schematic workflow of scFFPE-ATAC. **b–p** Comparison of scFFPE-ATAC from mouse FFPE spleen with scATAC-Seq from mouse fresh spleen and mouse FFPE spleen: **b** Sequencing signal enrichment at transcription start site (TSS). **c** Duplication rate comparison. **d** Fraction of reads in peaks (FRiP). **e** Number of unique fragments. Box boundaries in panels c–e represent the interquartile range (IQR), spanning from the first quartile (Q1) to the third quartile (Q3). Duplication rate (c)−scATAC from FFPE–RV ($n = 1980$, max = 68.96, min = 50.34, Q1 = 55.11, median = 57.41, Q3 = 60.83); scATAC from FFPE + RV ($n = 2687$, max = 74.35, min = 63.00, Q1 = 67.62, median = 68.85, Q3 = 70.10); scATAC from Fresh ($n = 5258$, max = 59.59, min = 0.19, Q1 = 18.77, median = 34.73, Q3 = 42.02); scFFPE-ATAC from FFPE ($n = 26,927$, max = 43.94, min = 5.85, Q1 = 28.77, median = 30.81, Q3 = 32.84). FRiP per cell (d)−scATAC from FFPE–RV ($n = 1980$, max = 75.71, min = 3.07, Q1 = 21.47, median = 29.94, Q3 = 43.30); scATAC from FFPE + RV ($n = 2687$, max = 79.05, min = 4.59, Q1 = 22.24, median = 28.93, Q3 = 37.15); scATAC from Fresh ($n = 5258$, max = 78.11, min = 2.72, Q1 = 33.38, median = 42.01, Q3 = 49.29); scFFPE-ATAC from FFPE ($n = 26,927$, max = 38.67, min = 6.54, Q1 = 18.59, median = 21.00, Q3 = 23.25). Fragments per cell (e)−scATAC from FFPE − RV ($n = 1980$, max = 13,354, min = 1501,

Q1 = 1632.5, median = 1865, Q3 = 2366); scATAC from FFPE + RV ($n = 2687$, max = 11,046, min = 1501, Q1 = 1646, median = 1873, Q3 = 2,354.5); scATAC from Fresh ($n = 5258$, max = 145,645, min = 1502, Q1 = 3303, median = 6722.5, Q3 = 12,302); scFFPE-ATAC from FFPE ($n = 26,927$, max = 118,714, min = 1501, Q1 = 2037, median = 2772, Q3 = 4115). **f** Parameters used for filtering low-quality single cells in scFFPE-ATAC analysis of mouse FFPE spleen tissue. **g** Number of peaks identified under each condition. **h** Genomic annotation of ATAC-seq peaks in each condition. **i** Genome-wide comparison of chromatin accessibility reads from mouse spleen FFPE samples using scFFPE-ATAC and from fresh mouse spleen samples using scATAC-seq. **j** Sensitivity comparison between scFFPE-ATAC on FFPE samples and scATAC-seq on fresh samples. Pearson correlation = correlation. **k** Identification of cell types from scFFPE-ATAC on FFPE samples and scATAC-Seq on fresh samples. **l** Proportional distribution of different cell types. **m** Examples of active genes in each cell type. **n** Identification of cell-type-specific active genes. **o** Cell-type-specific accessible ATAC-seq peaks under each condition. **p** Example of accessible chromatin peaks from genome browser tracks of merged single cells and individual cells for each cell type. Source data are provided as a Source Data file for Fig. 2b–o.

These rates are lower compared to those observed in conventional scATAC-Seq, which shows duplication rates of 57.41% under the −RV condition and 68.85% under the +RV condition. The median fraction of reads in peaks (FRiP) was 21% for FFPE samples, compared to 42% for fresh samples, 29.95% for the −RV condition, and 28.93% for the +RV condition (Fig. 2d). The median number of unique DNA fragments was 2722 for FFPE samples and 6722 for fresh samples (Fig. 2e), which is approximately 1.45 times higher compared to conventional scATAC in FFPE samples (1865 ( − RV) and 1863 ( + RV)). The increased number of unique DNA fragments in scFFPE-ATAC for FFPE samples compared to conventional scATAC-Seq indicates that the combination of the T7 promoter, IVT, and indexing barcode in scFFPE-ATAC indeed increase single-cell library complexity. The lower FRiP values and reduced number of unique fragments in FFPE samples compared to fresh samples may be attributed to prolonged formalin fixation, which can restrict the accessibility of Tn5 transposase to certain chromatin regions during tagmentation. Additionally, paraffin embedding and the harsh conditions of nuclei isolation may introduce side effects that negatively impact chromatin profiling. Similar to scATAC-seq in FFPE samples, the fragment distribution of the scFFPE-ATAC library was dominated by short fragments from FFPE samples (98.76% in the range of 0−300 bp for mouse spleen FFPE samples) (Supplementary Fig. 7b).

Using the same parameters as those applied to fresh samples and conventional scATAC-Seq for FFPE samples (unique fragments ≥1500 and TSS score ≥4), we obtained 13,954 cells exhibiting high-quality chromatin accessibility from mouse FFPE nuclei, representing 76.67% of all DNA barcodes in our assay, for subsequent downstream analysis (Fig. 2f). A total of 86,518 accessible chromatin peaks were identified in the merged scFFPE-ATAC data from FFPE samples, compared to 89,328 peaks in fresh samples (Fig. 2g). These peaks identified by scFFPE-ATAC in FFPE samples are distributed in similar proportions across different genomic regions in both FFPE and fresh samples (Fig. 2h). scFFPE-ATAC exhibits a strong genome-wide chromatin accessibility correlation with fresh samples (Pearson correlation = 0.83) (Fig. 2i). We also assessed the minimum number of cells required from scFFPE-ATAC in both FFPE and fresh samples to reliably capture accessible chromatin profiles by performing random downsampling and calculating correlations across a range of cell counts from 10 to 5000 (Fig. 2j). Data merged from only 200 cells in scFFPE-ATAC FFPE samples exhibited a strong correlation (Pearson correlation > 0.7) with data derived from 5,000 cells (Fig. 2j). Although the correlation in FFPE samples is not as strong as in fresh samples (Fig. 2j), the DNA damage repair provided by scFFPE-ATAC is sufficient to capture cellular components in FFPE samples (Fig. 2k–p). High-dimensional reduction technology was used to identify cell types from scFFPE-ATAC data (Fig. 2k). Same to fresh samples, T cells, B cells, and myeloid cells were

identified in both FFPE and fresh mouse spleen. The proportional distribution of these three cell types was similar between FFPE and fresh samples (Fig. 2l).

Additionally, gene activity analysis confirmed the presence of unique gene markers for each cell type (Fig. 2m, n, Supplementary Fig. 7c, 7d, Supplementary Table 3, Table 4), such as *Cd3e* for T cells, *Cd19* for B cells, and *Tppp3* and *Vcam1* for myeloid cells. We also calculated the minimum number of cells required to recapture cell components in scFFPE-ATAC from FFPE samples by comparing with fresh samples through random downsampling, using parameters such as cluster numbers and specific cell-type gene activity assessment (Supplementary Fig. 8). Clearly, 300 cells from both fresh and FFPE samples are sufficient to capture the cell components in mouse spleen. This result also rules out the possibility that conventional scATAC-Seq fails to resolve cell-type-specific epigenetic profiles in FFPE samples due to insufficient cell capture, as over 500 cells were obtained (Fig. 1j).

Furthermore, a comparable number of accessible chromatin peaks were identified for each cell type (Fig. 2o, p; Supplementary Tables 5, Table 6). Importantly, transcription factor motif enrichment analysis of these unique peaks for each cell type revealed similar transcription factors in both FFPE and fresh mouse spleen samples (Supplementary Fig. 9). We noticed a batch effect between scFFPE-ATAC data from FFPE samples and scATAC-Seq data from fresh samples. However, both datasets exhibited strong gene activity markers, identified accessible chromatin peaks for each cell type (Supplementary Fig. 10a, and Supplementary Table 7), unique accessible peaks (Supplementary Fig. 10b, Supplementary Table 8), and transcription factors (TFs) enriched at these accessible chromatin sites for each cell type (Supplementary Fig. 10c). In the B cell group, we found that key TFs involved in B cell development, such as EBF1, TCF3, POU family genes, and others, were enriched in both FFPE and fresh samples (Supplementary Fig. 10c). In the T cell group, TFs involved in T cell differentiation and development, including TCF7L2, LEF1, ETS family (ETV1-5, ETS1, etc.), RUNX2, RUNX3, TBX family, and SPDEF, were specifically enriched in T cells from both FFPE and fresh samples (Supplementary Fig. 10c). For myeloid cells, TFs such as FOS, MAF family, and GATA1 were enriched in both FFPE and fresh samples (Supplementary Fig. 10c). Taken together, we confirm that our scFFPE-ATAC allows us to decipher single-cell chromatin accessibility in FFPE samples at single-cell resolution.

## scFFPE-ATAC decodes single-cell chromatin accessibility in clinically archived FFPE human lymph node tissue stored for 8–12 years

Next, we purified nuclei from human FFPE lymph nodes using density gradient centrifugation (Supplementary Fig. 11a) and applied scFFPE-

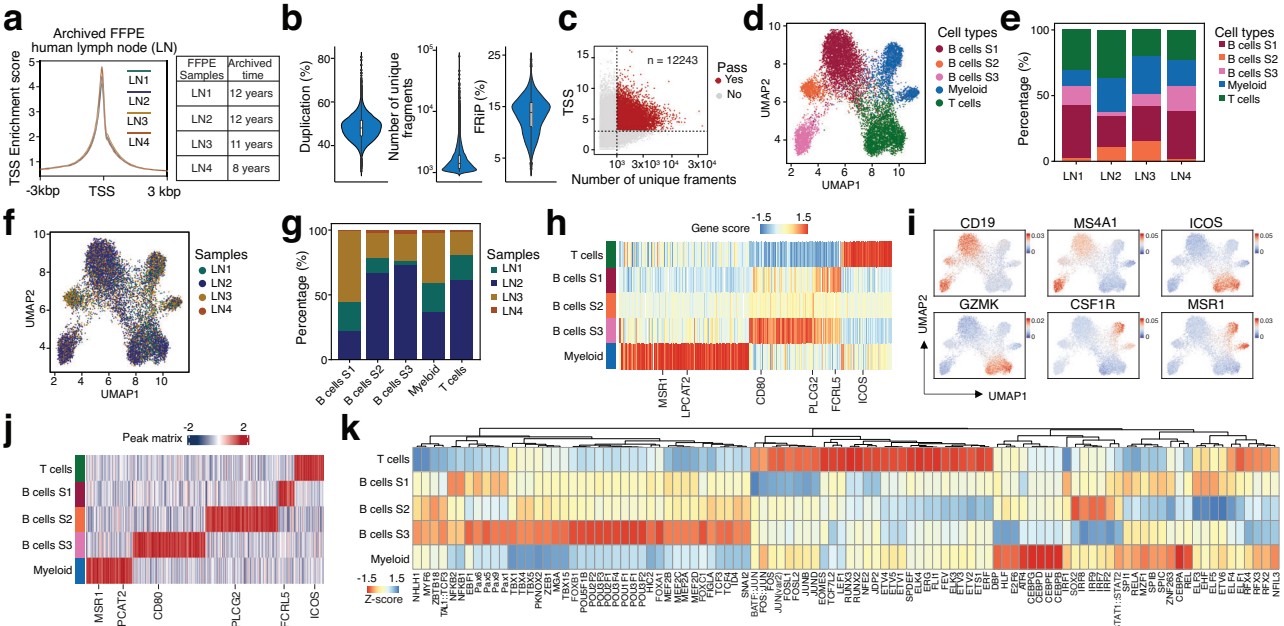

**Fig. 3 | scFFPE-ATAC decodes single-cell chromatin accessibility in clinically archived FFPE human lymph node tissue. a** Sequencing signal enrichment at transcription start site (TSS) from scFFPE-ATAC in the FFPE human lymph node. **b** Duplication rate (Left), number of unique DNA fragments (Middle), and fraction of reads in peaks (FRiP) (Right) from scFFPE-ATAC in the human lymph node. Box boundaries represent the interquartile range (IQR), spanning from the first quartile (Q1) to the third quartile (Q3). Duplication rate − $n$ = 17,143, max = 80.74, min = 28.77, Q1 = 45.31, median = 48.77, Q3 = 52.09. Unique fragments − $n$ = 17,143, max = 35,745, min = 1000, Q1 = 1133, median = 1344, Q3 = 1777. FRiP − $n$ = 17,143, max =

24.21, min = 3.99, Q1 = 11.62, median = 14.05, Q3 = 15.88. **c** Parameters for selecting high-quality single cells from scFFPE-ATAC. **d** Cell type identification using high-dimensional reduction technique. **e** Distribution of each cell type across different samples. **f** Occupancy of each sample in each cell type. **g** Proportional distribution of each sample within each cell type. **h** Active genes predicted from scFFPE-ATAC for each cell type. **i** Gene activity scores of example genes for each cell type. **j** Unique accessible chromatin peaks for each cell type. **k** Enriched transcription factors for each cell type. Source data are provided as a Source Data file for Fig. 3a-k.

ATAC to four clinically archived human lymph node (LN) FFPE samples −benign lymph nodes preserved for 8–12 years (Fig. 3). The sequencing signal was clearly enriched at the TSS site in these samples (TSS enrichment score ≥4) (Fig. 3a). For each cell, the median fraction of reads in peaks was 13.87% (Fig. 3b), the median number of unique DNA fragments for FFPE samples was 1356 (Fig. 3b), and the duplication rate was 48.5% (Fig. 3b). In total, we obtained 12,243 high-quality cells from four samples after removing doublet nuclei (Fig. 3c, Supplementary Fig. 11b): 1,883 cells for LN1, 7,035 cells for LN2, 3,116 cells for LN3, and 209 cells for LN4. The fragment distribution of the scFFPE-ATAC library from those long-term archived samples was dominated by short fragments (96.28% in the range of 0–100 bp) (Supplementary Fig. 11c).

High-dimensional reduction allowed us to identify five cell types in the archived lymph nodes based on chromatin accessibility, including myeloid cells, T cells, and three subtypes of B cells (B cells S1, B cells S2, and B cells S3) (Fig. 3d). All four LN samples contained all five cell types (Fig. 3e), and the distribution of these five cell types across the four LNs was heterogeneous (Fig. 3e–g, and Supplementary Fig. 11d). Gene activity prediction from chromatin accessibility clearly showed that there are cell type-specific active genes (Fig. 3h, i, andSupplementary Fig. 12a, and Supplementary Table 9). For example, the B cell-specific gene *CD19* was active in all three subtypes of B cells, T cell-specific genes such as *ICOS* and *GZMK* were active in T cells, and *CSF1R* and *MSR1* were active in myeloid cells. Differentially accessible chromatin peak analysis helped us identify unique accessible chromatin peaks for each cell type: 2234 peaks for T cells, 3407 peaks for myeloid cells, 1203 peaks for B cell S1, 5406 peaks for B cell S2, and 5389 peaks for B cell S3 (Fig. 3j, and Supplementary Table 10). These peaks were distributed across different parts of the genome (Supplementary Fig. 12b). Gene pathway enrichment on these unique chromatin peaks revealed cell type- and biologically relevant accessible

chromatin sites (Supplementary Fig. 12c, Supplementary Table 11). Importantly, TF enrichment analysis of the unique accessible chromatin peaks identified cell type- and lineage-specific TFs for each cell type (Fig. 3k). For example, BATF::JUN, FOS::JUN, EOMES, TCF7L2, LEF1, RUNX3, and RUNX2 are strongly enriched in T cells; CEBP family TFs (CEBPG, CEBPD, CEBPE, CEBPB) and ATF4, among others, strongly associated with myeloid cell differentiation and function, are highly enriched in myeloid cells; EBF1 and TCF3, which are central to B cell development during the early stages of lineage commitment and differentiation, are enriched in all three B cell populations, with stronger enrichment in B cell S3. At the same time, we found stronger enrichment of other TFs, such as SOX21, IRF2, and IRF7-9, for subsets of B cell S1 and S2, but not S3.

Taken together, our data strongly demonstrate that scFFPE-ATAC can resolve cell composition and single-cell epigenetic regulation in long-term archived clinical FFPE samples.

## scFFPE-ATAC uncovers spatially distinct epigenetic regulators driving tumor progression from the tumor center to the invasive edge in FFPE human lung cancer

Solid tumors have unique tumor physiology compared to normal tissue, and deciphering the tumor microenvironment in different parts of solid tumors could help us better understand the molecular mechanisms of tumorigenesis, progression, and relapse[45,46]. To better understand tumor heterogeneity and epigenetic regulation in different parts of solid tumors, we marked distinct regions on the tumor block based on the pathological analysis of hematoxylin and eosin (H&E) staining derived from the same tumor block. To do this, we collected samples from two marked regions−the tumor center (TC) and invasive edge (IE)−of a human FFPE lung cancer tissue block using a 1 mm puncher (Fig. 4a), to validate the workability and robustness of our scFFPE-

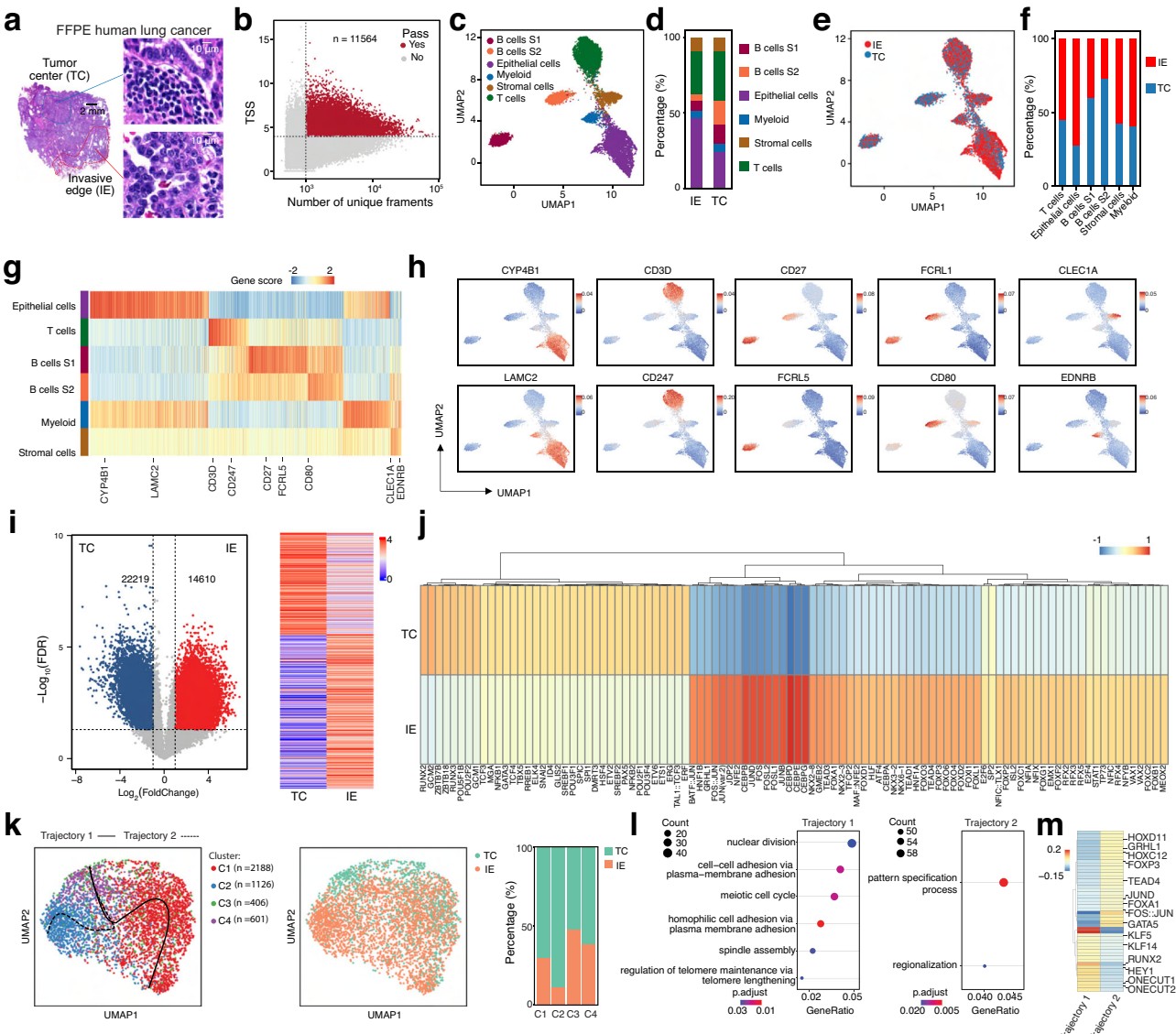

**Fig. 4 | scFFPE-ATAC reveals spatially distinct epigenetic regulators in the tumor center and invasive edge of FFPE human lung cancer at single-cell resolution. a** Hematoxylin and eosin (H&E) staining of human lung cancer tissue, with the tumor center (TC) and invasive edge (IE) indicated by colored circles. Left: Whole H&E-stained section. The blue circle marks the tumor center, and the red circles indicate the invasive edge of the tumor. Right: Zoomed-in regions of the areas pointed to by the colored lines (blue = TC; red = IE) in the left panel, showing the tumor center and invasive edge. Three adjacent sections were cut and stained, showing consistent results. **b** Parameters for selecting high-quality cells from scFFPE-ATAC. TSS = transcription start site. **c** Cell type identification using a high-dimensional reduction technique. **d** Proportional distribution of different cell types in TC and IE. **e** Distribution of TC and IE across different clusters. **f** Proportional distribution of TC and IE within different cell types. **g** Predicted active genes from scFFPE-ATAC for each cell type. **h** Representative active genes from scFFPE-ATAC

for each cell type. **i** Identification of differentially accessible chromatin peaks between TC and IE. Left: Volcano plot showing the cutoff (False Discovery Rate (FDR) ≤ 0.05, |Log$_2$(Fold change)| ≥ 1). Right: Heatmap displaying the identified peaks. A one-sided Fisher's exact test was applied in the significance test. The FDR was corrected using the Benjamini–Hochberg (BH) method. **j** Enriched transcription factors on differentially accessible chromatin peaks between TC and IE. **k** Pseudotime trajectory analysis of epithelial cells from the tumor center to the invasive edge. Left: Two identified trajectory paths and four cell clusters along the trajectories. Middle: Distribution of epithelial cells from TC and IE across the two trajectories. Right: Distribution of epithelial cells from TC and IE across the four clusters. **l** Specific gene pathways identified from the two trajectories. **m** Specific transcription factors identified in the two trajectories. Source data are provided as a Source Data file for Fig. 4b-m.

ATAC technique in solid tumors. We isolated single-cell nuclei (Supplementary Fig. 13a) from both the TC and the IE using the established protocol and performed scFFPE-ATAC. We performed quality control to assess data quality (Supplementary Fig. 13b-d). The TSS enrichment score, violin plots showing duplication rate, FRiP, and the number of unique fragments present in both the tumor center and invasive edge were comparable. Similar to the mouse spleen and human lymph node FFPE samples, the fragment distribution of the scFFPE-ATAC library from human FFPE lung samples was, as expected, primarily composed

of short fragments (99.33% in the range of 0–200 bp) (Supplementary Fig. 13e). In total, we captured 11,564 cells with high-quality chromatin accessibility (Fig. 4b), with 6731 cells from IE and 4833 cells from TC.

High-dimensional reduction of chromatin accessibility from these 11,564 cells helped us identify six components: Epithelial cells ($n$ = 4349), T cells ($n$ = 3757), Myeloid cells ($n$ = 636), Stromal cells ($n$ = 1098), B cells S1 ($n$ = 1317), and B cells S2 ($n$ = 1229). All of these components were found in both the IE and TC, though the distribution of different cell types varied between IE and TC, indicating differences

in the tumor microenvironment between these regions (Fig. 4d–f, and Supplementary Fig. 13f). Furthermore, gene activity prediction from scFFPE-ATAC for each cell type clearly showed that cell type-specific genes were active in each population (Fig. 4g, h, and Supplementary Fig. 14, Supplementary Table 12). For example, *CD3D* and *CD247* were actively expressed in T cells; *CYP4B1* and *LAMC2* (lung-specific epithelial gene marker) were clearly more expressed in epithelial cells; *CD27*, *FCRL1*, and *FCRL5* were highly active in B cells S1 and S2, with *CD80* being highly active in B cells S2, suggesting that B cells S1 may represent more mature B cells. We also identified specific accessible chromatin peak regions for each cell type (Supplementary Fig. 15a, Supplementary Table 13). Specifically, there were 11,128 accessible peaks for epithelial cells, 12,912 peaks for stromal cells, 24,709 peaks for myeloid cells, 3528 peaks for T cells, 11,008 peaks for B cells S1, and 11,452 peaks for B cells S2. Further transcription factor (TF) enrichment analysis of these unique peaks helped us identify TFs uniquely enriched on these peaks for each cell type (Supplementary Fig. 15b). For example, ETV family TFs, ETS1, and others were strongly enriched in T cells; POU and IRF family TFs were strongly enriched in B cells S1; NFKB1, NFKB2, and PAX5 were strongly enriched in B cells S2; FOX family TFs and NKX family TFs were strongly enriched in epithelial cells. Stromal and myeloid cells showed similar TF enrichment, including TEAD family TFs, JUN, and FOS. However, these TFs were also enriched in epithelial cells, while NFATC1 was much more strongly enriched in myeloid cells.

We also compared the chromatin accessibility of epithelial cells between the TC and IE and identified specific peaks: 22,219 peaks for TC and 14,610 accessible peaks for IE ( $|\text{Log}_2(\text{Fold change})| \geq 1$, False Discovery Rate (FDR) $\leq 0.05$) (Fig. 4i, and Supplementary Table 14). These differential peaks are located in different parts of the genome but show a similar distribution between TC and IE (Supplementary Fig. 16a). Gene pathway enrichment uncovered different enriched pathways (Supplementary Fig. 16b, and Supplementary Table 15). Notably, pathways such as Wnt signaling, cell growth, mesenchymal cell differentiation, and Ras protein signal transduction were specifically enriched in the IE, indicating stronger invasive and migratory features in the IE compared to TC. The IE of a tumor usually exhibits stronger features of migration, invasion, and tissue remodeling, as epithelial cells adopt mesenchymal-like traits to invade surrounding tissues. This process is achieved through the activation of pathways related to actin cytoskeleton organization, cell junctions, and Wnt signaling. Tumor cells at the edge undergo processes similar to embryonic development or wound healing to enable invasion and metastasis[45,46]. Our enriched signal pathways in the IE reflect this hypothesis and further indicate that our scFFPE-ATAC technique effectively captures chromatin accessibility. On the other hand, pathways related to cell-cell adhesion and the regulation of leukocyte cell-cell interactions were observed in TC, reflecting the different tumor microenvironments between TC and IE within the same tumor. Tumor cells in TC may modulate immune cell interactions to evade immune surveillance and maintain tumor growth.

Furthermore, the TF enrichment analysis of these differential accessible chromatin peaks showed that different TFs are enriched in the TC and IE (Fig. 4j). Specifically, in the tumor center, TFs such as POU5F1B (Oct4) and RUNX2 are likely maintaining stem cell-like properties and driving tumor cell proliferation. TFs like PAX5 and SPI1 might regulate the immune cell composition in the tumor, including myeloid cell recruitment. In the invasive edge, TFs such as FOS, JUN, and FOXC2 are likely driving epithelial-to-mesenchymal transition (EMT), enabling migration, invasion, and metastasis of tumor cells. TFs like STAT3 and FOXA1 could be involved in immune evasion and cell motility.

Epithelial cells may migrate from the tumor center toward the invasive edge to facilitate tumor progression and invasion. To understand how epithelial regulation governs this process, we performed pseudotime trajectory analysis to reconstruct the dynamic progression of epithelial tumor cells from the tumor center to the invasive edge using scFFPE-ATAC data from both tumor regions (Fig. 4k). This analysis revealed two distinct epigenetic trajectories (Fig. 4k), suggesting alternative regulatory programs underlying spatial tumor evolution. Four clusters were identified along the trajectory; clusters 3 and 4—comprising over 50% of the cells—were predominantly derived from the tumor center and formed the root of the bifurcating trajectory (Fig. 4k). Clusters 1 and 2 corresponded to the two divergent paths: trajectory 1 and trajectory 2, respectively. Bootstrap resampling confirmed the robustness of the inferred two trajectories, with trajectory 1 and trajectory 2 consistently reproduced in 97.5 and 93.5% of bootstraps, respectively (Supplementary Fig. 17a, and 17b). Furthermore, pseudotime analysis across 1000 bootstrap replicates (with 95% confidence intervals) reproduced the same trajectory path (Supplementary Fig. 17c). We identified 1,241 peaks specific to cluster 1 and 4,399 unique accessible peaks specific to cluster 2 (Supplementary Table 16). Gene ontology analysis of these differential peaks showed that trajectory 1 (cluster 1) was enriched in cell division–related pathways, including nuclear division, spindle assembly, and meiotic cell cycle, indicating a highly proliferative cell state (Fig. 4l). In contrast, trajectory 2 (cluster 2) was enriched in pathways related to regionalization, suggesting a role in spatial patterning and possibly cell migration (Fig. 4l). Transcription factor enrichment analysis further supported these findings (Fig. 4m, and Supplementary Table 17): Lineage 1 showed increased enrichment of transcription factors associated with epithelial differentiation and identity—such as FOXA1, GRHL1, HOX, and FOXP3—indicating a more differentiated, proliferative, and less migratory cell state. In contrast, Lineage 2, exhibited increased enrichment of transcription factors including KLF5, RUNX2, HEY1, and ONECUT2, which are associated with dedifferentiation, stem-like properties, and invasiveness. Interestingly, the transcription factors enriched in Lineage 2, including KLF5, RUNX2, HEY1, and ONECUT2, have been linked to hypoxia-responsive pathways[47–50]. Hypoxia-inducible factor (HIF) signaling can directly or indirectly regulate KLF5, promoting stem-like traits and survival under low oxygen[47]. RUNX2 is induced by hypoxic stress and drives invasive behavior in solid tumors[48]. HEY1 is a canonical Notch target, and Notch–HIF1A crosstalk is well documented in hypoxic tumor microenvironments[49]. ONECUT2 has recently emerged as a master regulator of aggressive, hypoxia-associated tumor phenotypes[50]. These results suggest that there are potentially two distinct trajectory paths as epithelial tumor cells undergo epigenetic reprogramming during spatial progression from the tumor center to the tumor edge, possibly driven by differences in the tumor microenvironment—such as hypoxia—and contributing to variations in invasiveness and metastatic potential at the invasive front. These results further demonstrate that our scFFPE-ATAC method effectively captures single-cell chromatin accessibility from different regions of solid tumors and provides a unique perspective on the epigenetic regulation of tumor progression within the tumor microenvironment. Such spatially oriented single-cell accessibility assays from solid tumor FFPE samples also pave the way for spatial chromatin accessibility assays for FFPE samples in the future.

## scFFPE-ATAC identifies key epigenetic drivers of tumor relapse from paired primary and relapsed tumor FFPE samples

Tumor relapse is the biggest challenge in cancer therapy, with more than 90% of cancer patients dying due to tumor relapse[51–53]. When comparing paired primary and relapse tumors, no specific genetic mutations have been identified as drivers of tumor relapse[19,20]. However, tumor microenvironment, and epigenetic regulation play a crucial role in controlling tumor relapse[20,54–57]. Therefore, deciphering epigenetic regulation by comparing epigenetic profiles between paired primary and relapse samples from the same patient could help

us understand the basic molecular mechanisms and identify key epigenetic regulators driving tumor relapse[58]. This, in turn, could lead to the identification of targets for therapeutic design in clinical settings. Since it is unpredictable when or if tumor relapse will occur, these paired samples are typically archived in FFPE format. Therefore, it is crucial to apply single-cell chromatin accessibility techniques to FFPE samples to decipher chromatin accessibility in paired primary and relapse samples from the same patient. Follicular lymphoma (FL) is a commonly diagnosed form of non-Hodgkin lymphoma, predominantly involving B-cells within the lymphatic system[59]. Classified as an indolent lymphoma, FL typically exhibits slow progression, allowing for extended periods of remission following treatment[60]. Despite this, relapse remains a significant clinical challenge, as the disease can return after remission, complicating long-term management and therapeutic intervention[60]. When FL relapses, there are two primary outcomes that can complicate treatment (Fig. 5a). In some cases, the disease remains as follicular lymphoma, manifesting as recurrent FL. However, in a subset of patients, the disease undergoes transformation into a more aggressive form, typically diffuse large B-cell lymphoma (DLBCL)[61]. The transformation from FL to DLBCL is particularly concerning, as it represents a substantial shift in the tumor's behavior, characterized by increased aggression and resistance to treatment, ultimately leading to a poorer prognosis. Understanding the underlying molecular and epigenetic mechanisms that drive either relapse or transformation is crucial for improving patient outcomes.

Next, we purified nuclei from two pairs of primary and relapsed FFPE tumor samples from FL patients (Supplementary Fig. 18a, and 18b) and applied scFFPE-ATAC to investigate chromatin accessibility changes underlying relapse and transformation in FL (Fig. 5a). The first pair represents a more complex case, where the disease transformed from primary FL to DLBCL after a seven-year interval. The second pair consist of primary FL and its corresponding relapsed FL tumor, with a two-year gap between the primary diagnosis and relapse. By comparing epigenetic regulation in primary versus relapsed or transformed tumors, we aim to identify key regulatory mechanisms contributing to disease recurrence or progression. This analysis will provide valuable insights into the epigenetic alterations driving the transition from indolent FL to either recurrent FL or aggressive DLBCL, potentially revealing biomarkers and therapeutic targets for managing relapse and preventing transformation in FL patients. Using scFFPE-ATAC, we captured 13,357 single cells with high-quality chromatin accessibility from these two patient pairs (Fig. 5b, and Supplementary Fig. 18c, 18d). In these FFPE archived samples, 97.72% of detected fragments in scFFPE-ATAC-seq were in the range of 0–100 bp (Supplementary Fig. 18e). The merged scFFPE-ATAC signal demonstrated strong TSS enrichment (Supplementary Fig. 19a), even after 6 to 13 years of storage in the FFPE format. High-dimensional reduction analysis of these lymphoma cells, along with LN samples, allowed us to identify two tumor B cell groups (Tumor B1 and Tumor B2), as well as Myeloid cells, B cells, and T cells (Fig. 5c). The distribution of these cell types varied between the two patient pairs, highlighting their heterogeneity (Fig. 5d–f).

Notably, in patient 1, who experienced a transition from FL to DLBCL after a 7-year interval, the proportion of Tumor B1 decreased from 85.22% to 12.75%, while Tumor B2 increased from 0.6% to 8.53%, suggesting that Tumor B2 may play a role in the transformation from FL to DLBCL. In patient 2, who remained as FL over a 2-year period, the proportion of Tumor B1 increased from 2.89% to 74.56%, while Tumor B2 decreased from 32.58% to 11.50%, indicating that Tumor B1 could contribute to FL relapse. The accuracy of cell type identification using scFFPE-ATAC was further validated through specific gene activity prediction (Fig. 5 g, 5 h, Supplementary Table 18). For example, *CD3D* is active in T cells, while *CD163* is active in myeloid cells. The *MS4A1* gene, which encodes the *CD20* protein, is active in both normal B cells and

tumor B1 and B2 cells, with higher expression in Tumor B2 compared to normal B cells and Tumor B1. Notably, *CD20* is a major target for monoclonal antibody therapies, such as rituximab, which is widely used in the treatment of B-cell malignancies like follicular lymphoma (FL) and diffuse large B-cell lymphoma (DLBCL). Oncogenes show specific activity in tumor cells (Fig. 5g, 5h), with *LMO2*, *LYN*, *TNFRSF17*, *CARD11*, and *BCL7A* being active in Tumor B1, while *BCL2* and *WAS* are specifically active in Tumor B2.

Specific chromatin-accessible peaks were identified for each cell type (Supplementary Fig. 19b, and Supplementary Table 19), including 2349 peaks for T cells, 710 for B cells, 4407 for myeloid cells, 2405 for Tumor B1, and 9446 for Tumor B2. Additionally, distinct TFs were enriched in these cell type-specific peaks (Supplementary Fig. 19c). B-cell lineage and differentiation TFs, such as POU family members (POU2F2, POU3F4), PAX family members (PAX1, PAX5, PAX9), and TCF family members (TCF3, TCF4), were enriched in normal B cells, Tumor B1, and Tumor B2. Epithelial-mesenchymal transition (EMT) and chromatin remodeling TFs, including ZEB1, SNAI2, and ID4, were more strongly enriched in Tumor B1, whereas HIC2, RHOXF1, and CFCF showed higher enrichment in Tumor B2. Such unique TF enrichment sets help us uncover the epigenetic drivers of tumor relapse in different patients. In patient 1 (FL transformation to DLBCL), the primary tumor exhibited minimal presence of Tumor B2 cells, whereas in the relapse sample, Tumor B2 accounted for 8.53%. This suggests that TFs enriched in Tumor B2, such as HIC2 and RHOXF1, may contribute to the transition from FL to DLBCL. In patient 2 (primary FL to relapse FL after a 2-year gap), there was a significant increase in Tumor B1 cells (from 2.89% to 74.56%), indicating that TFs enriched in Tumor B1, such as ZEB1, SNAI2, and ID4, may drive FL relapse. Although FL does not undergo classical epithelial–mesenchymal transition (EMT), transcription factors such as ZEB1 and SNAI2 (Slug) are well-established EMT regulators in solid tumors, where they drive plasticity, stemness, and therapy resistance[62–65]. These functions are highly relevant to tumor relapse, as similar transcriptional programs may be co-opted by malignant B cells to survive therapeutic pressure. ID4 is frequently silenced by promoter hypermethylation in FL[66], suggesting a tumor-suppressive role whose loss could facilitate disease progression. Taken together, enrichment of ID4, ZEB1, and SNAI2 in FL subclones highlights transcriptional circuits that may enable stress tolerance and clonal evolution, warranting deeper investigation in the context of FL relapse. Further in-depth studies will be required to clarify their mechanistic contributions and potential as therapeutic targets in the future.

Next, we decipher the epigenetic regulation of tumor origin and evolution during relapse by focusing on tumor cells from patient-specific primary and relapse tumors. We extracted normal B cells and tumor B cells from scFFPE-ATAC data and performed pseudotime trajectory analysis to reconstruct the dynamic trajectory of FL relapse and DLBCL transformation (Fig. 5i). Clearly, two trajectory pathways were identified from normal B cells to tumor B cells (Fig. 5i), with Tumor B1 forming trajectory 1 and Tumor B2 forming trajectory 2. Four clusters were identified along the trajectories (Supplementary Table 20). Bootstrap resampling confirmed the robustness of the two inferred trajectories, with trajectory 1 and trajectory 2 consistently reproduced in 83% of bootstraps each (Supplementary Figs. 20a, b). Furthermore, pseudotime analysis across 1000 bootstrap replicates (with 95% confidence intervals) reproduced the same trajectory path (Supplementary Fig. 20c). Cluster 3 (C3) mainly consists of normal B cells and forms the root of the trajectory. Cluster 1 (C1) represents an intermediate stage between normal B cells and Tumor B2. Cluster 2 (C2) corresponds to Tumor B1, and Cluster 4 (C4) is the main contributor to Tumor B2. We focused on C2 and C4, which are the endpoints of trajectories 1 and 2, respectively (Fig. 5i). The majority of patient 1's primary tumor (P1P FL) cells are located in C2 (Tumor B1), whereas the majority of patient 2's primary tumor (P2P FL) cells

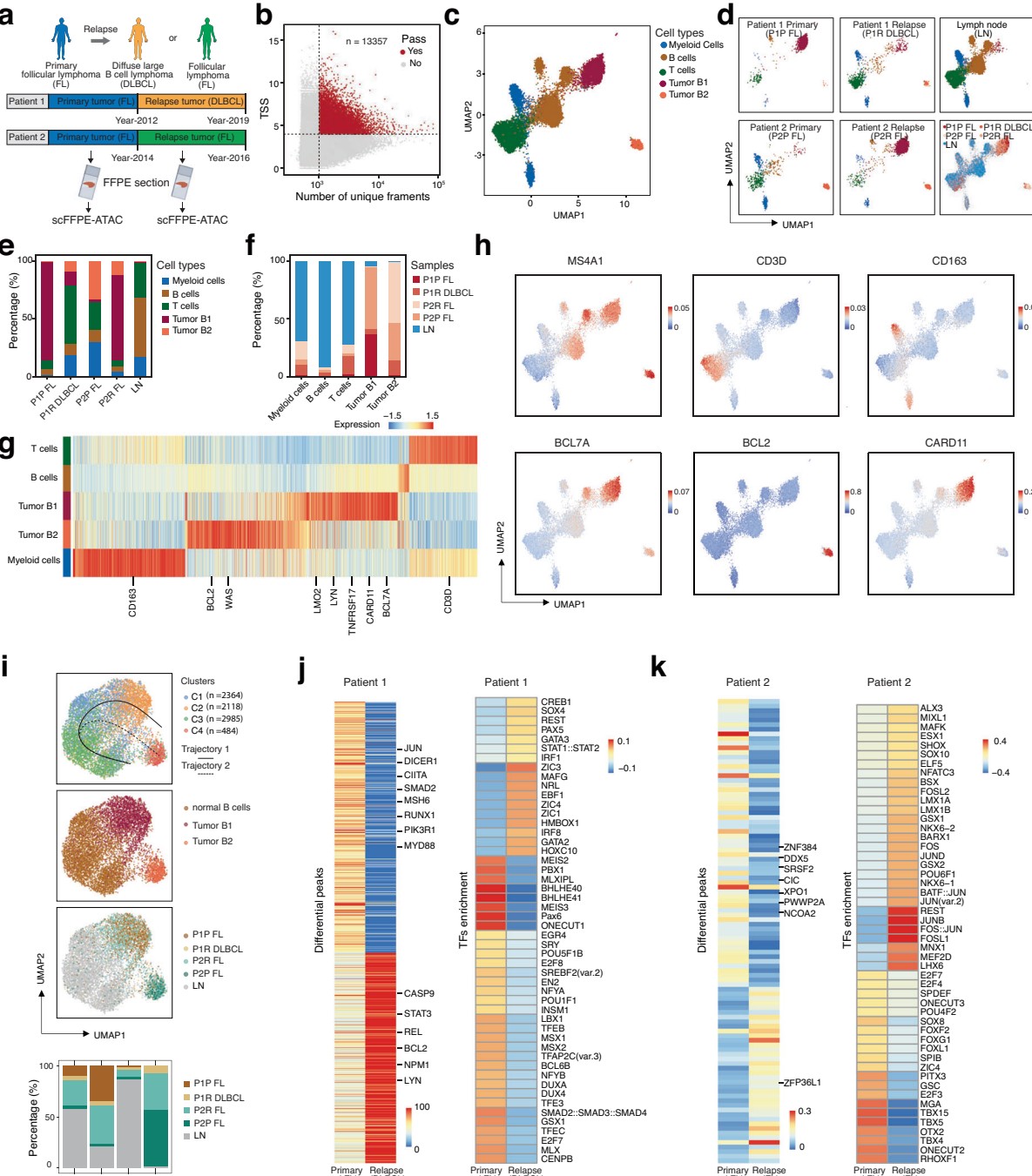

**Fig. 5 | scFFPE-ATAC deciphers epigenetic regulation underlying tumor relapse in paired primary and relapsed follicular lymphoma FFPE samples. a** Schematic showing two pairs of patients studied. One patient progressed from follicular lymphoma (FL) to diffuse large B-cell lymphoma (DLBCL), and the other patient relapsed from FL to FL. **b** Parameters for selecting high-quality single-cell ATAC-seq data from scFFPE-ATAC. TSS = transcription start site. **c** Five cell types identified using high-dimensional reduction technique. **d** Cell type distribution for each patient. **e** Proportional distribution of each cell type across different patients. **f** Proportional distribution of cells from different patients within each cell type. **g** Predicted active genes from scFFPE-ATAC for each cell type. **h** Representative active genes from scFFPE-ATAC for each cell type. **i** Pseudotime trajectory analysis of tumor B cells transitioning from normal B cells to tumor. The top three panels

display UMAP plots: the top panel shows two inferred trajectory paths and four cell clusters (C1–C4); the middle panel illustrates the distribution of normal B cells and two tumor B cell types along the trajectories; the third panel presents tumor B cells from each paired patient mapped onto the trajectories. The bottom panel shows bar plots depicting the distribution of cells from each patient across the four clusters (C1–C4) along the two trajectories. **j** Differentially accessible chromatin peaks (left panel) and enriched transcription factors (TFs) (right panel) identified from the primary tumor (FL) to transformed DLBCL in tumor B cell type B1 from patient 1. **k** Differentially accessible chromatin peaks (left panel) and enriched transcription factors (TFs) (right panel) identified from primary FL to relapsed FL in tumor B cell type B2 from patient 2. Source data are provided as a Source Data file for Fig. 5b-k.

dominate in C4 (Tumor B2) (Fig. 5i). The distinct trajectory pathways for Tumor B1 and Tumor B2 strongly indicate that the tumor origins of patient 1 and patient 2 differ. Although both patients had the same tumor type at primary diagnosis, the epigenetic profiles of their original tumor clones may contribute to different relapse outcomes: patient 1 experienced transformation to DLBCL, while patient 2 had relapse of FL. These results suggest distinct epigenetic evolutionary pathways during tumor relapse in these patients. Therefore, we focused on comparing the epigenetic profiles of Tumor B1 in patient 1 and Tumor B2 in patient 2.

In the patient 1, differential chromatin accessibility analysis identified 632 peaks specific to Tumor B1 in the primary tumor and 535 peaks specific to Tumor B1 in the relapsed tumor. These differentially accessible chromatin peaks are primarily located in promoter regions (Supplementary Fig. 21a, and Supplementary Table 21). In the primary FL tumor, we identified oncogenes with higher chromatin accessibility, including *JUN*, *DICER1*, *CIITA*, *SAMD2*, *MSH6*, *RUNX1*, *PIK3R1*, and *MYD88*. In contrast, the relapsed DLBCL tumor exhibited higher chromatin accessibility in genes such as *CASP9*, *STAT3*, *REL*, *BCL2*, *NPM1*, and *LYN* (Fig. 5j). Furthermore, gene pathway enrichment analysis of the differentially accessible chromatin peaks between primary FL and relapsed/transformed DLBCL revealed distinct pathway alterations (Supplementary Fig. 21b, Supplementary Table 22). In the primary FL samples, enriched terms such as chromosomal regions, centromeric regions, ribosome, and cytosolic large ribosomal subunit suggest a relatively stable chromosomal architecture and balanced protein synthesis, consistent with the indolent nature of FL. In contrast, relapsed DLBCL samples show enrichment for mitotic structures (midbody, kinetochore, spindle, condensed chromosome), mitochondrial metabolism (mitochondrial matrix, mitochondrial ribosome, mitochondrial inner membrane), and transcriptional regulation (RNA polymerase II transcription regulator complex, methyltransferase complex). These findings indicate increased proliferation, chromosomal instability, metabolic reprogramming, and transcriptional deregulation—hallmarks of aggressive lymphoma transformation. Overall, this shift reflects the transition from the slow-growing, genetically stable FL to the highly proliferative and metabolically active DLBCL, highlighting key molecular changes driving disease progression and relapse. TF enrichment from these differential peaks clearly shows that unique TFs are enriched at these accessible chromatin sites, with specific TF profiles for the primary tumor and relapse tumor (Fig. 5j). In primary FL tumors, a distinct set of TFs is enriched, including MEIS2, PBX1, POU5F1B, E2F8, TFEB, BCL6B, and SMAD2::SMAD3::SMAD4, among others. These TFs are involved in key processes such as B-cell differentiation, cell cycle regulation, and metabolic control. For example, MEIS2[67] and PBX1[68] are crucial for hematopoietic stem cell regulation and early B-cell development, contributing to the indolent nature of FL. The presence of BCL6B, a regulator of B-cell differentiation[69], suggests that FL remains in a less differentiated state compared to more aggressive lymphomas. TFEB, which is involved in autophagy and stress response, suggests FL cells' survival mechanisms in the tumor microenvironment, while SMAD2::SMAD3::SMAD4 points to the involvement of transforming growth factor-beta (TGF-β) signaling, which can promote immune evasion and contribute to FL pathogenesis. In contrast, in transformed DLBCL tumors, the TFs profile undergoes a distinct shift. Key TFs such as CREB1, SOX4, PAX5, GATA3, STAT1::STAT2, EBF1, IRF1, and ZIC3 dominate in DLBCL. These TFs are closely associated with the aggressive and rapidly proliferating nature of DLBCL. PAX5 and EBF1, critical players in B-cell development, are still present but likely operate in a context of loss of differentiation, promoting the aggressive transformation from FL to DLBCL. SOX4, a TF linked to stemness, cell survival, and differentiation, and STAT1::STAT2, involved in immune signaling, reflect the increased inflammatory and immune evasion activities typical of DLBCL. CREB1, known to regulate cell growth and survival, may help DLBCL cells resist

apoptosis and proliferate rapidly, contributing to relapse after initial treatment. Additionally, the inclusion of ZIC3, IRF1, and GATA3 suggests heightened immune responses and that immune escape mechanisms may play a crucial role in DLBCL relapse. The comparison of TF profiles between primary FL and relapsed DLBCL underscores the shift from a relatively indolent and differentiated state in FL to a more proliferative and aggressive phenotype in DLBCL. In FL, TFs support B-cell differentiation, survival, and immune evasion, while in DLBCL, TFs shift towards promoting cell proliferation, immune modulation, and stress response mechanisms, ultimately driving the aggressive nature of the disease and its relapse. These findings highlight the crucial role of transcription factors in the progression from FL to DLBCL and provide potential targets for therapeutic intervention to prevent relapse or transformation.

In patient 2 (from primary FL to relapse FL with a 2-year gap), we focused on comparing the chromatin accessibility differences in Tumor B1 between the primary tumor (FL) and the paired relapse (FL) and identified 62 peaks for the primary tumor and 38 peaks for the relapse tumor (Fig. 5k, and Supplementary Table 23), with oncogenes such as *ZNF384*, *DDX5*, *SRSF2*, *CIC*, *XPO1*, *PWWP2A*, and *NCOA2* showing higher accessibility in the primary tumor, while *ZFP36L1* was more accessible in the relapse FL. These differential ATAC-Seq peaks were mainly located in promoter regions of the genome (Supplementary Fig. 21c). We did not find any specific gene pathways enriched in the differential gene list. We examined the transcription factor (TF) enrichment between primary FL B2 tumors and relapsed FL B2 tumors to gain insights into the epigenetic changes driving tumor progression and relapse (Fig. 5k). Two distinct TF profiles were identified for primary and relapsed FL, revealing key differences that highlight shifts in tumor biology. The TFs enriched in primary FL B tumors included E2F7, SPDEF, SOX8, FOXF2, PITX3, TBX15, and RHOXF1, many of which are involved in the regulation of the cell cycle, differentiation, and developmental processes. For instance, E2F7 and E2F4 are known to regulate cell cycle progression and apoptosis, potentially contributing to the slower, indolent nature of primary FL tumors. Additionally, SOX8 and FOXF2 are critical for stem cell maintenance and differentiation, suggesting that primary FL cells retain a certain level of undifferentiated characteristics that may facilitate immune evasion. Other TFs, such as PITX3, RHOXF1, and TBX15, are involved in developmental and organogenesis pathways, suggesting a more plastic tumor phenotype that might support survival in diverse microenvironments. This indicates that primary FL tumors may possess characteristics that allow them to adapt and evade differentiation into more aggressive phenotypes. In contrast, relapsed FL B tumors showed a distinct enrichment in TFs such as ALX3, MIXL1, MAFK, SOX10, NFATC3, BSX, and FOSL2. These TFs are associated with processes like immune modulation, stress response, and aggressive tumor behavior. For example, SOX10 is well known for its role in neural crest development and has been implicated in the promotion of tumor metastasis, suggesting a shift toward a more migratory and invasive phenotype during relapse. Similarly, NFATC3 and BSX are involved in immune modulation, indicating that relapsed FL tumors might acquire mechanisms to evade immune detection and resist therapeutic treatments. Moreover, the enrichment of FOSL2, JUN, and BATF::JUN in relapsed FL tumors points to the activation of the AP-1 signaling pathway, which is known to regulate cellular stress responses, inflammation, and survival. These pathways are often upregulated in aggressive cancers, contributing to chemoresistance and immune evasion in relapsed FL. The comparison of transcription factor profiles between primary FL and relapsed FL tumors highlights a shift in tumor biology from a more differentiated and less aggressive phenotype in primary FL to a more invasive, immune-evasive, and proliferative phenotype in relapsed FL. In primary FL, TFs associated with cell cycle control, differentiation, and survival are more prominent, indicating a tumor that retains differentiated properties. In contrast, relapsed FL

tumors are enriched in TFs that promote tumor aggressiveness, plasticity, and immune modulation, suggesting that these tumors acquire mechanisms to resist immune surveillance and treatment. These findings underscore the dynamic nature of FL tumors and suggest that the epigenetic reprogramming driven by these transcriptional shifts contributes to the progression and relapse of the disease.

In summary, our scFFPE-ATAC-seq technology enables single-cell resolution analysis of epigenetic regulation in clinically archived, paired primary and relapsed FFPE samples preserved over extended periods. This approach provides valuable insights into the epigenetic drivers, cellular origins, and evolutionary trajectories underlying tumor relapse and progression in a clinical context.

## Discussion

Tumor heterogeneity remains a major challenge in cancer therapy, contributing to tumor progression, treatment resistance, and relapse[51]. Epigenetics—defined as stable, heritable phenotypic changes without alterations to the DNA sequence[5]—plays a critical role in tumor initiation, progression, and recurrence. Advances in single-cell epigenetic profiling and multimodal single-cell technologies have greatly improved our ability to dissect these mechanisms[9,42,43,70–73]. However, in clinical practice, over 99% of archived patient samples are stored as formalin-fixed paraffin-embedded (FFPE) tissue, particularly for matched primary, relapse, or metastatic cases. Despite the importance of these samples, technical barriers to assessing chromatin accessibility in FFPE specimens have long hindered our capacity to fully understand tumor heterogeneity, relapse, and metastasis. While RNA expression profiling in FFPE tissues is now feasible[32–34], enabling important mechanistic insights, epigenetic profiling in FFPE samples remains limited—mostly confined to bulk-level assays[23,25–27]. Conventional single-cell ATAC-seq (scATAC-seq) approaches applied to FFPE samples suffer from low library complexity and are ineffective at resolving single-cell chromatin accessibility due to severe DNA damage. To address this gap, we developed scFFPE-ATAC, a high-throughput single-cell chromatin accessibility profiling method optimized for FFPE samples. This approach integrates a newly designed FFPE-compatible Tn5 transposase, ultra-high-throughput DNA barcoding (> 56 million cell barcodes per run), T7 promoter-mediated DNA damage rescue, and in vitro transcription. scFFPE-ATAC enables reliable single-cell chromatin accessibility profiling in FFPE tissues. We demonstrate that scFFPE-ATAC can reveal single-cell epigenetic heterogeneity and cellular composition in a wide range of archived FFPE tissue formats—including punch cores and tissue slices—from various tissues such as spleen, lymph nodes, and lung tumors, even after more than a decade of storage.

As a case study, we applied scFFPE-ATAC to long-term archived human lymph node samples and successfully profiled single-cell epigenetic regulation after 8–12 years of FFPE preservation. We further compared single-cell epigenetic profiles between the tumor center and invasive edge in FFPE lung cancer samples, identifying region-specific epigenetic regulators in epithelial cells. Trajectory analysis revealed that epithelial tumor cells undergo spatially driven epigenetic reprogramming during progression from the tumor core to the invasive edge—potentially influenced by tumor microenvironmental differences, which may underlie increased invasiveness and metastatic potential. Tumor relapse is one of the most formidable challenges in oncology, accounting for over 90% of cancer-related deaths[51–53]. Emerging evidence indicates that both the tumor microenvironment and epigenetic regulation play pivotal roles in driving relapse[20,54–57]. By applying scFFPE-ATAC to paired FFPE samples from the same follicular lymphoma (FL) patients—one of whom experienced FL relapse after two years, and another whose disease transformed into diffuse large B-cell lymphoma (DLBCL) after seven years—we identified key epigenetic regulatory elements implicated in both relapse and transformation. Pseudotime trajectory analysis uncovered distinct epigenetic

evolutionary pathways in relapsed and transformed tumor cells, shedding light on lineage origins and the molecular events underlying disease progression.

Our scFFPE-ATAC represents a powerful tool for investigating tumor heterogeneity, relapse, and spatial progression at single-cell resolution in archived FFPE tissues. This technology opens avenues for retrospective clinical studies, facilitating the discovery of biomarkers and therapeutic targets. Looking ahead, scFFPE-ATAC has the potential to drive the development of spatial epigenomics and multi-omics platforms within the FFPE framework, thereby accelerating both fundamental cancer research and the advancement of precision medicine.

A few limitations were observed in our current scFFPE-ATAC approach. The overall recovery rate is approximately 20% after the split-and-pool step, and at least 250,000–500,000 nuclei are required to successfully perform the experiment. While our study demonstrates applicability across lymphoid tissues and lung cancer, broader validation in matrix-rich tumors such as pancreatic ductal adenocarcinoma (PDAC) remains an important next step. In this study, we employed density gradient centrifugation for nuclei isolation, which provided robust performance across the FFPE samples we tested. As a future direction, FACS-based nuclei purification[14,37,38] could serve as a complementary strategy, particularly for challenging samples such as PDAC, where higher purity may be beneficial. The quality of scFFPE-ATAC libraries in our study did not depend on DNA fragmentation patterns in FFPE samples (Supplementary Technical Note 1). At present, we rely on pre-assessment of scFFPE-ATAC quality based on nuclei purity after isolation. Establishing an additional pre-assessment protocol based on DNA quality would be valuable—similar to single-cell RNA-seq in FFPE samples, where DV200 (Distribution Value 200; the percentage of RNA fragments >200 nucleotides)[33,74] is widely used as a quality-control metric prior to library preparation. Fixation time and storage period in clinically archived samples also vary considerably. We successfully tested samples fixed for 16–72 h and stored for 6 months to 13 years, but further optimization of protocols for samples with longer fixation (> 72 h) or storage durations[28] will be an important direction for future development. Finally, the current workflow requires approximately five working days, and streamlining the protocol to reduce turnaround time would greatly enhance its usability.

## Methods

All research complies with the relevant ethical regulations approved by the regional ethical review committee at Uppsala University.

### Patient material, mouse tissue collection, ethics, and consent for publication

The regional ethical research committee at Uppsala University approved the study (Dnr: 2014/020 and 233/2014). Human lymph node, human lung cancer, human follicular lymphoma, and human diffuse large B-cell lymphoma FFPE tissue blocks were prepared at the Department of Clinical Pathology, Uppsala University Hospital, Uppsala, Sweden, according to standard procedures. Briefly, tissue from surgical specimens was fixed in buffered formalin for 24–72 h. The samples were then examined by a pathologist, excised, and placed in plastic cassettes. The human tissue samples used in this study were obtained from the U-CAN biobank[75]. Remaining clinical samples are archived in the U-CAN biobank and are available for research use upon request. *Mus musculus* (house mouse), strain C57BL/6 J, obtained from The Jackson Laboratory (Strain no. 000664). Mice were maintained on a C57BL/6 J genetic background under specific pathogen-free conditions with a 12-h light/dark cycle, at an ambient temperature of $22 \pm 2\,°C$ and relative humidity of $50 \pm 10\%$. Animals had ad libitum access to standard chow and water. Both male and female mice were used for experiments. Eight-week-old C57BL/6 J mice were sacrificed via inhalation euthanasia, and their spleens were collected.

### Fixation, paraffin embedding, and storage of FFPE samples

All specimens were fixed in buffered 4% formalin (Sigma-Aldrich, cat. no. F8775). The fixation times varied from 16 to 72 h. The mouse spleen samples were prepared in our laboratory with a fixation time of 16 h, while the human FFPE samples were obtained from the U-CAN biobank[75]. Although the exact fixation times for the human samples are difficult to retrieve, they followed routine clinical diagnostic protocols and ranged from 16 to 72 h. For paraffin embedding, all samples were processed under standardized conditions at the U-CAN biobank. Briefly, after fixation the tissues were dehydrated and paraffinized in a vacuum infiltration process overnight using a Tissue-Tek VIP 6 AI tissue processor (Sakura Finetek, cat. no. 6040) (1 h in 70% ethanol, 2.5 h in 95% ethanol, 3.5 h in 99.5% ethanol, 4 h in xylene (Histolab, cat. no. 2070), and 5.5 h in paraffin (Histolab, cat. no. 00402-1) at 63 °C). Finally, the paraffin-embedded tissue was oriented in a cassette, liquid paraffin was poured over it, and it was allowed to set, forming the FFPE block. Storage durations ranged from 2 to 13 years for human samples and up to 6 months for mouse samples. Detailed information for each sample is provided in Supplementary Table 1.

### Tumor center and invasive edge identification and micro-dissection in human lung cancer

To isolate specific regions of human lung cancer tissue, adjacent tissue section (at 5 μm thickness) was prepared and subjected to hematoxylin and eosin (H&E) staining. An experienced pathologist evaluated the stained sections to identify the tumor center and the invasive edge. Based on this histological assessment, the precise locations for tissue sampling were determined. Using this information, 1-mm tissue punches were obtained from the corresponding areas of the unstained tissue blocks, targeting both the tumor center and the invasive edge. These tissue punches were subsequently used for nuclear isolation and downstream analyses.

### Single cell nuclei isolation from FFPE tissue sections

Take a 1-mm punch core from the FFPE tissue block derived from the mouse spleen or human lung tumor block, or cut tissue sections from the human lymph node (20 μm thick), human follicular lymphoma (50 μm thick), and human diffuse large B-cell lymphoma (50 μm thick). Detailed information for each sample is provided in Supplementary Table 1.

Proceed with deparaffinization by placing a puncture or section into 1 mL of xylene (Histolab, cat. no. 2070) (three times), followed by ethanol (100% EtOH (VWR BDH Chemicals, cat. no. VWRC20816.552); twice) for 5 mins. Subsequently incubated the sample in a sequential rehydration series: 95%, 70%, 50%, and 30% EtOH, followed by incubation in sterilized water and PBS (Thermo Fisher Scientific, cat. no. 10010023) for 5 minutes each. Tissue microdissection was performed to obtain fine tissue pieces using a sharp blade. The tissue pieces were enzymatically digested with 1 mL of a collagenase (3 mg/mL, Sigma-Aldrich, cat. no. C9263) and hyaluronidase (1 mg/mL, Sigma-Aldrich, cat. no. HX0514) cocktail mix in PBS containing 0.5 mM CaCl₂ (Alfa Aesar, cat. no. J63122), 50 μg/mL sodium azide (Sigma-Aldrich, cat. no. C9263), and 100 μg/mL ampicillin (Serva, cat. no. A9518) at 37 °C for 4 h at 850 rpm. The sample was then washed with 1 mL PBS, and 1 mL tissue homogenization buffer (250 mM sucrose (Sigma-Aldrich, cat. no. S0389), 25 mM KCl (Thermo Fisher Scientific, cat. no. AM9640G), 5 mM MgCl₂ (Thermo Fisher Scientific, cat. no. AM9530G), 20 mM Tricine-KOH pH 7.8, 1 mM DTT (Thermo Fisher Scientific, cat. no. A39225), 0.5 mM spermidine (Sigma-Aldrich, cat. no. S2626), 0.15 mM spermine (Thermo Fisher Scientific, cat. no. 32750010), 0.3% IGEPAL CA-630 (Sigma-Aldrich, cat. no. I3021), and complete Protease Inhibitor Cocktail (Sigma-Aldrich, 11873580001). In addition, 0.1% Triton X-100 (Sigma-Aldrich, cat. no. T8787) was added to the tissue homogenization buffer for mouse spleen, human lymph node, and human lymphoma samples. The mixture was gently pipetted up and down and

incubated on ice for 20 min for tissue lysis. Tissue homogenization was performed using a Dounce homogenizer (Merck, cat. no. D9938) containing partially digested tissue pieces in homogenization buffer. The sample was processed with 10 strokes using a loose pestle, filtered through a 30 μm filter (MACS® SmartStrainers, cat. no.130-098-458), then subjected to another 10 strokes using a tight pestle and filtered sequentially through 20 μm and 10 μm filters (pluriStrainer Mini, cat. no.43-10020-50 and 43-10010-50). For human lung cancer samples, tissue homogenization buffer was directly added for tissue lysis. After 20 min of tissue lysis on ice, the sample was mixed by pipetting up and down (10-20 times) and directly filtered through 30 μm, 20 μm, and 10 μm filters without loose and tight homogenization. Following filtration, added 0.1% RNase A (Thermo Fisher Scientific, cat. no. EN0531) and incubated at room temperature (RT) for 5 min. The samples were then centrifuged at 400 g for mouse spleen, human lymph node and human lymphoma samples, and 1000 g for human lung samples at 4 °C for 5 min to obtain nuclei pellets. The nuclei were again pelleted and resuspended in 300 μL of fresh homogenization buffer.

To remove cell debris and extracellular matrices from the single-cell nuclei suspension, density gradient centrifugation was performed with modifications in previous publication[40]. First, the nuclei suspension was mixed (1:1) with a 50% iodixanol solution (Merk, cat.no. D1556) to create a 25% gradient mix containing a total of 600 μL of nuclei. This mixture was then loaded onto the top layer of the density gradient containing 600 μL per density gradient mix: 36% and 48% for mouse spleen, human lymph node and human lymphoma, and 33.5% and 48% for human lung cancer, respectively in 2 mL DNA LoBind tube (Eppendorf, cat. no. 0030108078). The tubes containing the gradient layers were centrifuged at 3,000 g for 20 min at 4 °C in a pre-chilled swinging bucket centrifuge. After centrifugation, 300 μL of the nuclei band visible at the 25%-36% (or 25%-33.6%)) interface (top layer) was collected and transferred to a 1.5 mL Protein LoBind tube (Sarstedt, cat. no.72.706.600). The nuclei suspension was then diluted by adding an equal volume (300 μl) of RSB-T buffer (10 mM Tris-HCl pH 7.5 (Thermo Fisher Scientific, cat. no. 15567027), 10 mM NaCl (Thermo Fisher Scientific, cat. no. AM9760G), 3 mM MgCl₂ (Thermo Fisher Scientific, cat. no. AM9530G), and 0.1% Tween-20 (Sigma-Aldrich, cat. no. P1379)). The tubes were gently mixed by pipetting and centrifuged at 600 g for 5 min to collect the nuclei. Finally, the nuclei pellet was resuspended in 200-500 μL of PBS containing sodium azide (50 μg/mL) and stored at 4 °C.

The quality and purity of isolated nuclei is a critical determinant of the success and reproducibility of single-cell FFPE-ATAC sequencing libraries. In our protocol development, we identified two major factors influencing nuclei quality: enzymatic digestion efficiency and debris removal. The digestion time could be tissue-dependent. Both the experimental considerations and the quality control measures used to optimize nuclei isolation and assess nuclei quality are described in Supplementary Technical Note 1. Comprehensive step-by-step instructions and notes are provided on protocols.io[76]: https://www.protocols.io/view/scffpe-atac-for-high-throughput-single-cell-chroma-4r3l21oq3g1y/v1.

### DNA adaptor sequence for Tn5 transposase and DNA sequence for ligation

The DNA oligonucleotides used in this study were synthesized at Integrated DNA Technologies (IDT), and all detailed DNA sequences for DNA adaptors, DNA indexing sequences, DNA ligation oligos, and PCR primers are provided in Supplementary Table 2.

### Tn5 transposase production

Hyperactive Tn5 transposase was produced following published procedures[77]. In brief, pTXB1-Tn5 plasmid (Addgene, cat. no. 60240) was introduced into T7 Express LysY/Iq Escherichia coli strain (New England Biolabs, cat. no. C3013) separately. 10 mL of overnight

cultured E. coli was inoculated to 500 mL LB medium. After incubation for 1.5 h at 37 °C, bacteria were incubated about 2.5 h at RT. When the OD600 = 0.9, Tn5 protein was induced by adding 0.25 mM IPTG for 4 h. E. coli pellet was resuspended in lysis buffer (20 mM HEPES–KOH (HEPES: Sigma-Aldrich, cat. no. H3375; KOH: Sigma-Aldrich, cat. no. 484016) pH 7.2, 0.8 M NaCl (Invitrogen, cat. no. AM9759), 1 mM EDTA (Invitrogen, cat. no. AM9260G), 10% glycerol (Sigma-Aldrich, cat. no. G9012), 0.2% Triton X-100 (Sigma-Aldrich, cat. no. T8787), complete proteinase inhibitor (Roche, cat. no.11697498001)) and lysed by sonication. 10% PEI was added to supernatant of lysate to remove bacterial genomic DNA. 10 mL chitin resin (New England Biolabs, cat. no. S6651L) was added to the supernatant and incubated with rotating for 1 h at 4 °C. The resin washed by lysis buffer extensively. In order to cleave Tn5 protein from intein, lysis buffer containing 100 mM DTT was added to the resin and stored in 4 °C. After 48 h, protein was eluted by gravity flow and collected in 1 mL fractions. 1 μl of each fraction was added to detergent compatible Bradford assay (Thermo Fisher Scientific, cat. no. 23246) and peaked fractions were pooled and dialyzed with 2× dialysis buffer (100 mM HEPES–KOH (HEPES: Sigma-Aldrich, cat. no. H3375; KOH: Sigma-Aldrich, cat. no. 484016) at pH 7.2, 0.2 M NaCl (Invitrogen, cat. no. AM9759), 0.2 mM EDTA (Invitrogen, cat. no. AM9260G), 2 mM DTT (Thermo Fisher scientific, cat. no. 20291), 0.2% Triton X-100 (Sigma-Aldrich, cat. no. T8787), 20% glycerol (Sigma-Aldrich, cat. no. G9012)). Dialyzed Tn5 protein were concentrated by using ultracel 30-K column (Millipore, cat. no. UFC903024) and the quantity of Tn5 was measured by Bradford assay and visualized on NuPAGE Novex 4–12% Bis–Tris gel (Thermo Fisher Scientific, cat. no. NP0321) followed by Coomassie blue staining.

## FFPE-Tn5 transposase assembly

Oligonucleotides (FFPE_Tn5_DNA_#1-64) (Supplementary Table 2) were resuspended in oligo resuspension buffer (10 mM Tris–HCl pH 8.0 (Invitrogen, cat. no. 15568-025), 1 mM EDTA (Invitrogen, cat. no. AM9260G)) to a final concentration of 100 μM each. Equimolar amounts of ME_bottom_blocked and FFPE_Tn5_DNA_#1-64 were mixed in separate 200 μL PCR tubes. Then, the adaptors were annealed on the PCR machine with the following PCR program (95 °C for 5 min first, then the temperature was slowly ramped down to 20 °C with the rate of −0.1 °C/s, 20 °C for 5 min). The Tn5 transposase was assembled with the following components: 4 μM. of ME_bottom_blocked/ FFPE_Tn5_DNA ramped hybrid oligonucleotides, 40% of 100% glycerol (Sigma-Aldrich, cat. no. G9012), 0.61x of 2× dialysis buffer containing ((100 mM HEPES – KOH (HEPES: Sigma-Aldrich, cat. no. H3375; KOH: Sigma-Aldrich, cat. no. 484016) at pH 7.2, 0.2 M NaCl (Invitrogen, cat. no. AM9759), 0.2 mM EDTA (Invitrogen, cat. no. AM9260G), 2 mM DTT (Thermo Fisher Scientific, cat. no. 20291), 0.2% Triton X-100 (Sigma-Aldrich, cat. no. T8787), 20% glycerol (Sigma-Aldrich, cat. no. G9012)) and 2 μM pure Tn5. Finally, the reaction mixture volume was made up to 50 μL by adding nuclease-free water (Invitrogen, cat. no. AM9932), followed by gently mixed, and incubated 1 hr at 25 °C for annealing of oligos to Tn5.

## Tn5 transposase activity quantification and reaction volume determination

The final concentration of Tn5 transposase in our study was determined following a previous report[43]. A brief description of the process is as follows: The homemade Tn5 was diluted with the dialysis buffer (50 mM HEPES–KOH pH 7.2 (Sigma-Aldrich, cat. no. H3375), 100 mM NaCl (Thermo Fisher Scientific, cat. no. AM9760G), 0.1 mM EDTA (Invitrogen, cat. no. AM9260G), 1 mM DTT (Thermo Fisher Scientific, cat. no. A39255), 0.1% Triton X-100 (Sigma-Aldrich, cat. no. T8787), and 50% glycerol (Sigma-Aldrich, cat. no. G9012)) at different concentrations. Tagmentation was performed on 50 ng of human genomic DNA (Promega, cat. no. G3041) instead of cells. We quantified the number of cycles required to reach one-third of the plateau fluorescence by qPCR

and determined the final dilution factor of homemade Tn5 that showed the most similar number of cycles as Nextera TDE1. In the bulk ATAC-seq assay, a 50 μL reaction system was used, while in the single-cell split-and-pool assay, a 100 μL reaction system was used to avoid nuclei aggregation, but with the same final enzyme concentration (Supplementary Technical Note 2).

## ATAC-Seq in FFPE samples

The standard Tn5 transposase was synthesized following published procedures[77]. The assembled Tn5 was used to tagment 50,000 isolated mouse FFPE nuclei. Post-tagmentation, the sample was divided for processing with and without reverse crosslinking condition (50 mM Tris-HCl, pH 8, 250 mM NaCl, 1 mM EDTA, and 1% SDS and 0.24 mg/mL proteinase K) for the FFPE samples. Fresh mouse spleen nuclei were isolated, and Omni-ATAC was conducted following established protocols[40]. DNA purification was performed using a Qiagen MinElute PCR Purification Kit (Qiagen, cat. no. 28004). DNA library preparation was executed using the purified DNA, NEBNext high-fidelity 2× PCR master mix (New England Biolabs, cat. no. M0541S), and unique forward (i5) and reverse (i7) primer combinations (DNA oligo sequences for PCR amplification). PCR amplification was carried out with the following cycling program: 72 °C for 5 min, 98 °C for 30 sec for 12 cycles as follows: 98 °C for 10 sec, 63 °C for 30 sec, and 72 °C for 1 min.

## scATAC-Seq in FFPE samples

Details of the DNA sequences for DNA adaptors, DNA indexing sequences, DNA ligation oligos, and PCR primers are provided in Supplementary Table 2. Oligonucleotides were resuspended in oligo resuspension buffer (10 mM Tris–HCl pH 8.0 (Invitrogen, cat. no. 15568-025), 0.1 mM EDTA (Invitrogen, cat. no. AM9260G)) to a final concentration of 100 μM each. Equimolar amounts of ME_bottom_blocked and FFPE_Tn5-DNA, as well as ME_bottom_blocked and Tn5-ME-A, were annealed in separate PCR tubes on a PCR machine at 95 °C for 5 min, followed by a gradual decrease to 20 °C at a rate of −0.1 °C/s. Equal volumes of the two differently ramped tubes were combined to form an FFPE Tn5/ME-A annealed oligo mix. The FFPE Tn5/ME-A transposase was subsequently assembled in a manner analogous to the standard Tn5 transposase assembly.

Three DNA-barcoded plates were prepared by ramping the barcodes (Ligation1_#1-96, Ligation2_#1-96, and Ligation3_#1-96, sequences without T7 sequences) with their respective linkers (Linker-Ligation 1, Linker-Ligation 2, and Linker-Ligation 3) oligos in a 96-well plate. Each well contained 1.2 μL of 100 μM barcode and 8.8 μL of 12.5 μM linker oligos. The plates were sealed and subjected to annealing in a thermocycler, as detailed in the Materials and "Methods" section. Comprehensive details of the DNA oligo sequences are provided in Supplementary Table 2. Before tagmentation, 50,000 purified FFPE single-cell nuclei were centrifuged at 2000 g for 5 min at 4 °C and resuspended in 0.1% lysis buffer (10 mM Tris-Cl, pH 7.5 (Thermo Fisher Scientific, cat. no. 15567027), 10 mM NaCl (Thermo Fisher Scientific, cat. no. AM760G), 3 mM MgCl$_2$ (Thermo Fisher Scientific, cat. no. AM9530G), 0.1% IGEPAL CA-630 (Sigma-Aldrich, cat. no. I3021)). The nuclei were then centrifuged again at 2000 g for 5 min at 4 °C. The nuclei pellet was resuspended in 95 μL of 1× tagmentation buffer (10 mM Tris-HCl, pH 7.5 (Thermo Fisher Scientific, cat. no. 15567027), 5 mM MgCl$_2$ (Thermo Fisher Scientific, cat. no. AM9530G), 10% dimethylformamide (Sigma-Aldrich, cat. no. D4551)), and 5 μL of barcoded Tn5 transposase (2 μM) was added per sample. Twenty barcoded FFPE nuclei reactions were mixed, followed by combinatorial indexing through three rounds of ligation using the split-and-pool technique. After third ligation, nuclei pellets were resuspended in PBS, counted, and divided into two tubes, each containing 50,000 nuclei, with and without reverse crosslinking buffer (50 mM Tris-HCl pH 8.0 (Invitrogen, cat. no. 15568-025), 250 mM NaCl (Thermo Fisher Scientific, cat. no. AM760G), 1 mM EDTA (Invitrogen, cat. no. AM9260G), and 1% SDS

(Thermo Fisher Scientific, cat. no. 1553-035) and 0.24 mg/mL proteinase K (Thermo Fisher Scientific, cat. no. E00491)). DNA purification was performed using the Qiagen MinElute PCR Purification Kit (Qiagen, cat. no. 28004), following the manufacturer's protocol. PCR amplification was carried out with the following cycling program: 72 °C for 5 min, 98 °C for 30 sec for 20 cycles as follows: 98 °C for 10 sec, 63 °C for 30 sec, and 72 °C for 1 min.

## Genomic DNA purification from FFPE and fresh nuclei

Genomic DNA was extracted from both fresh and formalin-fixed, paraffin-embedded (FFPE) mouse spleen nuclei utilizing TRIzol reagent (Invitrogen, cat. no. 15596026) in accordance with the manufacturer's protocol. Subsequently, the DNA pellet was resuspended in TE buffer (10 mM Tris−HCl pH 8.0 (Invitrogen, cat. no. 15568-025), 0.1 mM EDTA (Invitrogen, cat. no. AM9260G)). To assess the integrity of the DNA, all samples were subjected to electrophoresis on a 1% agarose gel.

Genomic DNA purification from human lung tumor, human lymph node, and human lymphoma was performed using the following procedure. One 1-mm punch core of human lung tumor and one 5-µm-thick tissue section from human lymph node and human lymphoma were cut from the stock tissue block. Deparaffinization and rehydration were performed following the procedure stated above. The tissue punch and tissue section were dissected into small pieces, then placed in reverse crosslinking buffer (50 mM Tris-HCl pH 8.0 (Invitrogen, cat. no. 15568-025), 250 mM NaCl (Thermo Fisher Scientific, cat. no. AM760G), 1 mM EDTA (Invitrogen, cat. no. AM9260G), and 1% SDS (Thermo Fisher Scientific, cat. no. 1553-035) and 0.24 mg/mL proteinase K (Thermo Fisher Scientific, cat. no. E00491)) and incubated in a thermomixer at 1200 rpm overnight at 65 °C. The next day, DNA was purified using the Zymo ChIP DNA Clean & Concentrator Kit (Zymo Research, cat. no. D5205), according to the manufacturer's instructions. After purification, the size distribution of the genomic DNA was measured using the Agilent Tapestation with the D5000 ScreenTape assay (Agilent, cat. no. 5067-5589).

## Oligo ramping for scFFPE-ATAC

Three DNA barcoded plates were prepared by ramping the barcodes (Ligation1_#1-96, Ligation2_#1-96, and Ligation3_#1-96) with respective linkers (Linker-Ligation 1, Linker-Ligation 2, and linker- Ligation 3) oligos in a 96-well petri plate containing 1.2 µL of 100 µM barcode plus 8.8 µL of 12.5 µM linker oligos per well. The plates were then sealed and annealed in a thermocycler with the following program: 95 °C for 5 min, then slow cooling to 20 °C with a temperature ramp of −0.1 °C/s. The ramped oligos could be stored at -20 °C until use. All detail of DNA oligos sequence is provided in Supplementary Table 2.

## scFFPE-ATAC: Tagmentation and combinatorial indexing via ligation

Before tagmentation, 50,000 purified FFPE single-cell nuclei were centrifuged at 2000 g for 5 min at 4 °C and resuspended in 0.1% lysis buffer (10 mM Tris-Cl, pH 7.5 (Thermo Fisher Scientific, cat. no. 15567027), 10 mM NaCl (Thermo Fisher Scientific, cat. no. AM760G), 3 mM MgCl₂ (Thermo Fisher Scientific, cat. no. AM9530G), 0.1% IGE-PAL CA-630 (Sigma-Aldrich, cat. no. I3021)). The nuclei were then centrifuged again at 2,000 g for 5 min at 4 °C. The nuclei pellet was resuspended in 95 µL of 1× tagmentation buffer (10 mM Tris-HCl, pH 7.5 (Thermo Fisher Scientific, cat. no. 15567027), 5 mM MgCl₂ (Thermo Fisher Scientific, cat. no. AM9530G), 10% dimethylformamide (Sigma-Aldrich, cat. no. D4551)), and 5 µL of barcoded FFPE-Tn5 transposase (2 µM) was added per sample. The reaction mix was gently mixed and incubated at 37 °C for 30 min at 400 rpm. The reaction was stopped by directly adding an equal volume (100 µl) of 50−60 mM EDTA (Invitrogen, cat. no. AM9260G) per sample. Twenty barcoded FFPE nuclei

reactions were then pooled and centrifuged to obtain a nuclei pellet. Next, three rounds of ligation were performed using the split-and-pool technique in a 96-well plate, with each well containing a ramped specific linker and ligation barcode. Detailed DNA sequences are provided in Supplementary Table 2. In brief, the ligation process was performed as follows: the pooled nuclei were resuspended in 1 mL of 1× NEBuffer 3.1 (New England Biolabs, cat. no. B7203S) and mixed with the ligation mix (100 µL of 10× NEBuffer 3.1, 22 µL of 50 mg/mL BSA (Miltenyi Biotech MACS, cat. no. 130-091-376), 500 µL of 10× T4 DNA ligase buffer, 2278 µL of ultrapure water (Invitrogen, cat. no. AM9932), and 100 µL of T4 DNA ligase (New England Biolabs, cat. no. M0202L)). A total of 40 µL of this mix was transferred per well, containing 10 µL of the ramped Ligation1 mixture, and incubated at 37 °C for 30 min at 400 rpm. Then, 10 µL of Blocker-Ligation1 solution (2.64 µL of 100 µM Blocker-Ligation1, 2.50 µL of 10× T4 ligation buffer, and 4.86 µL of ultrapure water) was added per well, and the reactions were incubated at 37 °C for 20 min at 400 rpm. The samples were then pooled into a 15 mL DNA LoBind tube (Eppendorf, cat. no.0030122208) pre-coated with 0.5% BSA (Miltenyi Biotech MACS, cat. no. 130-091-376), followed by centrifugation at 1500 g for 10 min at 4 °C. The second and third ligation reactions were performed identically to the first. After the second and third ligations, blocking reactions were carried out using: 10 µL of Blocker-Ligation2 solution per well (2.64 µL of 100 µM Blocker-Ligation2, 2.50 µL of 10× T4 ligation buffer, and 4.86 µL of ultrapure water); 7.5 µL of Terminator_Ligation3 solution per well (2.64 µL of 100 µM Terminator_Ligation3, 2.50 µL of 0.5 M EDTA (Invitrogen, cat. no. AM9260G), and 2.36 µL of ultrapure water (Invitrogen, cat. no. AM9932)) for reaction termination. After gently mixing, the samples were pooled again into a 15 mL DNA LoBind tube pre-coated with 0.5% BSA (Miltenyi Biotech MACS, cat. no. 130-091-376), followed by centrifugation at 1500 g for 10 min at 4 °C. After the third ligation, each nucleus acquired a unique combination of barcodes. Finally, the pellet was resuspended in PBS, nuclei were counted, and the sample was split into tubes containing 30,000−80,000 nuclei per tube for reverse crosslinking.

## scFFPE-ATAC: Reverse-crosslinking, gap filling and in vitro transcription

After the third ligation, reverse cross-linking buffer (50 mM Tris-HCl pH 8.0 (Invitrogen, cat. no. 15568-025), 250 mM NaCl (Thermo Fisher Scientific, cat. no. AM760G), 1 mM EDTA (Invitrogen, cat. no. AM9260G), and 1% SDS (Thermo Fisher Scientific, cat. no. 1553-035) and 0.24 mg/mL proteinase K (Thermo Fisher Scientific, cat. no. EO0491) were added to each tube, following previous publications[38,44]. The reaction mixture was incubated in a thermomixer at 1200 rpm overnight at 65 °C. The next day, a second proteinase K digestion (0.24 mg/mL) was performed at 37 °C for 2 h at 850 rpm to ensure complete protein digestion. DNA was then purified using the Zymo ChIP DNA Clean & Concentrator Kit (Zymo Research, cat. no.D5205), following the manufacturer's instructions. For gap filling, an equal volume of NEBNext High-Fidelity 2X PCR Master Mix (New England Biolabs, cat. no. M0541S) was added to the eluted DNA, and the reaction mixture was incubated at 72 °C for 8 min in a thermal cycler. The sample was then purified using the Qiagen MinElute PCR Purification Kit (Qiagen, cat. no.28004), following the manufacturer's instructions. Further DNA purification was carried out using the 1x SPRIselect beads (Beckman Coulter, cat. no.B23317) to remove fragments shorter than 150 bp. Finally, the DNA was eluted in nuclease-free water. For in vitro transcription (IVT), a T7 high yield RNA synthesis kit (New England Biolabs, cat. no. E2040S) was used. The following reagents were mixed:

- 12.4 µL of template DNA
- 2.5 µL of 10X T7 buffer
- 1 µL of DTT
- 2 µL each of ATP, CTP, UTP, and GTP
- 1 µL of T7 RNA polymerase

- 0.1 µL of RNase inhibitor (Thermo Fisher Scientific, cat. no. 10777019)

The IVT mixture was incubated at 37 °C overnight (16 h), followed by DNase I treatment and RNA purification using the ZYMO RNA Clean & Concentrator-5 Purification Kit (Zymo Research, cat. no.R1013), according to the manufacturer's instructions. Finally, RNA was eluted in 20 µL of nuclease-free water. The RNA concentration was measured using NanoDrop, and the sample was either immediately used for DNA library preparation or stored at −80 °C for future use.

## Single cell FFPE-ATAC DNA library preparation

Single-stranded cDNAs are prepared from 0.5–1 µg of purified IVT transcripts using the FirstStrand_cDNA oligo (DNA oligo sequences are provided in Supplementary Table 2) with SMART MMLV kit (TAKARA, cat. no. 639524). The reaction mixture (11.25 µL), containing 500ng-1 µg of RNA, 1 µL of 100 µM FirstStrand_cDNA oligo, and 0.25 µL of RNase Inhibitor (20 U/µL, Thermo Fisher Scientific, cat. no. N8080119), is heated at 70 °C for 3 min, then immediately cooled on ice. Next, the master mix is added, which consists of:

- 4 µL of 5× First-Strand Buffer (from SMART MMLV kit (Takara, cat. no. 639524))
- 2 µL of dNTP Mix (Thermo Fisher Scientific, cat. no. R0192)
- 2 µL of 100 mM DTT (from SMART MMLV kit (Takara, cat. no. 639524))
- 0.25 µL of RNase Inhibitor (20 U/µL) (Thermo Fisher Scientific, cat. no. N8080119)
- 0.5 µL SMART MMLV Reverse Transcriptase (from SMART MMLV kit (Takara, cat. no. 639524))

The reaction is gently mixed and incubated at 42 °C for 60 min and then at 70 °C for 15 min. Next, 2.2 µL of 10× RNase H buffer and 0.2 µL of RNase H (5 U/µL, Thermo Fisher Scientific, cat. no. EN0201) are added to each reaction and incubated at 37 °C for 20 min. The RNAClean XP beads (Beckman Coulter, cat. no. A63987) are used (with 1.8x) to purify the cDNA reaction mixture, and the cDNA is eluted in 20 µL of Qiagen Elution Buffer. A sequencing library mix is prepared using purified cDNA (20 µL), 25 µL of NEBNext high-fidelity 2× PCR master mix (New England Biolabs, cat. no. M0541S), and 0.4 µL of 10 µM of unique forward (i5) and 0.4 µL of 10 µM of reverse (i7) primer combination (DNA oligo sequences are provided in Supplementary Table 1). PCR amplification is run with the following cycling program:

- 98 °C for 30 sec
- 12 cycles of:
- 98 °C for 20 sec
- 63 °C for 20 sec
- 72 °C for 1 min

The sample is purified using the Qiagen MinElute PCR Purification Kit (Qiagen, cat. no. 28004) and run on a 6% PAGE gel. The gel region corresponding to 250–800 bp is cut, and the gel is made into fine pieces by placing cut gel slice into 0.5 mL punched tube (made a hole in the bottom of tube with 21 G needle (BD Microlance, cat. no. 302200)) inside 2 mL tube, centrifuge at 16,000 × g for 5 min and discard the 0.5-mL punched tube, leaving the small gel pieces collected in the 2-mL Eppendorf tube which is incubated overnight at 55 °C in 300 µL of crush-soak buffer (500 mM NaCl (Thermo Fisher Scientific, cat. no. AM9760G), 1 mM EDTA (Invitrogen, cat. no. AM9260G), 0.5% SDS (Thermo Fisher Scientific, cat. no. 1553-035)) at 1200 rpm. The DNA is purified from the gel using Costar spin-X centrifuge (Coster, cat. no.8162) tubes and the Zymo ChIP DNA Clean & Concentrator Kit (Zymo Research, cat. no.D5205), following the manufacturer's instructions. Agilent High sensitivity DNA kit (Agilent, cat. no. AGLS5067-4626) is used to check the DNA library size distribution and quantify the DNA concentration of the DNA library in the bioanalyzer.

Finally, deep sequencing is performed with Illmina NovaSeq X Plus on the mouse and human scFFPE-ATAC samples for single-cell analysis.

## Collision rate estimation

A triple-round barcoding strategy was used during hybridization and ligation, utilizing 96 unique barcodes per round. The collision rate was estimated using the birthday paradox approach, as implemented in SHARE-seq[43]. It was determined using the following formula:

$$\text{Collision rate(\%)} = \frac{N - D + D(\frac{D-1}{D})^N}{N} \times 100, \tag{1}$$

Where $N$ represents the total number of cells, and $D$ denotes the total number of barcode combinations within a sub-library. Each sub-library contained $96 \times 96 \times 96$ barcodes, and with an input of 50,000 cells, the resulting collision rate was estimated to be 2.77%.

## Quantification of nuclei recovery

Quantification of nuclei recovery was performed for each experiment using an automated cell counter (Fisher Scientific, cat. no. 16832556). Detailed recovery information for each step is provided in Supplementary Table 1. In brief, nuclei recovery rates during density gradient centrifugation ranged from 29.33% to 67% across different FFPE samples. After split-and-pool ligation, 11.05%–22.36% of cells were captured (specifically, 11.05%–18.94% for mouse FFPE spleen, 16.50% for human FFPE lymph node, 22.50% for human lung FFPE tumor, and 22.36% for paired human primary and relapse lymphoma). For example, starting with 1 million cells for split-and-pool yielded approximately 100,000–200,000 cells after processing.

## Cost

The nuclei isolation and library preparation cost for scFFPE-ATAC in our hands is approximately $632 for 100,000–200,000 cells. This includes about $57 for oligos and $575 for chemicals and enzymes (e.g., Tn5, T4 DNA ligase, etc.), as detailed in Supplementary Table 24.

## Demultiplexing split-pooled single cell ATAC-seq data

An in-house script was developed to demultiplex the split-pool single-cell FFPE-ATAC data and remove adapter sequences from genomic reads. Sample-specific Tn5 indices, cellular barcodes, and linker sequences were extracted from Read 2. The full barcoding sequence ranges from 88 to 91 bases in length. The Tn5 index is 3 bases long, the cellular barcodes are 7 bases each, and the linker sequence spans 17 bases. The first bases of BC #1, BC #2, BC #3, and the Tn5 index are expected to be located at the 1st, 22–26th, 43–47th, and 66–68th positions of Read 2, respectively. To accurately identify the location of all barcodes and the Tn5 index, we used the linker sequence between BC #2 and BC #3 as an anchor, allowing up to three mismatches. A total of 96 barcodes were used, and since the last five bases of each barcode provide sufficient distinction, we leveraged these bases to enhance the demultiplexing rate. We extracted the last 50 bases from Read 2 as the genomic sequence and trimmed Tn5-ME sequences from both Read 1 and Read 2. Finally, Read 1 and Read 2 corresponding to each Tn5 index were merged into a single FASTQ file, with cellular barcode information stored in the header lines of each file. The detailed scripts and code are provided in the GitHub link: https://github.com/pengweixing/scFFPE.

## Fragment length optimization in FFPE samples for genome mapping

Formalin-fixed, paraffin-embedded (FFPE) samples typically yield highly fragmented and degraded genomic DNA. To maximize library complexity, read-length parameters for single-cell decoding and mapping were optimized. In our sequencing setup, R1 contains genomic DNA with adaptors, while R2 contains genomic DNA along with

single-cell barcodes. The initial read length was 91 bp. After adaptor trimming, the effective genomic DNA portion in R2 was ~59 bp, from which we retained 50 bp. To maintain consistency, R1 was truncated to 75 bp. Consequently, adaptor-trimmed reads ranged from 1–75 bp in R1 and 1–50 bp in R2. Our debarcoding script includes the parameter -ml, which specifies the minimum retained read length. To identify the optimal read length for FFPE-derived DNA, we tested a series of minimum thresholds (50 bp down to 14 bp) for both R1 and R2 using 50 million paired sequencing reads from mouse FFPE spleen. Reads were mapped to the mouse genome (mm10) with BWA (v0.7.17)[78]. Each dataset was then processed through the single-cell analysis pipeline, including mapping, duplicate removal, and quality control.

The number of decoded reads, unique fragments, and mapping rate (from 50 million reads) were calculated for each minimum length tested. Reducing the minimum fragment length from 50 bp to 14 bp increased the decoding rate from 48% (23.77 million) to 70% (35.10 million) and increased the number of unique fragments by 35.55% (from 18.18 million to 24.64 million), with only a minor decrease in mapping rate (from 97.85% to 92.68%). To preserve uniquely mapped features and avoid potential multi-mapping fragments, a minimum read length of 17 bp was selected for our study. This strategy maximized usable information from heavily fragmented FFPE DNA while maintaining high mapping quality. The same mapping strategy was applied to both scATAC-seq and scFFPE-ATAC in FFPE samples.

## Pre-processing of demultiplexed single cell data

The data processing workflow was implemented using Snakemake[79]. Sequencing reads were aligned to the mouse reference genome (mm10) (mouse spleen) or the human reference genome (hg38) (human lymph node, human lung cancer or human lymphoma) using BWA (v0.7.17)[78] with the 'mem -k17' algorithm. The resulting SAM files were converted to BAM format, sorted, and indexed using 'samtools (v1.17)[80] view', 'samtools (v1.17) sort', and 'samtools (v1.17) index'. Duplicates originating from both PCR and linear amplification were removed using a custom in-house script. Signal tracks in BigWig format were generated with 'bamCoverage (v3.5.1)' with parameters '--normalizeUsing CPM'. Transcription start site (TSS) enrichment analysis was performed using 'computeMatrix (v3.5.1)" with parameters '--binSize 10 --beforeRegionStartLength 3000 --afterRegionStartLength 3000', followed by heatmap visualization with 'plotHeatmap (v3.5.1)'[81]. BAM files were converted to fragment files using an in-house script. Peak calling was conducted using 'MACS2 (v2.1.2) callpeak'[82] with parameters '--nomodel --shift 0 -q 0.01', and blacklist regions were filtered using 'bedtools (v2.30.0) intersect'[83]. The fraction of reads within the TSS region (FRiT) was calculated using a window spanning -1000 bp to +200 bp relative to the TSS. For mouse samples, high-quality cells were defined as those with FRiT > 10 and > 1500 unique fragments, whereas for human samples, high-quality cells were defined as those with FRiT > 7 and > 1000 unique fragments.

## Single-cell chromatin accessibility clustering and gene activity analysis

Single-cell clustering analysis was performed using the SnapATAC2[84] and Scanpy[85] packages. For mouse samples, we imported data using 'snap.pp.import_data' with the mm10 reference genome and filtered out cells with TSS enrichment scores below 4. A 5 kb bin size was used to generate the tile matrix with 'snap.pp.add_tile_matrix'. For human samples, we used the hg38 reference genome and a 10 kb bin size to construct the tile matrix.

To assess and remove potential doublets, we selected the top 250,000 features and estimated the doublet rate using 'snap.pp.scrublet', followed by filtering with 'snap.pp.filter_doublets'. Top features were retained for dimension reduction using 'snap.pp.select_features' with n_features=250000. Dimensionality reduction was performed using spectral embedding ('snap.tl.spectral'), and cell

clustering was conducted with k-nearest neighbor (KNN) graph construction 'snap.pp.knn'. The batch effects were controlled using the integration pipeline provided in SnapATAC2, specifically applying the snap.pp.harmony(max_iter_harmony=20) function to harmonize data across multiple samples. Leiden clustering 'snap.tl.leiden', and UMAP visualization 'snap.pl.umap'.

Gene activity scores were calculated using 'snap.pp.make_gene_matrix' with parameters 'pstream=5000, downstream=500'. To annotate cell types, cell type-specific marker gene activity was projected onto the UMAP. Lowly expressed genes were filtered using 'sc.pp.filter_genes' with 'min_cells=5', and gene activity normalization was performed using 'sc.pp.normalize_total' followed by log transformation 'sc.pp.log1p'. Data imputation and smoothing were conducted with the MAGIC algorithm 'sc.external.pp.magic'. Highly variable genes were identified using 'scanpy.pp.highly_variable_genes' with 'min_mean=0.0125, max_mean=3, min_disp=0.5'. Differentially expressed genes were determined using 'sc.tl.rank_genes_groups' with the t-test method, and those with an adjusted $p$ value < 0.05 and log fold change (LogFC) > 0.25 were used to compute average expression per cell type and visualized as a heatmap.

Peak calling was performed using 'snap.tl.macs3' with 'groupby=CellType', and peaks were merged using 'snap.tl.merge_peaks' with 'half_width=500'. A peak-to-cell matrix was then generated using 'snap.pp.make_peak_matrix'.

## Pseudotime trajectory analysis from scFFPE-ATAC

To infer cellular trajectories, the gene activity matrix, generated from chromatin accessibility data using SnapATAC2[84], was used as input to represent the transcriptional potential of each cell. Slingshot[86] was applied to the low-dimensional embedding, with cell clusters identified through unsupervised clustering serving as the starting points for lineage inference. To evaluate the robustness of lineage assignments and pseudotime ordering, a bootstrap resampling approach was applied. Cells were resampled with replacement 1000 times, and pseudotime inference was repeated for each bootstrap replicate using the Slingshot framework. Lineages were inferred with the 'getLineages' and 'getCurves' functions, while cluster ordering along each lineage was obtained using 'slingLineages'. Pseudotime values for individual cells were then calculated with the 'slingPseudotime' function, generating a pseudotime matrix and all for downstream summarization.

For each bootstrap replicate, lineage branch support was defined as the proportion of bootstrap replicates in which a given lineage branch was recovered. Pseudotimes were re-aligned to the reference lineages and stored in a three-dimensional array (cells × lineages × bootstraps). To quantify pseudotime uncertainty, we summarized the bootstrap distribution for each cell and lineage. Specifically, the median pseudotime was calculated across all bootstrap replicates, and 95% confidence intervals (CIs) were obtained by taking the 2.5th and 97.5th percentiles of the bootstrap pseudotime distribution.

## Cell-type specific ATAC-Seq peaks identification

The peak-to-cell matrix was generated using the make_peak_matrix function in SnapATAC2 and normalized by sequencing depth for each cell. Cell type–specific peaks were identified using a one-versus-rest strategy, where peaks for each cell type (e.g., T cells) were compared against all remaining cells. To address the sparsity of scFFPE-ATAC-seq data, pseudo-bulk profiles were constructed by randomly sampling 500 cells per condition and computing the average accessibility signal. Multiple testing correction was applied using the Benjamini–Hochberg method (multipletests with fdr_bh), and peaks were considered differentially accessible if they met both an FDR < 0.05 and a fold change ≥ 2. Peaks showing significant differences across more than one cell type were excluded to ensure specificity. The resulting differential peak matrix was derived from the peak-to-cell matrix, and average accessibility values were computed for each cell type.

## Differential peaks identification across conditions

For condition-specific comparisons, the peak-to-cell matrix was generated using make_peak_matrix in Snapatac2 and then normalized by sequencing depth for each cell. To mitigate the sparsity inherent in scFFPE-ATAC-seq data, pseudo-bulk datasets were generated by randomly selecting 500 cells per condition and computing the average accessibility signal. This process was repeated five times for mouse samples and ten times for human clinical samples. Differential accessibility between conditions (e.g., tumor edge versus tumor core) was assessed using a t-test to evaluate statistical significance alongside fold-change calculations. False discovery rate (FDR) correction was performed using the Benjamini-Hochberg method ('multipletests' with 'fdr_bh'), and differentially accessible peaks were defined using an FDR threshold of < 0.05 and a fold change 2. Peaks exhibiting significant differences were considered as differential peaks in different conditions. The final differential peak matrix was extracted from the peak-to-cell matrix, and the condition-level average accessibility was computed accordingly.

## Transcription factor enrichment for differential peaks

For transcription factor (TF) motif enrichment analysis, the peak-to-cell type matrix was used as input for chromVAR[87]. GC bias was corrected using 'addGCBias', and known motifs were retrieved from the JASPAR database using 'getJasparMotifs'. Background peaks were estimated with 'getBackgroundPeak', followed by deviation score computation using 'computeDeviations'. TF variability was assessed with 'computeVariability', and the top 100 most variable TFs were selected for visualization.

## Genomic annotation and GO enrichment

Differential peaks were annotated to corresponding genes using the 'annotatePeak' function in ChIPseeker[88]. Gene ontology (GO) enrichment analysis for genes nearest to the peaks was performed using the 'enrichGO' function from the clusterProfiler package[89], with a $q$ value cutoff of 0.05.

## Data visualization

Bar plots were generated using the 'geom_bar' function from the ggplot2 package. Scatter plots and volcano plots were created using the 'ggscatter' function in the ggpubr package. Heatmaps were visualized with the pheatmap package.

## Data availability

All source Data are provided with this paper. The raw sequencing data generated in this study have been deposited in the NCBI GEO database under accession number GSE291155, GSE299388, GSE306401, and GSE111586 [https://www.ncbi.nlm.nih.gov/geo/query/acc.cgi?acc=GSM3034634]. The Bulk ATAC-Seq from fresh mouse spleen used in this study are available in the NCBI GEO database under accessible number SRP167062. Source data are provided with this paper.

## Code availability

The code to perform all analyses and regenerate all the figures in this study is provided[90]. Code link (https://github.com/pengweixing/scFFPE).

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

## Acknowledgements

This work is supported by Swedish Cancer Society (23 0761 PT to R.P.Y, 24 3484 Pj and 22 0491 JIA to X.C.), Swedish Research Council (2022-00658, 2024-03756 to X.C.), and Knut and Alice Wallenberg foundation (KAW 2023.0046 and KAW 2024.0166 to X.C.). Part of this work was facilitated by the Protein Science Facility at Karolinska Institute, Stockholm. Sequencing was performed by the SNP&SEQ Technology Platform in Uppsala. The facility is part of the National Genomics Infrastructure (NGI) Sweden and Science for Life Laboratory. The SNP&SEQ Platform is also supported by the Swedish Research Council and the Knut and Alice Wallenberg Foundation. The funders played no role in the study design, data acquisition and interpretation, or decision to publish.

## Author contributions

X.C. conceived and designed the study. R.P.Y. performed all single cell experiments., P.X. performed all the data mining in the study. M.X., M.S., and E.T. worked together with R.P.Y. to optimize tissue microdissection. M.Z., P.H., C.S. and M.H. provided samples. R.A., G.E., and P.M. performed pathological scanning. F.J.S., I.G., P.M., M.H. and T.S. with funding support and scientific discussion. R.P.Y. and X. C. wrote the manuscript with input from all authors. X.C. supervised all aspects of this work.

## Funding

## Competing interests

R.P.Y. and X.C. have filed patent applications with Swedish Intellectual Property office (Application No. 2550657-7) related to the work described here. The remaining authors declare no competing interests.
