## [Transparent Peer Review file · Nature Communications]

scFFPE-ATAC enables high-throughput single cell chromatin accessibility profiling in formalin-fixed paraffin-embedded samples

Corresponding Author: Dr Xingqi Chen

Version 2:

Reviewer comments:

Reviewer #1

(Remarks to the Author)

scATAC-seq has played a key role in the research of epigenetic regulatory mechanisms. However, the application of scATAC-seq in FFPE samples has long been limited because of the unsatisfactory data quality. Ram et al. reported a scATAC-seq method for FFPE samples (scFFPE-ATAC), successfully addressing the low data quality issue of scATAC-seq caused by DNA damage in FFPE samples. This method exhibits remarkable technological innovation and provides an important tool for single-cell epigenetic research using FFPE samples, especially demonstrating broad application prospects in the fields of tumor heterogeneity and single-cell epigenomics. The experimental design is rigorous, and the data support the main conclusions, but some technical details need to be further clarified to enhance persuasiveness.

Comments:

1. A schematic diagram of the scFFPE-ATAC library construction details will facilitate readers' quick understanding.
2. In Figure 2b, the TSS enrichment score of scFFPE-ATAC is lower than that of conventional scATAC from FFPE +RV. Please discuss potential mechanisms. For example: 1) Whether multiple alignment caused by short fragments is the main influencing reason; 2) Whether the T7 in vitro transcription causes sequence preference interference; 3) Other technical reasons.
3. Whether the dose of 5 μ l Tn5 transposase used in scFFPE-ATAC is consistent with that in the existing scATAC-seq method (scATAC-seq from fresh or FFPE)? Does this dosage have sample preference?
4. Is scFFPE-ATAC sensitive to the fixation time, paraffin embedding conditions, and storage time of FFPE samples?
5. It's better to provide the maximum number of cells captured and the estimated cost in a single experiment, which is of great significance for clinical application.
6. What are the fragments features (distribution) of the scFFPE-ATAC library?

(Remarks on code availability)

It looks OK.

Reviewer #2

(Remarks to the Author)

(Remarks on code availability)

Reviewer #3

(Remarks to the Author)

The study by Yadav et al. presents scFFPE-ATAC, a novel method for performing single-cell chromatin accessibility profiling on archived FFPE tissue samples. These samples are widely used in clinical and research settings but have historically posed challenges for genomic and epigenomic studies due to RNA and DNA degradation. The authors engineered an FFPE-compatible Tn5 transposase with a T7 promoter, enabling in vitro transcription (IVT) to rescue fragmented DNA. They also applied an ultra-high-throughput barcoding strategy (56 million unique cell barcodes per run) using split-and-pool combinatorial indexing to capture scATAC data from FFPE-isolated nuclei. Additionally, they optimized nuclei isolation using a sucrose gradient specifically tailored for FFPE samples and demonstrated that IVT significantly improves scATAC performance in this context.

The method was benchmarked using FFPE mouse spleen, showing improved library complexity and cell-type resolution (T cells, B cells, myeloid cells), nearly matching the performance seen with fresh tissue. Next, the authors applied scFFPE-ATAC to archived human lymph node samples (8–12 years old), successfully profiling ~12K nuclei across multiple immune cell types, thereby demonstrating robustness across long archival times. Finally, they applied the method to human lung cancer samples, resolving cell types from both the tumor core and invasive edge and identifying spatially distinct regulatory elements and transcription factors, including two epigenetic trajectories of epithelial tumor progression.

In summary, this method enables robust, high-resolution epigenetic profiling from a variety of archived FFPE sample types, overcoming longstanding limitations such as DNA damage and low library complexity. It opens the door for retrospective clinical studies using archives of FFPE material. I support publication of this method in Nature Communications. Below are several comments that I believe would strengthen the manuscript and clarify certain aspects:

1. Sucrose Gradient vs. Sorting

The authors optimized a sucrose gradient (25%, 36%, 48%) to separate nuclei from debris in FFPE samples. While this represents an improvement over standard gradients, they acknowledge the persistence of residual debris and its potential impact on nuclei quality and ATAC signal. The manuscript does not mention attempts to use FACS-based nuclei sorting, which might yield higher purity. Given that the limitations of the gradient method are recognized, could the authors comment on why sorting methods (e.g., PI+ gating) were not used? Was this due to technical, yield-related, or FFPE-specific integrity constraints?

2. Sequencing Length and Alignment

The manuscript notes that sequencing depth was increased 2–3× for FFPE samples due to lower complexity, but it does not appear to provide details on read length, alignment strategies, or whether different read lengths were tested. Given that FFPE DNA is heavily fragmented, shorter read lengths (e.g., 50 bp vs. 100 bp) might improve mapping and complexity. Could the authors clarify what read lengths were used and whether any optimization was attempted?

3. Nuclei Recovery

The paper mentions that, for instance, ~18K nuclei were captured from FFPE spleen using scFFPE-ATAC, and that 76.67% passed quality thresholds. However, more specific recovery metrics would be helpful:

- What was the initial nuclei yield from the sucrose gradient (pre and post) from the different FFPE processed in this study?
- What was the recovery efficiency after split-pool barcoding? Is this different than when doing fresh samples?

A comparative table showing yields at each step (e.g., tissue input → isolation → barcoding → QC-passing nuclei) would be valuable, especially for others attempting this protocol.

4. FFPE Protocol – Fixation and Archiving

The ability to profile samples stored for more than a decade is a major strength. However, can the authors provide details on formalin fixation time? FFPE protocols vary widely across institutions, from overnight fixation to several days or even weeks, especially for older archived samples. Over-fixation increases crosslinking and DNA fragmentation. Including information on fixation conditions and pre-storage variables would improve interpretability and reproducibility.

5. Assessing scFFPE-ATAC Success based on FFPE Sample Quality

The paper provides standard QC metrics (TSS enrichment, duplication rate, unique fragment count), but a pre-ATAC sample quality control metric, analogous to DV200 in FFPE scRNA-seq, would be highly useful. Is there a measurable property (e.g., DNA integrity score, fragment distribution) that correlates with ATAC performance in FFPE? Including such correlations would help users evaluate sample suitability before investing in library prep.

6. Sample Diversity

Most of the data presented come from lymphoid tissues (mouse spleen, human lymph nodes, FL/DLBCL), with one example in lung cancer. While the lung cancer dataset is promising, additional examples in more challenging solid tumors such as pancreatic ductal adenocarcinoma (PDAC) for instance could better demonstrate method robustness across tissue types. If not included here, this could be noted as a valuable future direction.

7. Reproducibility and Protocol Access

To facilitate broader adoption, I strongly encourage the authors to deposit a step-by-step protocol (e.g., on Protocols.io) and, if possible, this would support method replication and adoption by other laboratories.

8. Engineered FFPE-Compatible Tn5

The FFPE-compatible Tn5 enzyme is described as “newly engineered” with a T7 promoter to support IVT-based DNA rescue. However, it appears that a similar enzyme may have been previously published and deposited to Addgene. Could the authors clarify:

- Whether this specific Tn5 is novel?
- If so, how it differs from previously published versions?

If it is reused or modified, please explain the novelty in this context.

In Summary

This is a substantial technical advance for single-cell ATAC profiling from long-archived FFPE tissues. The manuscript is rich in innovation, especially in integrating IVT rescue, but few practical and methodological questions could be better explained. Clarifying points related to nuclei isolation, sample preparation, tissue diversity, and protocol accessibility will strengthen the impact and utility of this important work.

(Remarks on code availability)

I am not a computational biologist and don't feel skilled to review the code.

Reviewer #4

(Remarks to the Author)

1. Summary of Key Findings

The manuscript by Yadav et al. introduces scFFPE-ATAC, a novel method for profiling chromatin accessibility at single-cell resolution in formalin-fixed paraffin-embedded (FFPE) samples. The method combines a custom Tn5 transposase, combinatorial barcoding, and a T7 promoter-based DNA damage rescue strategy. The authors demonstrate the method's utility in mouse spleen, human lymph nodes, and clinical cancer samples, including spatial profiling of lung cancer and analysis of follicular lymphoma relapse. The data show that scFFPE-ATAC recovers cell-type-specific chromatin accessibility patterns in archived FFPE tissues with performance approaching that of fresh samples.

2. Noteworthy Results

This study presents a significant technical advance with scFFPE-ATAC, enabling high-resolution chromatin accessibility profiling in FFPE tissues. Key achievements include:

- Successful resolution of cell-type-specific epigenetic profiles in 8-12 year-old FFPE samples
- Demonstration of spatial epigenetic heterogeneity in lung cancer (center vs. invasive edge)
- Identification of relapse-associated epigenetic trajectories in lymphoma

3. Significance and Originality

The work addresses a critical gap in translational epigenomics by overcoming FFPE limitations. Its potential to unlock archival clinical samples represents a substantial advance beyond existing methods (e.g., FACT-seq, bulk ATAC). The spatial and longitudinal applications could significantly impact:

- Cancer evolution studies
- Biomarker discovery from clinical archives
- Retrospective therapeutic response analysis

4. Suggestions for Improvement

- Reproducibility and Protocol Details:

- > The enzymatic digestion step (collagenase/hyaluronidase) uses concentrations that may vary by tissue type. Please provide more details on optimization and quality control metrics for nuclei isolation across different sample types (e.g., how nuclear integrity was assessed).
- > If relevant, include a troubleshooting guide or common pitfalls in the Methods to aid reproducibility.

- Biological Validation:

- > For the lymphoma relapse analysis, the conclusions about transcription factor drivers (e.g., ZEB1 in FL relapse) would be strengthened by either orthogonal validation (e.g., immunohistochemistry on the same samples) in the Results or literature support in the Discussion.
- > In the lung cancer spatial analysis, include a discussion of how microenvironmental factors (e.g., hypoxia) might influence the observed epigenetic trajectories.

- Statistical Rigor:

- > The pseudotime trajectory analysis in Figures 4k and 5i lacks metrics for trajectory robustness (e.g., bootstrapping confidence intervals). Please add statistical validation of the branching patterns.
- > For differential peak calling, specify how batch effects were controlled in the human samples (e.g., integration methods).

- Technical Limitations:

Discuss the method's sensitivity limits (e.g., minimum input requirements, performance in highly degraded samples) and failure rates.

- Minor Typos:

- Line 375: probably missed listing specific TFs
- Line 936: extra comma
- Figure 2a: typo chromatin

Figure 2b & 3a: Enrichment

Fig 3: Archived

Fig 5: Patient

5. Methodology and Reproducibility

The methods are well-described overall, but the following additions would enhance reproducibility:

- > Provide exact reagent catalog numbers for critical steps (e.g., T4 ligase).
- > Include representative images of nuclei pre/post-purification for different sample types.
- > Clarify computational parameters for key analyses (e.g., MACS2 peak calling settings, SnapATAC2 parameters).

(Remarks on code availability)

The provided code (`get_qc.r`) is well documented within the script, with clear parameter specifications for quality control of scFFPE-ATAC data, and it uses standard R packages.

It successfully reproduces the key filtering and plotting steps described in the manuscript. To further improve usability and reproducibility, the Authors should consider:

- adding a brief README that includes installation instructions, usage examples, and especially a description or example of the expected input file format (such as the required headers and column structure for `Spleen.singlecell.qc.txt`)
- Replace hard-coded thresholds (e.g., 100000 fragments) with parameters and explain biological rationale for default thresholds.
- Consolidate duplicated FRiP/FRiT code blocks into functions
- Add version requirements (e.g., `ggplot2>=3.4.0`)

Version 3:

Reviewer comments:

Reviewer #1

(Remarks to the Author)

All my concerns have been well addressed. Given the rigor of its experimental design and the persuasiveness of its data, this manuscript now meets a high standard of quality.

(Remarks on code availability)

The code is OK.

Reviewer #2

(Remarks to the Author)

(Remarks on code availability)

Reviewer #3

(Remarks to the Author)

After reviewing the rebuttal, I am satisfied that the authors have addressed all of my concerns, and I have no further comments.

(Remarks on code availability)

Reviewer #4

(Remarks to the Author)

I did not identify any outstanding concerns regarding the data analysis, interpretation, or conclusions in the revised version. The methodology meets the expected standards for reproducibility and robustness, and the reporting is complete. All reviewer points have been thoroughly addressed, with method clarification, protocol expansion, statistical robustness, QC metric transparency, figure revision, and code documentation now clearly visible in the resubmitted manuscript. In summary, the work represents an important and convincingly demonstrated technical advance that will be of broad interest for epigenomics, pathology, and translational cancer research.

(Remarks on code availability)

Reviewer requests for improved documentation and clarity on input formats have been addressed in this version, and with these updates, the code serves as a valuable and accessible resource for the community. The repository includes a README file with installation and usage instructions, and the major scripts are clearly organized. Quality control and cell selection workflows are well annotated, and example data and parameters are provided, which facilitates reproducibility of the main analyses presented in the manuscript. The expected format for key input files (such as the single-cell QC tables) is described, making it straightforward for users to adapt the pipeline to their own data.

1

2 **Re: Revision of NCOMMS-25-22823B-Z**3 **Summary of revision:**

We would like to express our sincere gratitude to the reviewers for their thoughtful
evaluation of our manuscript and for their constructive comments, which have been
invaluable in improving the quality and clarity of our work. In response to the
reviewers' feedback, we have thoroughly revised the manuscript. The major
revisions are summarized below:

- 1. We created a detailed protocol for our scFFPE-ATAC and deposited it
on *protocols.io*.
<https://www.protocols.io/private/3E7641CD72FF11F0AB4F0A58A9FEAC02>
- 2. We have provided a detailed description of our revised methodology, including
both the experimental procedures and the computational pipeline, along with
the corresponding protocols.io version.
- 3. We added detailed information on sample preparation (fixation, embedding,
storage, etc.) and included recovery rates for each experimental step in the
sample sheet.
- 4. We included detailed instructions for nuclei isolation and density gradient
centrifugation in two Supplementary Technical Notes.
- 5. We described the pre-assessment of FFPE samples for scFFPE-ATAC,
based on the quality and purity of purified FFPE nuclei, and provided the
corresponding data in the revised manuscript.
- 6. We performed statistical tests of trajectory robustness in our pseudotime
analysis of human lung cancer and paired primary and relapse samples.
- 7. We expanded the discussion to address the limitations of our method.
- 8. We uploaded the new data to GEO.

A detailed, point-by-point response to each reviewer's comment is provided below:

Reviewer #1 (Remarks to the Author):

scATAC-seq has played a key role in the research of epigenetic regulatory
 mechanisms. However, the application of scATAC-seq in FFPE samples has long
 been limited because of the unsatisfactory data quality. Ram et al. reported a
 scATAC-seq method for FFPE samples (scFFPE-ATAC), successfully addressing
 the low data quality issue of scATAC-seq caused by DNA damage in FFPE samples.
 This method exhibits remarkable technological innovation and provides an important
 tool for single-cell epigenetic research using FFPE samples, especially
 demonstrating broad application prospects in the fields of tumor heterogeneity and
 single-cell epigenomics. The experimental design is rigorous, and the data support
 the main conclusions, but some technical details need to be further clarified to
 enhance persuasiveness.

**RESPONSE:** We sincerely thank the reviewer for the positive evaluation of our study
 design and the support for our main conclusions. We appreciate the reviewer's
 suggestion regarding clarification of technical details. In the revised manuscript, we
 have added additional methodological information to improve clarity and
 reproducibility. We believe these revisions address the reviewer's concern and
 further strengthen the persuasiveness of our work.

**ACTION:** Accordingly, we have thoroughly revised the manuscript and supplied
 detailed, point-by-point responses to each of your comments.

Comments:

1. A schematic diagram of the scFFPE-ATAC library construction details will facilitate
 readers' quick understanding.

**RESPONSE:** Thank you for your insightful comments. We agreed the schematic
 diagram of scFFPE-ATAC library construction will facilitate readers' quick
 understanding.

**ACTION:** We have added the schematic workflow of the library construction in the
 revised **Supplementary Figure 6** (see **Rebuttal Figure 1**).

Rebuttal Figure 1: The workflow of scFFPE-ATAC sequencing library construction.

2. In Figure 2b, the TSS enrichment score of scFFPE-ATAC is lower than that of

69 conventional scATAC from FFPE +RV. Please discuss potential mechanisms. For
example: 1) Whether multiple alignment caused by short fragments is the main
influencing reason; 2) Whether the T7 in vitro transcription causes sequence
preference interference; 3) Other technical reasons.

**RESPONSE:** Thank you for your valuable comments. We have observed a similar
pattern in both bulk and single-cell FFPE-ATAC. One potential explanation is that
standard scATAC-seq in FFPE samples relies on PCR amplification using two
different Tn5 adaptors, without additional DNA breaks or damage within the amplified
DNA fragments. In contrast, the T7 in vitro transcription-mediated method requires
only a single T7 promoter adaptor insertion. Even if DNA breaks or damage occur
between two insertion sites, amplification of the insertion site can still proceed. While
this approach can rescue damaged DNA regions, it may also introduce noise into the
sequencing library, as flanking regions with a single insertion could also be amplified.
Another possibility is that T7 in vitro transcription introduces sequence preference
bias, which could interfere with library uniformity. We respectfully disagree with the
suggestion that multiple alignments caused by short fragments are the main
influencing factor, as our alignment rates are similar between fresh/frozen samples
(92.68%) and FFPE samples (94.26%) (Revised Supplementary Table 1).

**ACTION:** In the revised manuscript, we discussed such valuable points, revised lines
295-302.

3. Whether the dose of 5 μ L Tn5 transposase used in scFFPE-ATAC is consistent
with that in the existing scATAC-seq method (scATAC-seq from fresh or FFPE)?
Does this dosage have sample preference?

**RESPONSE:** Thank you for your valuable comments. In conventional ATAC-seq and
scATAC-seq, a 50 μ L reaction system is typically used¹⁻⁴. To avoid potential nuclei
aggregation during Tn5 tagmentation, we instead used a 100 μ L reaction system
with 5 μ L of Tn5 transposase in all single-cell experiments. The final concentration of
Tn5 transposase, however, remained the same. We hypothesize that the
concentration of Tn5 transposase, rather than the reaction volume, primarily
determines tagmentation efficiency.

The final concentration of Tn5 transposase in our study was determined following a
previous report⁵. A brief description of the process is as follows: The homemade Tn5
was diluted with the dilution buffer (50 mM Tris, 100 mM NaCl, 0.1 mM EDTA, 1 mM
DTT, 0.1% NP-40, and 50% glycerol) at different concentrations. Tagmentation was
performed on 50 ng of purified genomic DNA instead of cells. We quantified the
number of cycles required to reach one-third of the plateau fluorescence by qPCR
and determined the final dilution factor of homemade Tn5 that showed the most
similar number of cycles as Nextera TDE1.

To confirm our hypothesis that the concentration of Tn5 transposase does not have
a dosage effect, we performed two experiments:

1. We tested Tn5 transposase activity in both 50 μ L reactions (with 2.5 μ L Tn5
transposase) and 100 μ L reactions (with 5 μ L Tn5), keeping the final

concentration of Tn5 transposase constant. Notably, there are more free DNA
 adaptors in the 100 μ L reaction system (with 5 μ L Tn5). We observed that the
 activity was similar in both reaction volumes, with comparable size
 distributions of genomic DNA after tagmentation (Revised **Supplementary**
 **Technical Note Figure 2a**; see **Rebuttal Figure 2a**).

 2. We conducted a single-cell FFPE-ATAC experiment using mouse FFPE
 spleen in a 50 μ L reaction system and found that the quality of scFFPE-ATAC
 was comparable to that obtained with the 100 μ L system (as shown in the first
 submission), in terms of TSS enrichment, cell-type detection, and single-cell
 epigenetic profiles (Revised **Supplementary Technical Note Figure 2b–d**;
 see **Rebuttal Figure 2b–d**).

**Rebuttal Figure 2: Different reaction with same concentration of Tn5 transposase for scFFPE-ATAC. a, *In***
 ***vitro* Tn5 transposase activity comparison with 50 μ L and 100 μ L reaction volume but with same concentration; b,**
 **The TSS enrichment score in single cell FFPE-ATAC with different reaction volume; c, Cell type identification in**
 **single cell FFPE-ATAC with different reaction volume, left panel: UMAP, right panel: proportion comparison; d:**
 **Examples of active genes in each cell type.**

In summary, our results indicate that the concentration of Tn5 transposase—rather
 than the reaction volume or total enzyme dosage—is the primary determinant of
 tagmentation efficiency. Therefore, our results indicate that both the 50 μ L and 100
 μ L reaction systems are effective.

**ACTION:** We have provided detailed information on our reaction system in all
 single-cell experiments in the revised Methods. To give readers a clearer
 understanding, we included an explanation of how we determined the Tn5
 concentration, together with both an *in vitro* Tn5 activity assay using different
 reaction volumes and scFFPE-ATAC experiments with mouse FFPE spleen in a 50
 μ L reaction system. These results are now presented in Revised **Supplementary**
 **Technical Note 2**.

4. Is scFFPE-ATAC sensitive to the fixation time, paraffin embedding conditions, and
 storage time of FFPE samples?

**RESPONSE:** Thank you for this insightful comment. In our study, we analyzed
mouse FFPE spleen, human FFPE lymph node, human FFPE lung tumor, as well as
paired primary and relapse human FFPE lymphoma samples.

All specimens were fixed in buffered 4% formalin, with fixation times ranging from 16
to 72 hours. The mouse spleen samples were prepared in our laboratory with a
fixation time of 16 hours, while the human FFPE samples were obtained from the U-
CAN biobank⁶. Although the exact fixation times for the human samples are difficult
to determine, they followed routine clinical diagnostic protocols and fell within the
16–72 hour range.

For the paraffin embedding, all samples were processed under standardized
conditions at the U-CAN biobank. Briefly, after fixation the tissues were dehydrated
and paraffinized in a vacuum infiltration process overnight using a Tissue-Tek VIP 6
Al tissue processor (Sakura Finetek) (one hour in 70% ethanol, 2.5 hours in 95%
ethanol, 3.5 hours in 99.5% ethanol, 4 hours in xylene, and 5.5 hours in paraffin).

The storage durations ranged from 2 to 13 years for human samples and up to 6
171 months for mouse samples.

Despite variability in fixation times and storage durations, we consistently obtained
high-quality single cell chromatin accessibility profiles, indicating that scFFPE-ATAC
is robust across a wide range of sample preparation conditions. However, we did not
test samples fixed for more than 72 hours, which are rare in clinical practice. We
recognize this as a limitation of our study and have noted it in the revised
Discussion.

**ACTION:** We have now included detailed information on fixation times, embedding
conditions, and storage durations in the Methods and in Revised **Supplementary**
**Table 1**. In addition, we added a brief discussion to the manuscript regarding
variability in fixation times and other factors in clinical FFPE processing (revised
**lines 833–836**).

Here come the text in the revised manuscript:
We successfully tested samples fixed for 16–72 hours and stored for 6 months to 13
188 years, but further optimization of protocols for samples with longer fixation (>72
189 hours) or storage durations⁷ will be an important direction for future development.

5. It's better to provide the maximum number of cells captured and the estimated
cost in a single experiment, which is of great significance for clinical application.

**RESPONSE:** Thank you for your insightful comments. We agree that providing
detailed information on the number of cells captured and the estimated cost per
experiment is of great significance for clinical applications.

In our experiments, we captured 11.05%–22.36% of cells after split-and-pool ligation
(specifically, 11.05%–18.94% for mouse FFPE spleen, 16.50% for human FFPE
lymph node, 22.50% for human lung FFPE tumor, and 22.36% for paired human
primary and relapse lymphoma). For example, starting with 1 million cells for split-

and-pool yielded approximately 100,000–200,000 cells after processing. In addition,
 we have included details of nuclei recovery rates during gradient centrifugation,
 which ranged from 29.33% to 67% across different FFPE samples.

We also calculated the cost per experiment. The total cost was approximately \$632
 (including ~\$57 for oligos and ~\$575 for chemicals and enzymes such as T4 DNA
 ligase and Tn5 transposase).

**ACTION:** In the revised manuscript, we now provide the number of cells captured
 212 per experiment after gradient centrifugation and split-and-pool, as well as the
 213 estimated cost of a single-cell experiment, both in the revised Methods
 **Supplementary Table 1 and Table 24.**

6. What are the fragments features (distribution) of the scFFPE-ATAC library?

**RESPONSE:** Thank you for the comments. As expected, DNA in FFPE samples is
 highly fragmented. After sequencing and adaptor trimming, the size distributions of
 scFFPE-ATAC libraries were as follows: 98.76% in the range of 0–300 bp for mouse
 spleen FFPE samples (stored ~6 months), 99.33% in the range of 0–200 bp for
 human lung FFPE tumor samples (stored ~2 years), 96.28% in the range of 0–100
 222 bp for human lymph node FFPE samples (stored ~8–12 years), and 97.72% in the
 223 range of 0–100 bp for human follicular lymphoma FFPE samples (stored ~6–13
 224 years) (**Rebuttal Figure 3**). The split size distributions for each sample are provided
 in the revised **Supplementary Figures: 7b** (mouse spleen FFPE), **11c** (human
 lymph node FFPE LN1–4), **13e** (human lung FFPE), and **18e** (human follicular
 lymphoma FFPE).

**Rebuttal Figure 3: Fragment size distribution of scFFPE-ATAC libraries from different FFPE samples.**
 Top row: fragment length distribution (counts). Bottom row: fragment length distribution (percentage).

**ACTION:** This information was not included in the initial submission. In the revised
 manuscript, we have added the fragment size distribution of scFFPE-ATAC, which is
 now presented in revised **Supplementary Figures 7b** (mouse spleen FFPE), **11c**
 (human lymph node FFPE LN1–4), **13e** (human lung FFPE), and **18e** (human
 follicular lymphoma FFPE), and described in the main text.

Reviewer #1 (Remarks on code availability):

It looks OK.

**RESPONSE:** Thank you for your insightful comments.

**ACTION:** In the revised manuscript (Methods section) and in our shared GitHub
repository, we have updated the code and instructions to improve usability and
reproducibility. Specifically, we: (1) added a README file to facilitate usage; (2)
replaced hard-coded thresholds with adjustable parameters and provided detailed
explanations of the biological rationale for the default values; (3) specified the
system requirements for all relevant packages and dependencies; and (4) included a
schematic workflow of our data-mining process.

Reviewer #2 (Remarks to the Author):

I co-reviewed this manuscript with one of the reviewers who provided the listed
reports. This is part of the Nature Communications initiative to facilitate training in
peer review and to provide appropriate recognition for Early Career Researchers
who co-review manuscripts.

**RESPONSE:** We sincerely thank you for your valuable comments and suggestions
on our work.

**ACTION:** We have carefully revised the manuscript accordingly and provided
detailed, point-by-point responses to each comment.

Reviewer #3 (Remarks to the Author):

The study by Yadav et al. presents scFFPE-ATAC, a novel method for performing
single-cell chromatin accessibility profiling on archived FFPE tissue samples. These
samples are widely used in clinical and research settings but have historically posed
challenges for genomic and epigenomic studies due to RNA and DNA degradation.
The authors engineered an FFPE-compatible Tn5 transposase with a T7 promoter,
enabling in vitro transcription (IVT) to rescue fragmented DNA. They also applied an
ultra-high-throughput barcoding strategy (56 million unique cell barcodes per run)
using split-and-pool combinatorial indexing to capture scATAC data from FFPE-
isolated nuclei. Additionally, they optimized nuclei isolation using a sucrose gradient
specifically tailored for FFPE samples and demonstrated that IVT significantly
improves scATAC performance in this context.

The method was benchmarked using FFPE mouse spleen, showing improved library
complexity and cell-type resolution (T cells, B cells, myeloid cells), nearly matching
the performance seen with fresh tissue. Next, the authors applied scFFPE-ATAC to
archived human lymph node samples (8–12 years old), successfully profiling ~12K
nuclei across multiple immune cell types, thereby demonstrating robustness across
long archival times. Finally, they applied the method to human lung cancer samples,
resolving cell types from both the tumor core and invasive edge and identifying
spatially distinct regulatory elements and transcription factors, including two
epigenetic trajectories of epithelial tumor progression.

In summary, this method enables robust, high-resolution epigenetic profiling from a
variety of archived FFPE sample types, overcoming longstanding limitations such as
DNA damage and low library complexity. It opens the door for retrospective clinical
studies using archives of FFPE material. I support publication of this method in
Nature Communications.

**RESPONSE:** We are grateful for your positive evaluation and for the constructive
comments and suggestions you provided, which have been invaluable in improving
the quality of our work.

**ACTION:** Accordingly, we have thoroughly revised the manuscript and supplied
detailed, point-by-point responses to each of your comments.

Below are several comments that I believe would strengthen the manuscript and
clarify certain aspects:

1. Sucrose Gradient vs. Sorting

The authors optimized a sucrose gradient (25%, 36%, 48%) to separate nuclei from
debris in FFPE samples. While this represents an improvement over standard
gradients, they acknowledge the persistence of residual debris and its potential
impact on nuclei quality and ATAC signal. The manuscript does not mention
attempts to use FACS-based nuclei sorting, which might yield higher purity. Given
that the limitations of the gradient method are recognized, could the authors
comment on why sorting methods (e.g., PI+ gating) were not used? Was this due to
technical, yield-related, or FFPE-specific integrity constraints?

**RESPONSE:** Thank you for your comments. When we began this project, our group
did not yet have expertise in FACS sorting. Following the Omni-ATAC protocol^{3,8}, we

first optimized gradient centrifugation and observed that nuclei from FFPE samples
were enriched in different layers compared with those from frozen or fresh samples.
We fully agree that FACS-based nuclei sorting (e.g., PI⁺ gating)⁹⁻¹¹ is an effective
approach to achieve higher purity when a FACS machine is available, particularly for
extracellular matrix-rich samples such as pancreatic ductal adenocarcinoma
(PDAC).

**ACTION:** To provide readers with a fair comparison of nuclei isolation methods, we
included descriptions of both FACS sorting and gradient centrifugation (Revised **line**
**111-117**), along with a discussion of the potential advantages of FACS-based sorting
for challenging samples (Revised **line 824-826**), such as pancreatic ductal
adenocarcinoma (PDAC), which we highlight as a potential future direction.

2. Sequencing Length and Alignment

The manuscript notes that sequencing depth was increased 2–3× for FFPE samples
due to lower complexity, but it does not appear to provide details on read length,
alignment strategies, or whether different read lengths were tested. Given that FFPE
DNA is heavily fragmented, shorter read lengths (e.g., 50 bp vs. 100 bp) might
improve mapping and complexity. Could the authors clarify what read lengths were
used and whether any optimization was attempted?

**RESPONSE:** We thank the reviewer for this insightful comment. In our sequencing
data, we agree that it is important to provide detailed information regarding
sequencing length and alignment strategies.

All sequencing in our study was performed using 2 × 150 bp reads on the Illumina
NovaSeq X Plus system. In our sequencing libraries, R1 (Read 1) contains genomic
DNA with adaptors, while R2 (Read 2) contains genomic DNA together with single-
cell barcodes, with an initial read length of 91 bp. After adaptor trimming, the
effective genomic DNA length in R2 was ~59 bp, of which we retained 50 bp. For
FFPE samples, most genomic DNA fragments are highly degraded and short. To
account for this, we truncated R1 to 75 bp for input. After adaptor trimming, R1 reads
ranged from 1–75 bp and R2 reads from 1–50 bp.

To optimize read length for FFPE-derived DNA, we tested a range of minimum
retained fragment lengths (50 bp down to 14 bp) for paired R1 and R2 reads, using a
random subset of 50 million fragments from a mouse spleen FFPE ATAC-seq library.
As shown in **Rebuttal Figure 4** (Revised **Supplementary Figure 3b**), reducing the
minimum fragment length from 50 bp to 14 bp increased the decoding rate from 48%
(23.77 million) to 70% (35.10 million) and the number of unique fragments by
35.55% (from 18.18 million to 24.64 million), with only a minor decrease in mapping
rate (from 97.85% to 92.68%).

To preserve uniquely mapped features and avoid potential multi-mapping fragments,
we chose to retain reads longer than 17 bp for mapping in our study. This strategy
maximized usable information from heavily fragmented FFPE DNA while maintaining

high mapping quality. The same mapping strategy was applied to both scATAC-
 seq and scFFPE-ATAC in FFPE samples.

 **Rebuttal Figure 4: Optimization of fragment length for mapping using 50 million fragments from scATAC-**
 **seq libraries of mouse FFPE spleen samples.** Left panel: number of decoded fragments; middle panel: number
 of unique fragments; right panel: mapping rate.

**ACTION:** The detailed sequencing length, alignment, and attempted strategy are
 provided in the Methods section of the revised manuscript. The data are also
 included in Revised **Supplementary Figure 3b**.

3. Nuclei Recovery

The paper mentions that, for instance, ~18K nuclei were captured from FFPE spleen
 using scFFPE-ATAC, and that 76.67% passed quality thresholds. However, more
 specific recovery metrics would be helpful:

- • What was the initial nuclei yield from the sucrose gradient (pre and post) from the
- different FFPE processed in this study?
- • What was the recovery efficiency after split-pool barcoding? Is this different than
- when doing fresh samples?

A comparative table showing yields at each step (e.g., tissue input → isolation →
 barcoding → QC-passing nuclei) would be valuable, especially for others attempting
 this protocol.s

 **RESPONSE:** Thank you for the suggestion. We agree that it is important to provide
 detailed information for the readers. We have now organized all relevant
 information—including tissue input, nuclei isolation, gradient centrifugation recovery,
 split-and-pool recovery, barcoding, and QC-passing data—and summarized them in
 a supplementary table. Specifically, our tissue input included 1-mm punches from
 mouse spleen FFPE samples and human lung tumors, as well as 20-50- μ m thick
 sections from human lymph node and human follicular lymphoma samples. The
 recovery rate of nuclei isolation using gradient centrifugation ranged from 29.33% to
 67% across different FFPE samples, which is comparable to fresh samples (42.80%
 in fresh mouse spleen vs. FFPE mouse spleen). The split-and-pool recovery ranged
 from 11.05% to 22%. All of this information has been provided in the revised
 **Supplementary Table**.

**ACTION:** We added a section **Quantification of nuclei recovery** in the Methods to
 describe how the recovery rate was calculated at each step, and we also included

detailed calculations of QC metrics for sequencing data processing at each step for
each sample in Revised **Supplementary Table 1**. The detailed recovery rates for
each step are provided in Revised **Supplementary Table 1** and are also described
in the revised **Methods**.

4. FFPE Protocol – Fixation and Archiving

The ability to profile samples stored for more than a decade is a major strength.

However, can the authors provide details on formalin fixation time? FFPE protocols
vary widely across institutions, from overnight fixation to several days or even weeks,
especially for older archived samples. Over-fixation increases crosslinking and DNA
fragmentation. Including information on fixation conditions and pre-storage variables
would improve interpretability and reproducibility.

**RESPONSE:** Thank you for this insightful comment. In our study, we analyzed
mouse FFPE spleen, human FFPE lymph node, human FFPE lung tumor, as well as
paired primary and relapse human FFPE lymphoma samples.

All specimens were fixed in buffered 4% formalin, with fixation times ranging from 16
to 72 hours. The mouse spleen samples were prepared in our laboratory with a
fixation time of 16 hours, while the human FFPE samples were obtained from the U-
CAN biobank⁶. Although the exact fixation times for the human samples are difficult
to determine, they followed routine clinical diagnostic protocols and fell within the
16–72 hour range.

For the paraffin embedding, all samples were processed under standardized
conditions at the U-CAN biobank. Briefly, after fixation the tissues were dehydrated
and paraffinized in a vacuum infiltration process overnight using a Tissue-Tek VIP 6
AI tissue processor (Sakura Finetek) (one hour in 70% ethanol, 2.5 hours in 95%
ethanol, 3.5 hours in 99.5% ethanol, 4 hours in xylene, and 5.5 hours in paraffin).

The storage durations ranged from 2 to 13 years for human samples and up to 6
437 months for mouse samples.

Despite variability in fixation times and storage durations, we consistently obtained
high-quality single cell chromatin accessibility profiles, indicating that scFFPE-ATAC
is robust across a wide range of sample preparation conditions. However, we did not
test samples fixed for more than 72 hours, which are rare in clinical practice. We
recognize this as a limitation of our study and have noted it in the revised
**Discussion**.

**ACTION:** We have now included detailed information on fixation times, embedding
conditions, and storage durations in the **Methods** and in Revised **Supplementary**
**Table 1**. In addition, we added a brief discussion to the manuscript regarding
variability in fixation times and other factors in clinical FFPE processing (revised
**lines 833–836**).

Here come the text in the revised manuscript:

We successfully tested samples fixed for 16–72 hours and stored for 6 months to 13
454 years, but further optimization of protocols for samples with longer fixation (>72
455 hours) or storage durations ⁷ will be an important direction for future development.

5. Assessing scFFPE-ATAC Success based on FFPE Sample Quality

The paper provides standard QC metrics (TSS enrichment, duplication rate, unique
fragment count), but a pre-ATAC sample quality control metric, analogous to DV200
in FFPE scRNA-seq, would be highly useful. Is there a measurable property (e.g.,
DNA integrity score, fragment distribution) that correlates with ATAC performance in
FFPE? Including such correlations would help users evaluate sample suitability
before investing in library prep.

**RESPONSE:** Thank you for the thoughtful comment, which encouraged us to think
more deeply about the workflow. We agree that it is important for users to evaluate
samples before performing scFFPE-ATAC.

When we started this project a couple of years ago, we used mouse FFPE spleen to
benchmark our work, since clinical samples are very precious. During method
evaluation, we optimized many conditions, including tagmentation, washing, nuclei
isolation, and sequencing library construction. Unfortunately, none of these
optimizations enabled us to capture single-cell chromatin accessibility profiles from
mouse FFPE spleen. We then began to consider nuclei purity, particularly that debris
in purified nuclei could negatively affect library quality. To address this, we
introduced density gradient centrifugation, as used in Omni-ATAC for fresh/frozen
nuclei^{3,8}. We found that nuclei and debris from FFPE samples could not be
effectively separated using a 25–30–40% iodixanol interface. However, with a 25–
36–48% iodixanol interface, we were able to separate nuclei (top layer) from debris
(bottom layer).

We did not observe a major difference in DNA size distribution between the
conditions with and without density centrifugation in mouse FFPE spleen nuclei
(**Rebuttal Figure 5a; Supplementary Technical Note Figure 3a**). We used cutoffs
of 200 bp, 300 bp, 400 bp, and 500 bp to measure the ratio of DNA below and above
each cutoff, and the ratios were very similar between samples with and without
density gradient centrifugation.

We compared single-cell FFPE-ATAC libraries prepared with and without density
gradient centrifugation using mouse FFPE spleen tissue. In the scFFPE-ATAC
library analysis (**Rebuttal Figure 5b–f; Supplementary Technical Note Figure 3b–
f**), the libraries prepared without density gradient centrifugation showed lower TSS
enrichment, higher duplication rates, and a lower fraction of reads in peaks (FRiP)
compared with those prepared with density gradient centrifugation (**Rebuttal
Figures 5b, 5c; Supplementary Technical Note Figures 3b, 3c**). We were only
able to decipher single-cell chromatin accessibility profiles in the condition with
density gradient centrifugation; without it, we could not clearly separate cell types or
identify gene activities (**Rebuttal Figures 5d–f; Supplementary Technical Note
Figures 3d–f**).

Thus, we hypothesize that unpurified nuclei, particularly debris, may promote
nonspecific Tn5 binding during tagmentation, as reflected by low TSS enrichment

scores, high duplication rates, and lower FRiP values compared with gradient-
 purified nuclei. In addition, debris may reduce ligation efficiency by sequestering
 DNA oligos, thereby further compromising library quality.

**Rebuttal Figure 5: Comparison of DNA length distribution and scFFPE-ATAC profiles from mouse FFPE**

**spleen nuclei with and without density gradient centrifugation.**

a. DNA length distribution comparisons.

b. Sequencing signal enrichment at transcription start sites (TSSs).

c. Left: duplication rate; middle: number of unique fragments; right: fraction of reads in peaks. Median values are

labeled in each comparison.

516 d. Identification of cell types from scFFPE-ATAC under different conditions.

e. Proportional distribution of different cell types or clusters.

f. Examples of active genes in each cell type or cluster.

We also purified DNA from human lymph nodes (stored for 8–12 years), human lung

tumors (stored for 2 years), and paired primary and relapse human lymphoma

samples (stored for 6–13 years), and quantified the DNA fragment size distributions

from these samples (**Rebuttal Figure 6; Supplementary Technical Note Figure 4**).

The DNA showed a wide range of fragment sizes (15–10,000 bp), from very short to

long, after years of storage under different conditions. We observed very few DNA

fragments shorter than 100 bp across all samples, and in each case more than 80%

of DNA fragments were longer than 200 bp. We used cutoffs of 200 bp, 300 bp, 400

529 bp, and 500 bp to measure the ratio of DNA fragments below and above each

530 threshold, and these ratios showed diverse distributions among samples (**Rebuttal**

**Figure 6; Supplementary Technical Note Figure 4**). Importantly, despite this

diversity in DNA fragment distribution, we did not observe significantly lower

scFFPE-ATAC quality in these samples.

531
532

c

Ratio Samples	N ≥ 200bp N < 200bp	N ≥ 300bp N < 300bp	N ≥ 400bp N < 400bp	N ≥ 500bp N < 500bp
Human lymph node (LN1)	88.7% : 11.3%	79% : 21%	70.1% : 29.9%	62.1% : 37.9%
Human lymph node (LN2)	87.2% : 12.8%	76.9% : 23.1%	67.7% : 32.3%	59.7% : 40.3%
Human lymph node (LN3)	91.4% : 8.6%	83.7% : 16.3%	76.0% : 24.0%	68.9% : 31.1%
Human lymph node (LN4)	89.0% : 11.0%	80.4% : 19.6%	72.6% : 27.4%	65.2% : 34.8%
Human FL Patient 1 primary (P1P)	95.2% : 4.8%	89.7% : 10.3%	83.7% : 16.3%	77.8% : 22.2%
Human FL Patient 1 relapse (P1R)	94.6% : 5.4%	88.3% : 11.7%	81.8% : 18.2%	75.4% : 24.6%
Human FL Patient 2 primary (P2P)	93.5% : 6.5%	85.9% : 14.1%	78.6% : 21.4%	71.4% : 28.6%
Human FL Patient 2 relapse (P2R)	80.2% : 19.8%	63.8% : 36.2%	49.9% : 50.1%	39.5% : 60.5%
Human lung tumor (LT)	94.8% : 5.2%	90.4% : 9.6%	85.2% : 14.8%	79.9% : 20.1%

533

Rebuttal Figure 6: Quantification of DNA length distribution in archived human FFPE samples, including lymph nodes (LN1–4), Patient 1 primary follicular lymphoma (P1P), Patient 1 relapse follicular lymphoma (P1R), Patient 2 primary follicular lymphoma (P2P), Patient 2 relapse follicular lymphoma (P2R), and a human lung tumor (LT).

- a. Gel images from Tapestation.
- b. DNA length distribution ratio of fragments ≥200 bp vs. <200 bp.
- c. DNA length distribution ratios with cutoffs at 200 bp, 300 bp, 400 bp, and 500 bp.

In our protocol development, we identified two major factors influencing nuclei quality and purity: enzymatic digestion efficiency and debris removal. Below, we outline the experimental considerations and quality control measures used to optimize nuclei isolation. We have also included this point in the technical note of the revised manuscript.

1. Enzymatic digestion to release nuclei

Complete enzymatic digestion is essential for releasing nuclei from the surrounding extracellular matrix. Incomplete digestion can result in nuclear doublets and hinder accurate single-cell chromatin profiling, particularly when distinct cell types such as T cells and tumor cells are embedded together within fibrotic tissue regions. We optimized the enzymatic digestion step using collagenase and hyaluronidase, noting that both enzyme concentration and digestion time vary significantly by tissue type. To determine optimal conditions, we performed a matrix of digestion experiments varying enzyme concentrations and incubation durations. At each time point, small aliquots were taken, and digestion efficiency was assessed under a fluorescence

microscope using nuclear staining (e.g., DAPI or Hoechst) (**Rebuttal Figure 7;**
 **Supplementary Technical Note Figure 1).**

Rebuttal Figure 7: Comparison of human lung tumor nuclei after complete and incomplete digestion, visualized by nuclear staining under microscopy.

2. Density Gradient Centrifugation for Debris Removal

The second key parameter was gradient centrifugation, which proved essential for
 removing extracellular matrix components and cellular debris. In the absence of
 density gradient centrifugation, we observed considerable debris, whereas with
 density gradient centrifugation, little to no debris was present (**Rebuttal Figure 8;**
 **Supplementary Technical Note Figure 2).**

Rebuttal Figure 8: Comparison of mouse FFPE spleen nuclei with and without density gradient centrifugation. The white box indicates the zoomed-in area of the upper row shown in the lower row.

Thus, our current findings indicate that assessing nuclei purity after nuclei isolation, rather than relying on DNA fragment size distribution, is a more reliable parameter prior to scFFPE-ATAC. DV200 (Distribution Value 200; the percentage of RNA

fragments >200 nucleotides) is a widely used parameter for RNA-seq and single-cell
RNA-seq in FFPE samples^{12,13}, but it is not applied to DNA measurement in FFPE
samples¹⁰. Our measurement focuses on accessible chromatin DNA fragments,
where even shorter fragments can be enriched by adding a T7 promoter at the end
of the DNA fragment.

However, we also acknowledge the limitations of this measurement and have
addressed this point in the Discussion section. In the future, it will be valuable to
perform a larger cohort study to compare DNA fragment distributions and to
establish a standardized metric based on DNA fragment size, analogous to DV200 in
FFPE single-cell RNA-seq^{12,13}. We have included this point in the revised manuscript
as one of the limitations of our study.

**ACTION:** We have included our new results in **Supplementary Technical Note 1**
**of the revised manuscript, describing how to assess scFFPE-ATAC success based**
**on the purity of isolated FFPE nuclei. We also discuss the potential limitations in the**
**Discussion section. (revised line 828-832).**

6. Sample Diversity

Most of the data presented come from lymphoid tissues (mouse spleen, human
lymph nodes, FL/DLBCL), with one example in lung cancer. While the lung cancer
dataset is promising, additional examples in more challenging solid tumors such as
pancreatic ductal adenocarcinoma (PDAC) for instance could better demonstrate
method robustness across tissue types. If not included here, this could be noted as a
valuable future direction.

**RESPONSE:** Thank you for the valuable suggestion. In our current manuscript, we
included lymphoid tissue and lung cancer samples. We agree that it would be highly
valuable to include more challenging samples, such as pancreatic ductal
adenocarcinoma (PDAC). PDAC is well known for its dense extracellular matrix
(ECM), often referred to as desmoplastic stroma. The ECM can comprise a large
portion of the tumor mass—sometimes more than 80%—and consists of
components such as collagen, fibronectin, hyaluronic acid, and proteoglycans, along
with stromal cells including fibroblasts, immune cells, and endothelial cells.

Unfortunately, we do not currently have access to such samples. We acknowledge
that handling PDAC tissue, particularly for nuclei isolation, is highly challenging. To
process such tissue, one may need to increase digestion time and use additional
enzymes, such as trypsin. It may also be necessary to employ FACS sorting to
obtain pure nuclei for downstream applications.

**ACTION:** To provide the audience with a clearer picture, in the revised manuscript,
we discussed this valuable future direction (**Revised line 820-828**).

7. Reproducibility and Protocol Access

To facilitate broader adoption, I strongly encourage the authors to deposit a step-by-
step protocol (e.g., on [Protocols.io](https://www.protocols.io)) and, if possible. This would support method

replication and adoption by other laboratories.

**RESPONSE:** Thank you for the great suggestion.

**ACTION:** We created a step-by-step protocol with detailed notes
on [protocols.io](https://www.protocols.io) and included the link in the revised manuscript. The link is as below:

<https://www.protocols.io/private/3E7641CD72FF11F0AB4F0A58A9FEAC02>

8. Engineered FFPE-Compatible Tn5

The FFPE-compatible Tn5 enzyme is described as “newly engineered” with a T7
promoter to support IVT-based DNA rescue. However, it appears that a similar
enzyme may have been previously published and deposited to Addgene. Could the
authors clarify:

•Whether this specific Tn5 is novel?

•If so, how it differs from previously published versions?

If it is reused or modified, please explain the novelty in this context.

**RESPONSE:** Thank you for this helpful comment. To clarify, the core Tn5
transposase protein we use is identical to previously published versions and is not
newly engineered. The novelty of our work lies in the adaptor design: we introduced
an adaptor carrying 64 unique DNA barcodes (without altering the enzyme itself),
which enables indexing of different FFPE samples or spatial locations. Thus, while
the enzyme itself is not novel, our FFPE-Tn5 system represents a methodological
innovation in adaptor engineering and its application to FFPE material.

**ACTION:** In the revised manuscript, we rephrased our description in revised **lines**
**268–272** to clearly convey that our FFPE-Tn5 modification is at the level of the DNA
adaptors rather than the protein.

In Summary

This is a substantial technical advance for single-cell ATAC profiling from long-
archived FFPE tissues. The manuscript is rich in innovation, especially in integrating
IVT rescue, but few practical and methodological questions could be better
explained. Clarifying points related to nuclei isolation, sample preparation, tissue
diversity, and protocol accessibility will strengthen the impact and utility of this
important work.

**RESPONSE:** Thank you for the encouraging comments and great suggestions.

**ACTION:** We revised our manuscript according and the detail is showed above.

Reviewer #3 (Remarks on code availability):

I am not a computational biologist and dont feel skilled to review the code.

**RESPONSE:** Thank you for the comments.

**ACTION:** To enable all potential users, including non-computational biologists, to use
our pipeline, we have added detailed instructions in the code accompanying the

revised manuscript. Specifically, we have updated our code and documentation to
further enhance usability and reproducibility. In the revised manuscript (Methods
section) and in our shared GitHub repository, we have implemented the following
improvements:

- 1. Added a README file to facilitate usage.
- 2. Replaced hard-coded thresholds with adjustable parameters and provided a
detailed explanation of the biological rationale for the default values.
- 3. Specified the system requirements for all relevant packages and
dependencies.
- 4. Included a schematic workflow of our data processing in both GitHub and
Protocols.io.

Reviewer #4 (Remarks to the Author):

1. Summary of Key Findings

The manuscript by Yadav et al. introduces scFFPE-ATAC, a novel method for
profiling chromatin accessibility at single-cell resolution in formalin-fixed paraffin-
embedded (FFPE) samples. The method combines a custom Tn5 transposase,
combinatorial barcoding, and a T7 promoter-based DNA damage rescue strategy.
The authors demonstrate the method's utility in mouse spleen, human lymph nodes,
and clinical cancer samples, including spatial profiling of lung cancer and analysis of
follicular lymphoma relapse. The data show that scFFPE-ATAC recovers cell-type-
specific chromatin accessibility patterns in archived FFPE tissues with performance
approaching that of fresh samples.

2. Noteworthy Results

This study presents a significant technical advance with scFFPE-ATAC, enabling
high-resolution chromatin accessibility profiling in FFPE tissues. Key achievements
include:

- - Successful resolution of cell-type-specific epigenetic profiles in 8-12 year-old FFPE
- samples
- - Demonstration of spatial epigenetic heterogeneity in lung cancer (center vs.
- invasive edge)
- - Identification of relapse-associated epigenetic trajectories in lymphoma

3. Significance and Originality

The work addresses a critical gap in translational epigenomics by overcoming FFPE
limitations. Its potential to unlock archival clinical samples represents a substantial
advance beyond existing methods (e.g., FACT-seq, bulk ATAC). The spatial and
longitudinal applications could significantly impact:

- - Cancer evolution studies
- - Biomarker discovery from clinical archives
- - Retrospective therapeutic response analysis

**RESPONSE:** We are grateful for your positive evaluation and for the constructive
comments and suggestions you provided, which have been invaluable in improving
the quality of our work.

**ACTION:** Accordingly, we have thoroughly revised the manuscript and supplied
detailed, point-by-point responses to each of your comments.

4. Suggestions for Improvement

- Reproducibility and Protocol Details:

> The enzymatic digestion step (collagenase/hyaluronidase) uses concentrations
that may vary by tissue type. Please provide more details on optimization and quality
control metrics for nuclei isolation across different sample types (e.g., how nuclear
integrity was assessed).

> If relevant, include a troubleshooting guide or common pitfalls in the Methods to aid

reproducibility.

**RESPONSE:** Thank you for the great suggestion.

The quality and purity of isolated FFPE nuclei are critical determinants of the
success and reproducibility of single-cell FFPE-ATAC sequencing libraries.

In our protocol development, we identified two major factors influencing nuclei quality
and purity: enzymatic digestion efficiency and debris removal. Below, we outline the
experimental considerations and quality control measures used to optimize nuclei
isolation. We have also included this point in the technical note of the revised
manuscript.

**1. Enzymatic digestion to release nuclei**

Complete enzymatic digestion is essential for releasing nuclei from the surrounding
extracellular matrix. Incomplete digestion can result in nuclear doublets and hinder
accurate single-cell chromatin profiling, particularly when distinct cell types such as T
cells and tumor cells are embedded together within fibrotic tissue regions. We
optimized the enzymatic digestion step using collagenase and hyaluronidase, noting
that both enzyme concentration and digestion time vary significantly by tissue type.
To determine optimal conditions, we performed a matrix of digestion experiments
varying enzyme concentrations and incubation durations. At each time point, small
aliquots were taken, and digestion efficiency was assessed under a fluorescence
microscope using nuclear staining (e.g., DAPI or Hoechst) (**Rebuttal Figure 7**;
**Supplementary Technical Note Figure 1**).

**Rebuttal Figure 7: Comparison of human lung tumor nuclei after complete and incomplete digestion,**
**visualized by nuclear staining under microscopy.**

**2. Density Gradient Centrifugation for Debris Removal**

The second key parameter was gradient centrifugation, which proved essential for
removing extracellular matrix components and cellular debris. In the absence of
density gradient centrifugation, we observed considerable debris, whereas with
density gradient centrifugation, little to no debris was present (**Rebuttal Figure 8**;
**Supplementary Technical Note Figure 2**).

**Rebuttal Figure 8: Comparison of mouse FFPE spleen nuclei with and without density gradient centrifugation.**

The white box indicates the zoomed-in area of the upper row shown in the lower row.

We evaluated nuclei quality using the following two criteria:

1. Morphological Integrity

Nuclei were examined under a fluorescence microscope following nuclear
 staining. High-quality nuclei appeared intact, round, and evenly stained, with
 minimal clumping or fragmentation. The presence of nuclei doublets, irregular
 shapes, or signs of lysis were used as indicators of suboptimal enzymatic
 digestion or incomplete removal of extracellular matrix.

2. Debris Content

Debris was visually assessed in the stained nuclei preparation. Low-quality
 samples displayed abundant granular debris and extracellular contaminants,
 which could interfere with downstream enzymatic steps such as Tn5
 tagmentation and ligation during the split-and-pool barcoding workflow. We
 aimed to minimize debris as much as possible, through multiple rounds of
 gradient centrifugation. An alternative strategy is to incorporate flow
 cytometric sorting, combining nuclear staining with gating strategies to
 exclude debris.

**ACTION:** We created a step-by-step protocol with detailed notes on [Protocols.io](https://www.protocols.io)
 and included the link in the revised manuscript. The protocol also contains a
 troubleshooting guide and highlights common pitfalls in the methods.

<https://www.protocols.io/private/3E7641CD72FF11F0AB4F0A58A9FEAC02>

In addition, we have added a technical note on nuclei isolation in the revised
 manuscript (**Supplementary Technical Note 1**).

- Biological Validation:

> For the lymphoma relapse analysis, the conclusions about transcription factor
 drivers (e.g., ZEB1 in FL relapse) would be strengthened by either orthogonal
 validation (e.g., immunohistochemistry on the same samples) in the Results or
 literature support in the Discussion.

Response:

We thank the reviewer for the suggestion to provide orthogonal validation of
transcription factor drivers such as ZEB1 in FL relapse. Unfortunately, due to the
lack of reliable antibodies for ID4, ZEB1, SNAI2 et al in the available samples, we
were unable to perform immunohistochemistry experiments. Instead, in line with the
reviewer's comments, we have incorporated supporting evidence from the literature
into the discussion. In the absence of direct experimental validation, we have
moderated the strength of our statement.

Action:

To address this point, we have strengthened the discussion by citing relevant studies
that support the role of ID4, ZEB1, SNAI2 in tumor relapse¹⁴⁻¹⁸. These references
provide independent evidence for the regulatory function of ZEB1 in similar contexts,
thereby supporting our conclusions. We have also noted that further in-depth studies
will be required to clarify their mechanistic contributions and potential as therapeutic
targets in the future (revised lines 622–632).

> In the lung cancer spatial analysis, include a discussion of how microenvironmental
factors (e.g., hypoxia) might influence the observed epigenetic trajectories.

**Response:** Thank you for the valuable comments.

Action:

We have included a discussion on how microenvironmental factors (e.g., hypoxia)
might influence the observed epigenetic trajectories in the revised manuscript
(revised lines 520–531).

- Statistical Rigor:

**RESPONSE:** Thank you for the insightful comments. We answer these two points
one by one.

> The pseudotime trajectory analysis in Figures 4k and 5i lacks metrics for trajectory
robustness (e.g., bootstrapping confidence intervals). Please add statistical
validation of the branching patterns.

**RESPONSE:** Pseudotime trajectories were inferred using the Slingshot package in
R. Dimensionality reduction was performed on the single-cell expression matrix, and
cell clusters were used as input for Slingshot to define lineages.

To evaluate the robustness of lineage assignments and pseudotime ordering, we
applied a bootstrap resampling approach. Cells were resampled with replacement
1,000 times, and pseudotime inference was repeated for each bootstrap replicate
using the Slingshot framework. Lineages were inferred with the 'getLineages' and
'getCurves' functions, while cluster ordering along each lineage was obtained using
'slingLineages'. Pseudotime values for individual cells were then calculated with the
'slingPseudotime' function, generating a pseudotime matrix and all for downstream

summarization. For each bootstrap replicate, lineage branch support was defined as
 the proportion of bootstrap replicates in which a given lineage branch was recovered.
 Pseudotimes were re-aligned to the reference lineages and stored in a three-
 dimensional array (cells \times lineages \times bootstraps). To quantify pseudotime uncertainty,
 we summarized the bootstrap distribution for each cell and lineage. Specifically, the
 median pseudotime was calculated across all bootstrap replicates, and 95%
 confidence intervals (CIs) were obtained by taking the 2.5th and 97.5th percentiles of
 the bootstrap pseudotime distribution.

Rebuttal Figure 9: Bootstrap validation of Slingshot trajectories in human lung tumor.

a, Representative trajectories examples inferred from 1000 bootstrap replicates.

b, Branch support proportions calculated across 1,000 bootstrap replicates.

c, Bootstrap-derived pseudotime estimates for Slingshot lineages. Left: Lineage 1; Right: Lineage 2. Each row represents a single cell, ordered by its bootstrap median pseudotime. The horizontal position of each point indicates the median pseudotime estimate, with horizontal error bars representing the 95% confidence interval across 1,000 bootstrap replicates. Cells are colored according to their cluster assignments, illustrating the distribution of clusters along each trajectory.

For the tumor edge and tumore core samples in human lung:

Bootstrap analysis demonstrated the stability of the inferred trajectories, with
 Lineage 1 reproduced in 97.5% of bootstrap replicates and Lineage 2 in 93.5% of
 replicates (**Rebuttal Figure 9a, 9b**; revised **Supplementary Figure 17a, 17b**).

In the bootstrap pseudotime analysis (**Rebuttal Figure 9c**; revised **Supplementary**
 **Figure 17c**), Lineage 1 originates from Cluster 4 and progresses toward Cluster 1 at
 the latest pseudotime, while Lineage 2 branches from Cluster 4 into Cluster 2,
 representing an alternative differentiation trajectory. The median pseudotime
 estimates, together with their 95% confidence intervals across 1,000 bootstrap
 replicates, confirmed that the ordering of clusters along both lineages is robust.
 Although a subset of cells exhibited broader confidence intervals, the overall
 conclusions remained unaffected. These results indicate that the developmental
 progression from Cluster 4 toward either Cluster 1 or Cluster 2 is statistically well
 supported.

For the lymphoma samples, we performed the same analysis and observed similar
 reproducibility. (**Rebuttal Figure 10**; revised **Supplementary Figure 20**)

**Rebuttal Figure 10: Bootstrap validation of Slingshot trajectories in paired primary and relapse samples.**

a, Representative trajectories inferred from bootstrap replicates.

b, Branch support proportions calculated across 1,000 bootstrap replicates.

c, Bootstrap-derived pseudotime estimates for Slingshot lineages. Left: Lineage 1; Right: Lineage 2. Each row
 represents a single cell, ordered by its bootstrap median pseudotime. The horizontal position of each point indicates
 the median pseudotime estimate, with horizontal error bars representing the 95% confidence interval across 1,000
 bootstrap replicates. Cells are colored according to their cluster assignments, illustrating the distribution of clusters
 along each trajectory.

**ACTION:** We included the statistical calculations in the revised Methods, as well as
 the new data in the revised manuscript and in revised **Supplementary Figure 17** and
 **Figure 20**.

> For differential peak calling, specify how batch effects were controlled in the
 human samples (e.g., integration methods).

**RESPONSE:** Thank you for your comments.

For differential peak calling in human samples, we controlled for batch effects using
 the integration pipeline implemented in SnapATAC2, specifically the
 `snap.pp.harmony()` function to harmonize data across samples (see SnapATAC2
 integration tutorial <https://scverse.org/SnapATAC2/tutorials/integration.html>).

For cell type-specific peak identification, we adopted the pseudo-bulk strategy from
 ArchR, generating replicates by aggregating all cells within each cell type.

Differential peaks were then identified using a one-versus-rest comparison (e.g., T
 cells vs. all other cells).

For differential analysis between conditions, the tumor edge and tumor core regions
 in human lung samples were processed as a single batch, and similarly, for primary
 versus relapse samples in human lymphoma (patients 1 and 2), no batch differences
 were present between conditions.

**ACTION:** We included the detail description of the method in the revised manuscript.

- Technical Limitations:

Discuss the method's sensitivity limits (e.g., minimum input requirements,

performance in highly degraded samples) and failure rates.

**RESPONSE:** Thank you for the valuable comments.

**ACTION:** In the revised manuscript, we included a discussion of the limitations of the
method (revised **lines 818–838**).

Here is the discussion of the limitations in the revised manuscript:

A few limitations were observed in our current scFFPE-ATAC approach. The overall

recovery rate is approximately 20% after the split-and-pool step, and at least

250,000–500,000 nuclei are required to successfully perform the experiment. While

our study demonstrates applicability across lymphoid tissues and lung cancer,

broader validation in matrix-rich tumors such as pancreatic ductal adenocarcinoma

(PDAC) remains an important next step. In this study, we employed density gradient

centrifugation for nuclei isolation, which provided robust performance across the

FFPE samples we tested. As a future direction, FACS-based nuclei purification⁹⁻¹¹

could serve as a complementary strategy, particularly for challenging samples such

as PDAC, where higher purity may be beneficial. The quality of scFFPE-ATAC

libraries in our study did not depend on DNA fragmentation patterns in FFPE

samples (**Supplementary Technical Note 1**). At present, we rely on pre-

assessment of scFFPE-ATAC quality based on nuclei purity after isolation.

Establishing an additional pre-assessment protocol based on DNA quality would be

valuable—similar to single-cell RNA-seq in FFPE samples, where DV200

(Distribution Value 200; the percentage of RNA fragments >200 nucleotides)^{12,13} is

widely used as a quality-control metric prior to library preparation. Fixation time and

storage period in clinically archived samples also vary considerably. We successfully

tested samples fixed for 16–72 hours and stored for 6 months to 13 years, but further

optimization of protocols for samples with longer fixation (>72 hours) or storage

durations⁷ will be an important direction for future development. Finally, the current

workflow requires approximately five working days, and streamlining the protocol to

reduce turnaround time would greatly enhance its usability.

- Minor Typos:

Line 375: probably missed listing specific TFs

Line 936: extra comma

Figure 2a: typo chromatin

Figure 2b & 3a: Enrichment

Fig 3: Archived

Fig 5: Patient

**RESPONSE:** Thank you very much for your comments and suggestions to improve
the language in our manuscript. We apologize for the previous lapses in writing and
have carefully revised the text.

**ACTION:** We have corrected typographical errors and sought professional
assistance for language proofreading in the revised manuscript.

989 5. Methodology and Reproducibility

The methods are well-described overall, but the following additions would enhance
reproducibility:

- > Provide exact reagent catalog numbers for critical steps (e.g., T4 ligase).
- > Include representative images of nuclei pre/post-purification for different sample
- types.
- > Clarify computational parameters for key analyses (e.g., MACS2 peak calling
- settings, SnapATAC2 parameters).

**RESPONSE:** Thank you for the detail suggestion.

**ACTION:** We created a step-by-step protocol with detailed notes on Protocols.io
and included the link in the revised manuscript. The protocol also contains a
troubleshooting guide and highlights common pitfalls in the methods.

<https://www.protocols.io/private/3E7641CD72FF11F0AB4F0A58A9FEAC02>

We included representative images of nuclei before and after purification for each
sample type (mouse FFPE spleen, Revised **Supplementary Figure 1b**; human
FFPE lymph node, Revised **Supplementary Figure 11a**; human FFPE lung tumor,
Revised **Supplementary Figure 13a**; and human primary and relapse follicular
lymphoma, Revised **Supplementary Figures 18a and 18b**).

Additionally, we provided detailed computational parameters in the Methods section
and in our GitHub repository.

Reviewer #4 (Remarks on code availability):

The provided code (get_qc.r) is well documented within the script, with clear
parameter specifications for quality control of scFFPE-ATAC data, and it uses
standard R packages.

It successfully reproduces the key filtering and plotting steps described in the
manuscript. To further improve usability and reproducibility, the Authors should
consider:

- - adding a brief README that includes installation instructions, usage examples, and
especially a description or example of the expected input file format (such as the
required headers and column structure for Spleen.singlecell.qc.txt)
- - Replace hard-coded thresholds (e.g., 100000 fragments) with parameters and
explain biological rationale for default thresholds.
- - Consolidate duplicated FRiP/FRiT code blocks into functions
- - Add version requirements (e.g., ggplot2>=3.4.0)

RESPONSE: We thank the reviewer for the valuable suggestion. We agree that providing clear instructions is important to enable all potential users, including non-computational biologists, to use our pipeline effectively.

ACTION: In the revised manuscript, methods section, and shared GitHub repository, we have updated our code and instructions to enhance usability and reproducibility, including:

1. Added a README file to facilitate usage. We have updated the README file in our GitHub repository (<https://github.com/pengweixing/scFFPE>) to include detailed information on environment requirements, installation instructions, usage examples, and a description of the expected input file format, including the required headers and column structure.
2. Replaced hard-coded thresholds with adjustable parameters and provided a detailed explanation of the biological rationale for the default values. We have replaced hard-coded thresholds with user-defined parameters (min_num and max_num) in the updated version of the code. The default values are set to 1,000 for the minimum number of unique fragments and 100,000 for the maximum number of unique fragments. These defaults were chosen based on typical single-cell ATAC-seq quality control practices: cells with fewer than 1,000 fragments generally have insufficient coverage for downstream analysis, while those with more than 100,000 fragments may represent doublets or multiplets. Users can adjust these thresholds according to their experimental design.
3. We have refactored the code to consolidate the duplicated FRiP/FRiT calculation logic into reusable functions for improved readability and maintainability. The updated implementation can be found in the following scripts:
https://github.com/pengweixing/scFFPE/blob/main/tools/extract_cell.sh
https://github.com/pengweixing/scFFPE/blob/main/tools/get_single_cell_qc.sh
4. Specified the system requirements and package version requirements for the various packages and dependencies.

Key references:

- 1 Buenrostro, J. D., Giresi, P. G., Zaba, L. C., Chang, H. Y. & Greenleaf, W. J. Transposition of native chromatin for fast and sensitive epigenomic profiling of open chromatin, DNA-binding proteins and nucleosome position. *Nat Methods* **10**, 1213-1218, doi:10.1038/nmeth.2688 (2013).

- 2 Buenrostro, J. D. *et al.* Single-cell chromatin accessibility reveals principles of
regulatory variation. *Nature* **523**, 486-490, doi:10.1038/nature14590 (2015).
- 3 Corces, M. R. *et al.* An improved ATAC-seq protocol reduces background and enables
interrogation of frozen tissues. *Nat Methods* **14**, 959-962, doi:10.1038/nmeth.4396
(2017).
- 4 Chen, X. *et al.* ATAC-seq reveals the accessible genome by transposase-mediated
imaging and sequencing. *Nat Methods* **13**, 1013-1020, doi:10.1038/nmeth.4031
(2016).
- 5 Ma, S. *et al.* Chromatin Potential Identified by Shared Single-Cell Profiling of RNA and
Chromatin. *Cell* **183**, 1103-1116 e1120, doi:10.1016/j.cell.2020.09.056 (2020).
- 6 Glimelius, B. *et al.* U-CAN: a prospective longitudinal collection of biomaterials and
clinical information from adult cancer patients in Sweden. *Acta Oncol* **57**, 187-194,
doi:10.1080/0284186X.2017.1337926 (2018).
- 7 Hahn, E. E. *et al.* Century-old chromatin architecture revealed in formalin-fixed
vertebrates. *Nat Commun* **15**, 6378, doi:10.1038/s41467-024-50668-4 (2024).
- 8 Grandi, F. C., Modi, H., Kampman, L. & Corces, M. R. Chromatin accessibility profiling
by ATAC-seq. *Nat Protoc* **17**, 1518-1552, doi:10.1038/s41596-022-00692-9 (2022).
- 9 Satpathy, A. T. *et al.* Transcript-indexed ATAC-seq for precision immune profiling. *Nat*
*Med* **24**, 580-590, doi:10.1038/s41591-018-0008-8 (2018).
- 10 Wang, K. *et al.* Archival single-cell genomics reveals persistent subclones during DCIS
progression. *Cell* **186**, 3968-3982 e3915, doi:10.1016/j.cell.2023.07.024 (2023).
- 11 Chen, X. Q. *et al.* Joint single-cell DNA accessibility and protein epitope profiling
reveals environmental regulation of epigenomic heterogeneity. *Nat Commun* **9**,
doi:ARTN 4590
10.1038/s41467-018-07115-y (2018).
- 12 Xu, Z. Y., Lyu, Y. X., Chen, H. D., Chen, Y. & Wang, Y. C. Single-nucleus total RNA
sequencing of formalin-fixed paraffin-embedded samples using snRandom-seq. *Nat*
*Protoc*, doi:10.1038/s41596-025-01170-8 (2025).
- 13 Xu, Z. Y. *et al.* High-throughput single nucleus total RNA sequencing of formalin-fixed
paraffin-embedded tissues by snRandom-seq. *Nat Commun* **14**, doi:ARTN 2734
10.1038/s41467-023-38409-5 (2023).
- 14 Hagiwara, K. *et al.* Frequent DNA methylation but not mutation of the ID4 gene in
malignant lymphoma. *J Clin Exp Hematop* **47**, 15-18, doi:10.3960/jslrt.47.15 (2007).
- 15 Zheng, H. & Kang, Y. Multilayer control of the EMT master regulators. *Oncogene* **33**,
1755-1763, doi:10.1038/onc.2013.128 (2014).
- 16 Saitoh, M. Transcriptional regulation of EMT transcription factors in cancer. *Semin*
*Cancer Biol* **97**, 21-29, doi:10.1016/j.semcancer.2023.10.001 (2023).
- 17 Yang, J. *et al.* Guidelines and definitions for research on epithelial-mesenchymal
transition. *Nat Rev Mol Cell Biol* **21**, 341-352, doi:10.1038/s41580-020-0237-9 (2020).
- 18 Massagué, J. Metastasis initiating cells and ecosystems. *Cancer Res* **83**,
doi:10.1158/1538-7445.Metastasis22-la001 (2023).

Re: Revision of NCOMMS-25-22823C

Summary of response to all reviewers:

We would like to express our sincere gratitude to the reviewers for their thoughtful evaluation of our manuscript and for their constructive comments, which have been invaluable in improving the quality and clarity of our work. We are pleased to confirm that all reviewer comments have now been fully addressed.

REVIEWERS' COMMENTS and response

Reviewer #1 (Remarks to the Author):

All my concerns have been well addressed. Given the rigor of its experimental design and the persuasiveness of its data, this manuscript now meets a high standard of quality.

Thank you for your positive comments.

Reviewer #1 (Remarks on code availability):

The code is OK.

Thank you for your positive comments.

Reviewer #2 (Remarks to the Author):

Thank you for your positive comments.

Reviewer #3 (Remarks to the Author):

After reviewing the rebuttal, I am satisfied that the authors have addressed all of my concerns, and I have no further comments.

Thank you for your positive comments.

Reviewer #4 (Remarks to the Author):

I did not identify any outstanding concerns regarding the data analysis, interpretation, or conclusions in the revised version. The methodology meets the expected standards for reproducibility and robustness, and the reporting is complete. All reviewer points have been thoroughly addressed, with method clarification, protocol expansion, statistical robustness, QC metric transparency, figure revision, and code documentation now clearly visible in the resubmitted manuscript. In summary, the work represents an important and convincingly demonstrated technical advance that will be of broad interest for epigenomics, pathology, and translational cancer research.

Thank you for your positive comments.

Reviewer #4 (Remarks on code availability):

Reviewer requests for improved documentation and clarity on input formats have been addressed in this version, and with these updates, the code serves as a valuable and accessible resource for the community. The repository includes a README file with installation and usage instructions, and the major scripts are clearly organized. Quality control and cell selection workflows are well annotated, and example data and parameters are provided, which facilitates reproducibility of the main analyses presented in the manuscript. The expected format for key input files (such as the single-cell QC tables) is described, making it straightforward for users to adapt the pipeline to their own data.

Thank you for your positive comments.